# EQUIVARIANT DENOISERS CANNOT COPY GRAPHS: ALIGN YOUR GRAPH DIFFUSION MODELS

**Najwa Laabid**[1*]**, Severi Rissanen**[1*]**, Markus Heinonen**[1]**, Arno Solin**[1]**, Vikas Garg**[1,2]

[1]Department of Computer Science, Aalto University
[2]YaiYai Ltd

`{najwa.laabid, severi.rissanen, markus.o.heinonen}@aalto.fi,`
`arno.solin@aalto.fi, vgarg@csail.mit.edu`

## ABSTRACT

Graph diffusion models, dominant in graph generative modeling, remain underexplored for graph-to-graph translation tasks like chemical reaction prediction. We demonstrate that standard permutation equivariant denoisers face fundamental limitations in these tasks due to their inability to break symmetries in noisy inputs. To address this, we propose *aligning* input and target graphs to break input symmetries while preserving permutation equivariance in non-matching graph portions. Using retrosynthesis (i.e., the task of predicting precursors for synthesis of a given target molecule) as our application domain, we show how alignment dramatically improves discrete diffusion model performance from 5% to a SOTA-matching 54.7% top-1 accuracy. Code is available at https://github.com/Aalto-QuML/DiffAlign.

## 1 INTRODUCTION

Graphs appear ubiquitously across domains from knowledge representation to drug discovery (Ingraham et al., 2019; Ji et al., 2022; Hoogeboom et al., 2022b; Verma et al., 2023). Graph-to-graph translation serves numerous applications: molecular editing (Jin et al., 2019b), where the goal is generating molecular graphs with specific structures or properties; chemical reaction prediction (Shi et al., 2020b), which requires predicting reactant graphs from product graphs and vice-versa; and any task predicting future graph states from initial configurations (Rossi et al., 2020). Generative models offer a natural, probabilistic approach to graph-to-graph translation. Graph diffusion models present a particularly promising framework given their excellent performance on graph data (Niu et al., 2020; Vignac et al., 2023; Mercatali et al., 2024) and the widespread success of diffusion models in generative modeling in general.

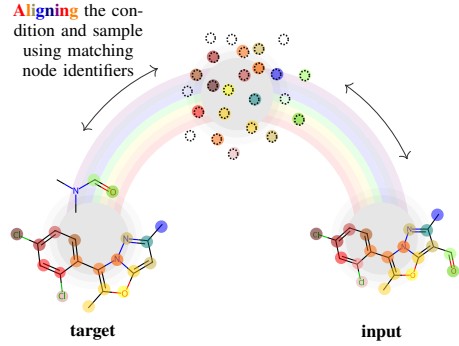

Figure 1: Nodes in the input and output graphs are given the same identifiers to enforce alignment, while the model is free to remain permutation equivariant w.r.t to the unmatched nodes.

Permutation equivariance forms a fundamental inductive bias in graph-based machine learning models, including graph diffusion models. This property ensures consistent model outputs when input nodes are reordered or relabeled, eliminating the need for data augmentation with different node permutations. While powerful, permutation equivariance faces fundamental limitations when mapping highly symmetric inputs to less symmetric outputs. We identify the root cause of this issue: when trying to satisfy equivariance with a symmetric input, an optimally-trained neural network can only predict identical distributions for all elements—essentially the marginal distribution of labels in the training data (see Theorem 1). Our findings align with previous and concurrent research highlighting

---

*Equal contribution

Figure 2: Overview of our contributions.

the limitations imposed by equivariance and symmetries in neural network (Lawrence et al., 2024; Xie & Smidt, 2024).

The key question becomes: how can we enable symmetry breaking while preserving the benefits of equivariance? We propose *aligned equivariance* as our solution, visualized in Fig. 1. This approach uses node identifiers in both input and target graphs to align corresponding nodes through various methods (see Sec. 3.3), relaxing equivariance constraints when identifiers match. This targeted symmetry breaking creates structural anchors that guide the generation process while maintaining permutation equivariance in non-matching subgraphs. Our experimental results shows that a combination of our alignment methods achieves SOTA-matching results on retrosynthesis, the task of predicting precursor molecules for a given target. We summarize our contributions in Fig. 2.

## 2 PRELIMINARIES

### 2.1 GRAPH-TO-GRAPH TRANSLATION

Consider a database of $N_{\text{obs}}$ graphs $\mathcal{D} = \{(\mathbf{X}_n, \mathbf{Y}_n, \mathbf{P}_n^{\mathbf{Y} \to \mathbf{X}})\}_{n=1}^{N_{\text{obs}}}$, where $\mathbf{X}_n$ represents the target graph, $\mathbf{Y}_n$ the input graph, and $\mathbf{P}_n^{\mathbf{Y} \to \mathbf{X}}$ are matrices defining *node mappings* between the two graphs. The *graph translation* task is: given that the data is sampled from an unknown distribution $p(\mathbf{X}, \mathbf{Y}, \mathbf{P}^{\mathbf{Y} \to \mathbf{X}})$, predict valid targets $\mathbf{X} \sim p(\mathbf{X} \mid \mathbf{Y})$ for a given input $\mathbf{Y}$.

We start by formally defining the graph objects $\mathbf{X}$ and $\mathbf{Y}$. To do so, we consider the space of one-hot vectors of dimension $K$ as $\text{vert}(\Delta^{K-1})$, where $\text{vert}(.)$ denotes the vertices of the probability simplex $\Delta^{K-1}$ with $K-1$ degrees of freedom. We then define the target $\mathbf{X}$ as a tuple $(\mathbf{X}^{\mathcal{N}}, \mathbf{X}^E)$ of a node feature matrix $\mathbf{X}^{\mathcal{N}} \in (\text{vert}(\Delta^{K_a-1}))^{N_X}$ and an edge feature matrix $\mathbf{X}^E \in (\text{vert}(\Delta^{K_b-1}))^{N_X \times N_X}$, where $K_a$ and $K_b$ are node and edge feature dimensions respectively. The input graph $\mathbf{Y}$ is similarly defined as $(\mathbf{Y}^{\mathcal{N}}, \mathbf{Y}^E)$ with the same node and edge features, but potentially a different number of nodes $N_Y$. The features consist of node labels (with $K_a$ possible values) and edge labels (with $K_b$ possible values), each including an empty value $\perp$ to represent missing nodes or edges.

Node mapping matrices establish correspondence between nodes in the input and target graphs. Formally, we define such matrices as: $\mathbf{P}^{\mathbf{Y} \to \mathbf{X}} \in \{0,1\}^{N_X \times N_Y}$, with $\mathbf{P}_{i,j}^{\mathbf{Y} \to \mathbf{X}} = 1$ if the $i^{\text{th}}$ node of the target is the atom-mapped counterpart of the $j^{\text{th}}$ node of the input, and zero otherwise. This node mapping plays a crucial role in breaking the inherent symmetries present in graph diffusion models. Without this correspondence information, a model cannot distinguish which specific nodes in the input should map to which specific nodes in the output, leading to averaged, symmetrical predictions that fail to capture the structural relationships between input and target graphs, as we discuss later.

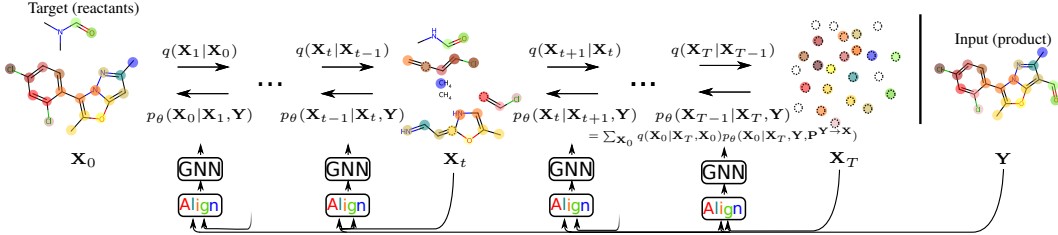

Figure 3: An overview of a graph translation task using discrete diffusion models, illustrated through chemical reactions. We adopt *absorbing state diffusion* (Austin et al., 2021), such that samples from the stationary distribution are made of nodes and edges with the 'none' label. The condition and sample graphs are aligned using node mapping information (highlighted with the matching colours) which identifies pairs of matched nodes across the input and target graphs.

The product $\mathbf{P}^{\mathbf{Y}\rightarrow\mathbf{X}}\mathbf{Y}^{\mathcal{N}}$ equals $\mathbf{X}^{\mathcal{N}}$ with the non-mapped nodes zeroed out. Similarly, $\mathbf{P}^{\mathbf{Y}\rightarrow\mathbf{X}}\mathbf{Y}^{E}(\mathbf{P}^{\mathbf{Y}\rightarrow\mathbf{X}})^{\top}$ equals $\mathbf{X}^{E}$ with edges to non-mapped nodes zeroed out. We expect an inductive bias such that if an edge exists between two mapped nodes in $\mathbf{X}$, there is a high probability of an edge between their corresponding nodes in $\mathbf{Y}$ as well. This reflects the fundamental intuition that graphs are structurally similar for their node-mapped portions.

## 2.2 DISCRETE DIFFUSION MODELS FOR CONDITIONAL GRAPH GENERATION

We follow the framework of Vignac et al. (2023), which adapts discrete diffusion models (Austin et al., 2021) to graphs. We present the main constituents of the framework. We assume a Markov *forward* process

$$q(\mathbf{X}_{t+1}\,|\,\mathbf{X}_t) = \prod_{i=1}^{N_X} q(\mathbf{X}_{t+1}^{\mathcal{N},i}\,|\,\mathbf{X}_t^{\mathcal{N},i}) \prod_{i,j=1}^{N_X} q(\mathbf{X}_{t+1}^{E,ij}\,|\,\mathbf{X}_t^{E,ij}), \tag{1}$$

to diffuse the reactant to noise, and a reverse process

$$p_\theta(\mathbf{X}_{t-1}\,|\,\mathbf{X}_t,\mathbf{Y}) = \prod_{i=1}^{N_X} p_\theta(\mathbf{X}_{t-1}^{\mathcal{N},i}\,|\,\mathbf{X}_t,\mathbf{Y}) \prod_{i,j}^{N_X} p_\theta(\mathbf{X}_{t-1}^{E,ij}\,|\,\mathbf{X}_t,\mathbf{Y}), \tag{2}$$

defining our generative model. Here, $\theta$ represents neural network parameters. Note that we always have time conditioning implicitly in $p_\theta(\mathbf{X}_{t-1}\,|\,\mathbf{X}_t,\mathbf{Y},t)$, but we drop $t$ for notational convenience. We will also condition on the node mapping $\mathbf{P}^{\mathbf{Y}\rightarrow\mathbf{X}}$ in Sec. 3.3, but we will not include it in the notation until then. The full generative distribution is $p_\theta(\mathbf{X}_{0:T}\,|\,\mathbf{Y}) = p(\mathbf{X}_T) \prod_{t=1}^{T} p_\theta(\mathbf{X}_{t-1}\,|\,\mathbf{X}_t,\mathbf{Y})$, where $p(\mathbf{X}_T)$ is a predefined prior such that $p(\mathbf{X}_T) = q(\mathbf{X}_T\,|\,\mathbf{X}_0)$. Following Hoogeboom et al. (2021) and Austin et al. (2021), we use the neural network specifically to predict ground truth labels from noised samples, meaning that the neural network outputs a distribution $\tilde{p}_\theta(\mathbf{X}_0\,|\,\mathbf{X}_t,\mathbf{Y})$.

The reverse process is then parameterized by

$$p_\theta(\mathbf{X}_{t-1}\,|\,\mathbf{X}_t,\mathbf{Y}) = \sum_{\mathbf{X}_0} q(\mathbf{X}_{t-1}\,|\,\mathbf{X}_t,\mathbf{X}_0)\tilde{p}_\theta(\mathbf{X}_0\,|\,\mathbf{X}_t,\mathbf{Y}). \tag{3}$$

Both $q(\mathbf{X}_{t-1}\,|\,\mathbf{X}_t,\mathbf{X}_0)$ and $\tilde{p}_\theta(\mathbf{X}_0\,|\,\mathbf{X}_t,\mathbf{Y})$ factorize over dimensions:

$$q(\mathbf{X}_{t-1}\,|\,\mathbf{X}_t,\mathbf{X}_0) = \prod_i^{N_\mathbf{X}} q(\mathbf{X}_{t-1}^{\mathcal{N},i}\,|\,\mathbf{X}_t^{\mathcal{N},i},\mathbf{X}_0^{\mathcal{N},i}) \prod_{i,j}^{N_\mathbf{X}} q(\mathbf{X}_{t-1}^{E,i,j}\,|\,\mathbf{X}_t^{E,i,j},\mathbf{X}_0^{E,i,j}) \tag{4}$$

$$\tilde{p}_\theta(\mathbf{X}_0\,|\,\mathbf{X}_t,\mathbf{Y}) = \prod_i^{N_\mathbf{X}} \tilde{p}_\theta(\mathbf{X}_0^{\mathcal{N},i}\,|\,\mathbf{X}_t,\mathbf{Y}) \prod_{i,j}^{N_\mathbf{X}} \tilde{p}_\theta(\mathbf{X}_0^{E,i,j}\,|\,\mathbf{X}_t,\mathbf{Y}) \tag{5}$$

We write the connection to Eq. (2) explicitly in App. B. Throughout the paper, we denote the direct output of the neural network as $D_\theta(\mathbf{X}_t,\mathbf{Y}) = (D_\theta(\mathbf{X}_t,\mathbf{Y})^{\mathcal{N}}, D_\theta(\mathbf{X}_t,\mathbf{Y})^{E})$, where $D_\theta(\mathbf{X}_t,\mathbf{Y})^{\mathcal{N}} \in (\Delta^{K_a-1})^{N_X}$ and $D_\theta(\mathbf{X}_t,\mathbf{Y})^{E} \in (\Delta^{K_b-1})^{(N_X \times N_X)}$, i.e., we have a probability vector for each node and edge. This implies $\tilde{p}_\theta(\mathbf{X}_0\,|\,\mathbf{X}_t,\mathbf{Y})$ is a distribution factorized over nodes and edges.

The single-step transition for nodes (and similarly for edges) is defined with a transition matrix $\mathbf{Q}_t^{\mathcal{N}}$:

$$q(\mathbf{X}_t^{\mathcal{N},i}\,|\,\mathbf{X}_{t-1}^{\mathcal{N},i}) = \text{Cat}(\mathbf{X}_t^{\mathcal{N},i}; \mathbf{p} = \mathbf{X}_{t-1}^{\mathcal{N},i}\mathbf{Q}_t^{\mathcal{N}}). \tag{6}$$

**Algorithm 1** Loss calculation

**Input:** condition $\mathbf{Y}$, target $\mathbf{X}_0$, and optional permutation matrix $\mathbf{P}^{\mathbf{X}\to\mathbf{Y}}$ for alignment
$t \sim \text{Uniform}(\{0,\dots,T\})$
$\mathbf{X}_t \sim q(\mathbf{X}_t \,|\, \mathbf{X}_0)$
$\tilde{\mathbf{X}}_0 = D_\theta(\mathbf{X}_t, \mathbf{Y}, \mathbf{P}^{\mathbf{Y}\to\mathbf{X}})$
**Return Cross-Entropy($\mathbf{X}_0, \tilde{\mathbf{X}}_0$)**

**Algorithm 2** Sampling

**Input:** condition $\mathbf{Y}$
**Choose (for alignment):** $\mathbf{P}^{\mathbf{Y}\to\mathbf{X}} \in \mathbb{R}^{N_{\mathbf{X}} \times N_{\mathbf{Y}}}$
$\mathbf{X}_T \propto p(\mathbf{X}_T)$
**for** $t = T$ **to** $1$ **do**
$\quad \tilde{\mathbf{X}}_0 = D_\theta(\mathbf{X}_t, \mathbf{Y}, \mathbf{P}^{\mathbf{Y}\to\mathbf{X}})$
$\quad \mathbf{X}_{t-1}^i \sim \sum_k q(\mathbf{X}_{t-1}^i \,|\, \mathbf{X}_t^i, \mathbf{X}_0^i)\tilde{\mathbf{X}}_0^i$
**Return $\mathbf{X}_0$**

In our experiments, we use the absorbing-state formulation from Austin et al. (2021), where nodes and edges gradually transfer to the *absorbing state*, defined as the empty state $\perp$. Formally, $\mathbf{Q}_t = (1 - \beta_t)I + \beta_t \mathbb{1}e_\perp^\top$, where $\beta_t$ defines the *diffusion schedule* and $e_\perp$ is one-hot on the absorbing state. Then, the marginal $q(\mathbf{X}_t \,|\, \mathbf{X}_0)$ and conditional posterior $q(\mathbf{X}_{t-1} \,|\, \mathbf{X}_t, \mathbf{X}_0)$ also have a closed form for the absorbing state transitions. The prior $p(\mathbf{X}_T)$ is correspondingly chosen to be a delta distribution at a graph with no edges and nodes set to the $\perp$ state. The noise schedule $\beta_t$ is defined using the mutual information criterion proposed in Austin et al. (2021). While other transitions, like uniform and marginal (Vignac et al., 2023) are equally possible, the absorbing state model is simpler and empirically outperforms others (Austin et al., 2021; Lou et al., 2024), motivating our choice.

We use the cross-entropy loss, as discussed in Austin et al. (2021) and Vignac et al. (2023):

$$-\mathbb{E}_{q(\mathbf{X}_0, \mathbf{Y})q(t)q(\mathbf{X}_t \,|\, \mathbf{X}_0)}[\log \tilde{p}_\theta(\mathbf{X}_0 \,|\, \mathbf{X}_t, \mathbf{Y})], \qquad (7)$$

where $q(t)$ is a uniform distribution over $t \in \{1\dots T\}$.

Fig. 3 provides an overview of the conditional graph diffusion framework we use. For completeness, a comprehensive definition of the model's components for nodes and edges is given in App. B. The training and sampling procedures with graph diffusion models are presented in Alg. 1 and Alg. 2, along with optional conditioning on $\mathbf{P}^{\mathbf{Y}\to\mathbf{X}}$, as described in Sec. 3.3.

**Permutation Equivariance and Invariance**  Permutation equivariance for the denoiser is defined as $D_\theta(\mathbf{P}\mathbf{X}) = \mathbf{P}D_\theta(\mathbf{X})$ where $\mathbf{P}$ is an arbitrary permutation matrix, and $\mathbf{P}\mathbf{X} = (\mathbf{P}\mathbf{X}^{\mathcal{N}}, \mathbf{P}\mathbf{X}^E\mathbf{P}^\top)$. This makes single-step reverse transitions equivariant as well, in the sense that $p_\theta(\mathbf{X}_{t-1} \,|\, \mathbf{X}_t) = p_\theta(\mathbf{P}\mathbf{X}_{t-1} \,|\, \mathbf{P}\mathbf{X}_t)$. For our conditional setting, permutation equivariance can be written as $D_\theta(\mathbf{P}\mathbf{X}, \mathbf{Y}) = \mathbf{P}D_\theta(\mathbf{X}, \mathbf{Y})$. The prior in graph diffusion models is also usually permutation invariant s.t. $p(\mathbf{P}\mathbf{X}_T) = p(\mathbf{X}_T)$ (Niu et al., 2020; Vignac et al., 2023; Hoogeboom et al., 2022b).

## 2.3 RELATED WORK

**Graph-to-graph translation methods**  Graph-to-graph models have demonstrated success in diverse tasks including handwritten mathematical expression recognition (Wu et al., 2021), molecular optimization (Jin et al., 2019a), and retrosynthesis (Lin et al., 2023). Recently, Igashov et al. (2024) introduced a Markov bridge model for product-to-reactant graph mapping, representing the closest approach to diffusion-based graph-to-graph translation. While their paper does not explicitly address equivariance and symmetry-breaking as design elements, the authors implement a technique similar to our "input alignment" described in Sec. 3.3. This suggests our theoretical framework may also explain their model's effectiveness.

**Equivariance and symmetry-breaking**  Equivariant neural networks have attracted interest for their ability to incorporate known symmetries, typically enhancing generalization. However, Smidt et al. (2021) identified a fundamental limitation: these networks struggle with self-symmetric inputs because they cannot break inherent data symmetries. This challenge appears across various applications, including prediction tasks on symmetric domains and generative models reconstructing from highly symmetric latent spaces (Lawrence et al., 2024). Researchers have explored this issue in different contexts, including graph representation learning (Srinivasan & Ribeiro, 2020), set generation (Zhang et al., 2022), and physical system modeling (Kaba et al., 2023).

**Permutation equivariance in graph diffusion**  Equivariance features prominently in graph diffusion models to parameterize the reverse process (Niu et al., 2020; Vignac et al., 2023; Hoogeboom et al., 2022b; Huang et al., 2022). One key motivation is that permutation equivariant neural networks induce permutation invariant distributions in diffusion models (Niu et al., 2020), ensuring different permutations of the same graph have identical probabilities under the model. In recent work,

Yan et al. (2023) demonstrated that relaxing permutation equivariance in graph diffusion through absolute positional encodings empirically improves performance.

# 3 WHY PERMUTATION EQUIVARIANCE FAILS AND HOW TO FIX IT

## 3.1 PERMUTATION EQUIVARIANT DENOISERS CANNOT LEARN THE IDENTITY FUNCTION

In this section, we consider a data set $\mathcal{D} = \{(\mathbf{X}_n, \mathbf{Y}_n, \mathbf{P}_n^{\mathbf{Y} \to \mathbf{X}})\}_{n=1}^{N_{\text{obs}}}$, dubbed the 'identity data', where for all data points $\mathbf{X}_n = \mathbf{P}_n^{\mathbf{Y} \to \mathbf{X}} \mathbf{Y}_n$. In other words, both the input and the target are equivalent, up to some permutation, as defined by the node-mapping matrix $\mathbf{P}_n^{\mathbf{Y} \to \mathbf{X}}$. This seemingly simple scenario reveals a fundamental limitation of standard graph diffusion models. Graph translation can be viewed as similar to copying graphs (learning the identity function) because of the structural similarity between input and target graphs. Yet, we demonstrate that standard graph diffusion models struggle with this basic task due to their inability to break symmetries.

A denoiser implementing the identity function should output $D_\theta(\mathbf{X}_t, \mathbf{Y}) = \mathbf{P}^{\mathbf{Y} \to \mathbf{X}} \mathbf{Y}$, placing all probability mass on the correct node/edge labels (up to a permutation of $\mathbf{Y}$). We pass the conditioning graph $\mathbf{Y}$ by concatenating it to the input $\mathbf{X}$ along the node dimension, creating a graph with $N_{\mathbf{X}} + N_{\mathbf{Y}}$ nodes and $(N_{\mathbf{X}} + N_{\mathbf{Y}}) \times (N_{\mathbf{X}} + N_{\mathbf{Y}})$ edges, with no edges between the $\mathbf{X}$ and $\mathbf{Y}$ subgraphs. However, permutation equivariant denoisers $D_\theta$ are constrained to output identical "mean" solutions for all nodes and edges in the early stages of generation.

To understand this intuitively, consider a highly noisy input at $t = T$ and a permutation equivariant denoiser. Permutation equivariance requires that $D_\theta(\mathbf{R}\mathbf{X}_T, \mathbf{Y}) = \mathbf{R}D_\theta(\mathbf{X}_T, \mathbf{Y})$ for any permutation matrix $\mathbf{R} \in \{0, 1\}^{N_Y \to N_X}$. Since $\mathbf{X}_T$ contains no information about $\mathbf{Y}$ (due to high noise), the model learns to ignore the $\mathbf{X}$ input, leading to $D_\theta(\mathbf{R}\mathbf{X}_T, \mathbf{Y}) = D_\theta(\mathbf{X}_T, \mathbf{Y}) = \mathbf{P}^{\mathbf{Y} \to \mathbf{X}} \mathbf{Y}$. This creates a contradiction: $\mathbf{R}\mathbf{P}^{\mathbf{Y} \to \mathbf{X}} \mathbf{Y} = \mathbf{P}^{\mathbf{Y} \to \mathbf{X}} \mathbf{Y}$, which cannot generally be true. The only way to satisfy both equivariance ($D_\theta(\mathbf{R}\mathbf{X}_T, \mathbf{Y}) = \mathbf{R}D_\theta(\mathbf{X}_T, \mathbf{Y})$) and input-independence ($D_\theta(\mathbf{R}\mathbf{X}_T, \mathbf{Y}) = D_\theta(\mathbf{X}_T, \mathbf{Y})$) is if $D_\theta(\mathbf{X}_T, \mathbf{Y})$ outputs identical probability vectors for each node and edge. This symmetry problem prevents the model from distinguishing between nodes that should have different labels. The following theorem formalizes this limitation:

**Theorem 1. (The optimal permutation equivariant denoiser)** *Let $D_\theta(\mathbf{X}_T, \mathbf{Y})$ be permutation equivariant s.t. $D_\theta(\mathbf{P}\mathbf{X}_T, \mathbf{Y}) = \mathbf{P}D_\theta(\mathbf{X}_T, \mathbf{Y})$, and let $q(\mathbf{X}_T)$ be permutation invariant. The optimal solution with respect to the cross-entropy loss with the identity data is, for all nodes $i$ and $j$*

$$\begin{cases} D_\theta(\mathbf{X}_T, \mathbf{Y})_{i,:}^{\mathcal{N}} = \hat{\mathbf{y}}^{\mathcal{N}}, \hat{\mathbf{y}}_k^{\mathcal{N}} = \sum_i \mathbf{Y}_{i,k} / \sum_{i,k} \mathbf{Y}_{i,k}, \\ D_\theta(\mathbf{X}_T, \mathbf{Y})_{i,j,:}^{E} = \hat{\mathbf{y}}^{E}, \hat{\mathbf{y}}_k^{E} = \sum_{i,j} \mathbf{Y}_{i,j,k} / \sum_{i,j,k} \mathbf{Y}_{i,j,k}, \end{cases} \qquad (8)$$

*where $\hat{\mathbf{y}}_k^{\mathcal{N}}$ and $\hat{\mathbf{y}}_k^{E}$ are the marginal distributions of node and edge values in $\mathbf{Y}$.*

We visualize the theorem in Fig. 4, and give the proof in App. A.2, with an extension to other time steps in App. A.7. The symmetry constraint makes it impossible for the model to solve the task in a single step with $T = 1$. With multiple smaller steps, the model can gradually break symmetries as sampling in the discrete generative process introduces slight asymmetries in $\mathbf{X}_t$, enabling correlations between node and edge representations to emerge. Each denoising step can leverage these small asymmetries from previous steps, slowly building node identity information. As $T \to \infty$ with absorbing diffusion, the model eventually becomes equivalent to an any-order autoregressive model (Hoogeboom et al., 2022a), thus exemplifying gradual symmetry-breaking by changing one element at a time. However, this process remains inefficient compared to approaches that directly address the symmetry problem.

## 3.2 SOLUTION: ALIGNED PERMUTATION EQUIVARIANCE

Given the failure of permutation equivariant denoisers to effectively copy graphs, we propose to relax the permutation equivariance constraint in a controlled way. We observe that it is sufficient to have permutation equivariance in the sense that *if we permute $\mathbf{X}$ and/or $\mathbf{Y}$, and accordingly permute the node mapping $P^{\mathbf{Y} \to \mathbf{X}}$ such that the matching between $\mathbf{Y}$ and $\mathbf{X}$ remains, the model output should be the same*. We call this *aligned permutation equivariance*. Formally, we use the node-mapping permutation matrix $\mathbf{P}^{\mathbf{Y} \to \mathbf{X}}$ as an input to the denoiser and consider denoisers that

satisfy the following constraint: $D_\theta(\mathbf{RX}, \mathbf{QY}, \mathbf{RP^{Y \to X}Q^\top}) = \mathbf{R}D_\theta(\mathbf{X}, \mathbf{Y}, \mathbf{P^{Y \to X}})$, where $\mathbf{R}$ and $\mathbf{Q}$ are permutation matrices of shapes $(N_{\mathbf{X}} \times N_{\mathbf{X}})$ and $(N_{\mathbf{Y}} \times N_{\mathbf{Y}})$, respectively.

With $\mathbf{P^{Y \to X}}$ and $\mathbf{Y}$ as input, an unconstrained denoiser can output the ground-truth permutation $\mathbf{P^{Y \to X}Y}$ for the identity data task. Restricting the function to the aligned equivariant class does not clash with this, as can be seen by writing out the optimal solution and the equivariance condition for a permuted input $\mathbf{RX}, \mathbf{QY}, \mathbf{RP^{Y \to X}Q^\top}$:

$$\text{(Identity data)} \quad D_\theta(\mathbf{RX}, \mathbf{QY}, \mathbf{RP^{Y \to X}Q^\top}) = \mathbf{RP^{Y \to X}Q^\top QY} = \mathbf{RP^{Y \to X}Y}, \quad (9)$$

$$\text{(Aligned Equivariance)} \quad D_\theta(\mathbf{RX}, \mathbf{QY}, \mathbf{RP^{Y \to X}Q^\top}) = \mathbf{R}D_\theta(\mathbf{X}, \mathbf{Y}, \mathbf{P^{Y \to X}}) = \mathbf{RP^{Y \to X}Y}. \quad (10)$$

Note that the permutation invariance of $p_\theta(\mathbf{X}_0)$ does not apply as it does for fully permutation equivariant models. However, we show that a generalized form of distribution invariance holds.

**Theorem 2. (Aligned denoisers induce aligned permutation invariant distributions)** *If the denoiser function $D_\theta$ has the aligned equivariance property and the prior $p(\mathbf{X}_T)$ is permutation invariant, then the generative distribution $p_\theta(\mathbf{X}_0 \,|\, \mathbf{Y}, \mathbf{P^{Y \to X}})$ has the corresponding property for any permutation matrices $\mathbf{R}$ and $\mathbf{Q}$:*

$$p_\theta(\mathbf{RX}_0 \,|\, \mathbf{QY}, \mathbf{RP^{Y \to X}Q^\top}) = p_\theta(\mathbf{X}_0 \,|\, \mathbf{Y}, \mathbf{P^{Y \to X}}). \quad (11)$$

A proof is given in App. A.4, while Figure 4 showcases the denoising output of an aligned equivariant model with a given $\mathbf{P^{Y \to X}}$. Informally, the theorem states that the graph pairing has the same probability for all isomorphisms of the input and target graphs, as long as the node mapping is not reassigned to different nodes. This means that during training, we can use the permutations present in the data and be confident that the model generalizes to other permutations. During sampling, we only have access to $\mathbf{Y}$ without node mapping information. The theorem ensures we can assign node mappings arbitrarily and still obtain effectively the same distribution over the targets. We show this mathematically in App. A.4. Lastly, we note that our use of node mapping here is not a constraint: it is merely an additional input which allows the denoiser to use the structure of the conditional graph efficiently, and as such we benefit even from a partially correct node mapping.

### 3.3 METHODS FOR ALIGNMENT IN GNNs

We have established how constraining permutation equivariance to *aligned* permutation equivariance is a key component for the success of the denoising model. In this section, we discuss multiple methods to induce an alignment between the input and target graphs. We visualize these methods in Fig. A9 and prove that each belongs to the class of aligned equivariant models in App. A.5.

**Node-mapped positional encodings** Graph positional encodings are a standard tool to break permutation symmetry in GNNs, making them a natural candidate for inducing aligned permutation equivariance. We consider uniquely identifying pairs of nodes matched via the node-mapping matrix in both $\mathbf{X}_t$ and $\mathbf{Y}$ by adding a positional encoding vector to each unique node pair. In practice, we generate a set of distinct vectors $\varphi \in \mathbf{R}^{N_Y \times d_\varphi}$ such that $\varphi = g(\mathbf{Y})$ for each of the input graph nodes $\mathbf{Y}$, with $g$ a function generating the encodings based on $\mathbf{Y}$. In this work, we use Laplacian positional encodings (Dwivedi et al., 2023), but in principle any positional encoding scheme is applicable. We then map the vectors $\phi$ to the corresponding inputs for the noisy graph nodes $\mathbf{X}_t$. Formally, we get $D_\theta(\mathbf{X}_t, \mathbf{Y}, \mathbf{P^{Y \to X}}) =$

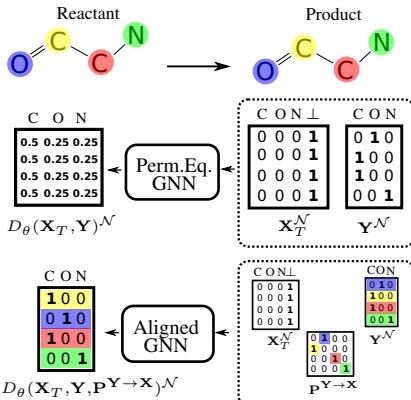

Figure 4: Comparing the optimal permutation equivariant denoiser to an aligned denoiser. As per Theorem 1, the permutation equivariant model (*top*) outputs the marginal distribution over node types whereas the aligned model (*bottom*) reconstructs the correct reactant nodes in a specific permutation $\mathbf{P^{Y \to X}}$.

$f_\theta([\mathbf{X}_t^\mathcal{N} \, \mathbf{P^{Y \to X}}\varphi], \mathbf{X}_t^E, [\mathbf{Y}^\mathcal{N} \, \varphi], \mathbf{Y}^E)$, where $f_\theta$ is the neural network that takes as inputs the augmented node features and regular edge features and $[\mathbf{X}_t^\mathcal{N} \, \mathbf{P^{Y \to X}}\varphi] \in \mathbf{R}^{N_X \times (K_a + d_\varphi)}$ corresponds

to concatenation along the feature dimension. The only change required for the neural network is to increase the initial linear layer size for the node inputs.

**Direct skip connection**    Next, motivated by the identity function analysis in Sec. 3.1, we propose an alignment method that solves the identity task in a minimal way. In particular, we modify the network to include a direct connection from the condition to the target output: $D_\theta(\mathbf{X}_t, \mathbf{Y}, \mathbf{P}^{\mathbf{Y} \to \mathbf{X}}) = \mathrm{softmax}(f_\theta^{\mathrm{logit}}(\mathbf{X}_t, \mathbf{Y}) + \lambda \mathbf{P}^{\mathbf{Y} \to \mathbf{X}} \mathbf{Y})$ where $f_\theta^{\mathrm{logit}}(\mathbf{X}_t, \mathbf{Y})$ are the logits at the last layer of the neural network for the nodes and edges, $\lambda$ is a learnable parameter and $\mathbf{P}^{\mathbf{Y} \to \mathbf{X}} \mathbf{Y} = (\mathbf{P}^{\mathbf{Y} \to \mathbf{X}} \mathbf{Y}^{\mathcal{N}}, \mathbf{P}^{\mathbf{Y} \to \mathbf{X}} \mathbf{Y}^E (\mathbf{P}^{\mathbf{Y} \to \mathbf{X}})^\top)$. The sum is possible because $\mathbf{Y}$ is in one-hot format and the dimensionalities of the denoiser output and $\mathbf{P}^{\mathbf{Y} \to \mathbf{X}} \mathbf{Y}$ are the same. It is easy to see that when $\lambda \to \infty$, $D_\theta(\mathbf{X}_t, \mathbf{Y}, \mathbf{P}^{\mathbf{Y} \to \mathbf{X}}) \to \mathbf{P}^{\mathbf{Y} \to \mathbf{X}} \mathbf{Y}$ because the direct connection from $\mathbf{Y}$ dominates in the softmax.

**Aligning $\mathbf{Y}$ in the input**    As a natural next step from aligning the graphs at the output via skip connections, we propose to align them in the input. This allows more expressivity from the network, since it can process the aligned graphs in the layers of the network before outputting the denoising solution. In practice, we align $\mathbf{Y}$ by concatenating $\mathbf{X}$ and $\mathbf{P}^{\mathbf{Y} \to \mathbf{X}} \mathbf{Y}$ along the feature dimension before passing it to the neural network. Thus, we can drop out the explicit $\mathbf{Y}$ graph entirely, and only process the $\mathbf{X}$ graph augmented with $\mathbf{Y}$: $D_\theta(\mathbf{X}_t, \mathbf{Y}, \mathbf{P}^{\mathbf{Y} \to \mathbf{X}}) = f_\theta([\mathbf{X}_t^{\mathcal{N}} \quad \mathbf{P}^{\mathbf{Y} \to \mathbf{X}} \mathbf{Y}^{\mathcal{N}}], [\mathbf{X}_t^E \quad \mathbf{P}^{\mathbf{Y} \to \mathbf{X}} \mathbf{Y}^E (\mathbf{P}^{\mathbf{Y} \to \mathbf{X}})^\top])$, where $[\cdot]$ means concatenation along the feature dimension.

### 3.4    ADDING POST-TRAINING CONDITIONING FOR DISCRETE DIFFUSION MODELS

To fully realize the potential of diffusion models for graph-to-graph translation, we derive a novel reconstruction guidance-like method for discrete models (Ho et al., 2022; Chung et al., 2023; Song et al., 2023a). This approach differs from the classifier-guidance used by Vignac et al. (2023) in their conditional generation. We use the denoiser output to evaluate the likelihood $p(y \mid \mathbf{X}_0)$ of the condition $y$ (e.g., probability of synthesizability) given the reactants $\mathbf{X}_0$, and backpropagate to get an adjustment of the backward update step:

$$\log \mathbf{P}_\theta(\mathbf{X}_{t-1} \mid \mathbf{X}_t, \mathbf{Y}, y) \propto \nabla_{\mathbf{X}_t} \log(\mathbb{E}_{p_\theta(\mathbf{X}_0 \mid \mathbf{X}_t, \mathbf{Y})} p(y \mid \mathbf{X}_0)) + \log \mathbf{P}_\theta(\mathbf{X}_{t-1} \mid \mathbf{X}_t, \mathbf{Y}). \quad (12)$$

The derivation starts with similar steps as the classifier guidance method in Vignac et al. (2023). We provide more details in App. F. The method can be implemented with a few lines of code and works as long as $p(y \mid \mathbf{X}_0)$ is differentiable. For the retrosynthesis application, we experiment with a toy model of synthesizability based on the count of atoms in the reaction (Ertl & Schuffenhauer, 2009), a concept also known as atom economy (Trost, 1991).

## 4    EXPERIMENTS

We assess the effect of aligned denoisers on the performance of regular discrete diffusion models. To this effect, we first demonstrate our model on a toy example: copying simple graphs. We then evaluate our method more rigorously on the real-world task of retrosynthesis, the task of defining precursors for a given compound. We also show how diffusion enables a number of downstream tasks within retrosynthesis, including inpainting and property-guided generation.

### 4.1    COPYING GRAPHS

We evaluate the model's ability to copy a graph as it is. To do so, we define a simple dataset made of $5 \times 5$ grids. We define an aligned and an equivariant denoiser, both with the same architecture: a 2-layer GNN with 16 nodes in the hidden dimension. We train both models on the same 100 samples with a batch size of 32 for 10 epochs. Figure 5 shows that our aligned denoiser recovers the original graph almost perfectly, while a fully equivariant denoiser outputs only a fraction of the same components.

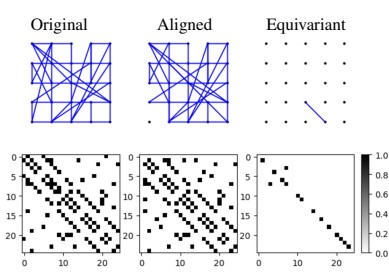

Figure 5: Showcasing performance on graph copying of grids.

## 4.2 RETROSYNTHESIS

Retrosynthesis is a crucial step in the drug discovery pipeline, as it provides a concrete plan to create an identified compound. Single-step retrosynthesis in particular seeks to define the reactants in a single chemical reaction, which can then be chained to create a more comprehensive synthesis plan. The task lends itself naturally to graph translation models, since it seeks to generate a target graph (reactants) given an input graph (product). In this experiment, we compare the performance of the aligned denoiser to that of an equivariant denoiser using real-world chemical reactions.

**Experimental setup** We use the benchmark dataset USPTO-50k for our experiments. The dataset consists of 50000 chemical reactions, in SMILES format (Weininger, 1988), curated by Schneider et al. (2016) from an original 2 million reactions extracted through text mining by Lowe (2012). More information on the benchmark dataset USPTO and its various subsets can be found in App. C.2. We use the graph transformer architecture introduced by Dwivedi & Bresson (2021) and used by Vignac et al. (2023). The network architecture and hyperparameters are detailed in App. C.6. For each product, we generate 100 reactant set, we deduplicate them and rank the unique reactants from most-to-least likely, as judged by the model, using the likelihood lower bound and duplicate counts as a proxy. Based on the types of alignment discussed in Sec. 3.3, we experiment with *(1)* a permutation equivariant model (*Unaligned*), *(2)* a model augmented with atom-mapped positional encodings, calculated from the graph Laplacian eigendecomposition (*DiffAlign-PE*), *(3)* A model with Laplacian positional encodings as well as the skip connection (*DiffAlign-PE+skip*), *(4)* A model where $\mathbf{X}$ and $\mathbf{Y}$ are aligned at the input as well as in the output (*DiffAlign-input alignment*). We use $T = 100$. We trained the models for 400–600 epochs and chose the best checkpoint based on the Mean Reciprocal Rank (MRR, (Liu et al., 2009)) score with $T = 10$ of the validation set. For more details on our experimental setup, see App. C.6.

**Evaluation** We follow previous practices and use the accuracy of obtaining the ground-truth reactants as our main metric. This is measured by top-$k$ accuracy, which counts the number of ground-truth matches in the deduplicated and ranked samples among the top-$k$ generated reactions. We also report Mean Reciprocal Rank (MRR, (Liu et al., 2009)) as used by Maziarz et al. (2023). We focus here on a comparison between aligned and equivariant denoisers. In order to place our models in the context of the available literature, we also compare to existing baselines in App. D.

**Results** We visualize the aligned denoiser and the permutation equivariant denoiser in Fig. 6a. As predicted by Theorem 1, a sample from the permutation equivariant denoising distribution at high levels of noise $p_\theta(\mathbf{X}_0|\mathbf{X}_T, \mathbf{Y})$ has no information about the structure of the product. This is a direct consequence of the symmetry limitation: without alignment, the equivariant denoiser cannot distinguish between different possible node mappings, producing identical outputs for all nodes and effectively averaging across all possible configurations. In contrast, an aligned denoiser is able to copy the product structure, and the initial denoising output is of much higher quality.

Table 1: Top-$k$ accuracy and MRR on the USPTO-50k test data set. Aligned models outperform the unaligned one with the combined PE+skip model reaching the highest results. For a comparison to other retrosynthetic baselines see App. D.

| Method | k = 1 ↑ | k = 3 ↑ | k = 5 ↑ | k = 10 ↑ | $\widehat{\text{MRR}}$ ↑ |
|---|---|---|---|---|---|
| Unaligned | 4.1 | 6.5 | 7.8 | 9.8 | 0.056 |
| DiffAlign-input | 44.1 | 65.9 | 72.2 | 78.7 | 0.554 |
| DiffAlign-PE | 49.0 | 70.7 | 76.6 | **81.8** | 0.601 |
| DiffAlign-PE+skip | **54.7** | **73.3** | **77.8** | 81.1 | **0.639** |

We report results for the different aligned models and the unaligned model for top-$k$ and the combined MRR score on the USPTO-50k test set in Table 1. The model without alignment performs worst, while combining the input alignment with positional encodings achieves the highest top-$k < 10$ and MRR scores across all models, making our diffusion-based model competitive with SOTA in retrosynthesis. Specifically, Table 2 shows our top-$k$ accuracy and MRR score compared to other baselines. We also outperform template-based models in all top-$k$ scores (see note on the evaluation of (Igashov et al., 2024) to understand why its results are not comparable), and outperform all non-pretrained models on top-1. While we use a large value for $T$ during training in our best models, we also highlight that the performance of the aligned model does not degrade significantly when reducing the count of sampling steps to a fraction of $T = 100$. See Fig. 6 for an ablation study

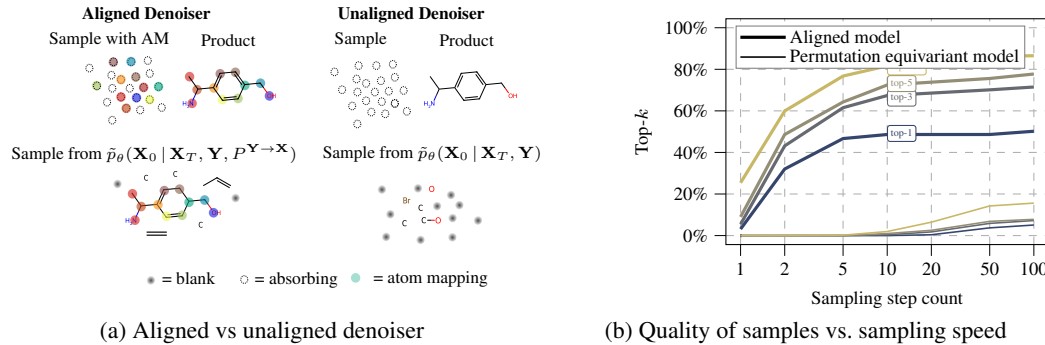

(a) Aligned vs unaligned denoiser

(b) Quality of samples vs. sampling speed

Figure 6: Top-$k$ scores for sampling step counts $T$ for our PE-skip model and a model with a standard permutation equivariant denoiser, using the first 10% of validation set reactions.

with the number of sampling steps for our best model with positional encodings and skip connections. We also compare to a permutation equivariant model and show that the top-$k$ scores go to near zero at 10 steps, while the aligned model sometimes recovers the ground truth even with a single denoising step. We provide further ablations for different transition matrices, a model using only the skip connections, and a model with matched Gaussian noise positional encodings in App. E. With the latter experiment, we show that the benefits brought by the positional encoding having the graph inductive bias due to the graph Laplacian eigenvector structure are not particularly significant, compared to the inductive bias of alignment brought by matching the positional encodings across sides.

Table 2: An extended comparison with top-$k$ accuracy and MRR on the USPTO-50k test data set. We include models with pretraining on larger data sets, and Retrobridge (Igashov et al., 2024), a model whose evaluation is done with a relaxed metric that does not consider charges or stereochemistry.

| | Method | k = 1 ↑ | k = 3 ↑ | k = 5 ↑ | k = 10 ↑ | $\widehat{\text{MRR}}$ ↑ |
|---|---|---|---|---|---|---|
| Pre-trained | RSMILES (Zhong et al., 2022) | 56.3 | **79.2** | **86.2** | **91.0** | 0.680 |
| | PMSR (Jiang et al., 2023) | **62.0** | 78.4 | 82.9 | 86.8 | **0.704** |
| Temp. | Retrosym (Coley et al., 2017b) | 37.3 | 54.7 | 63.3 | 74.1 | 0.480 |
| | GLN (Dai et al., 2019) | 52.5 | 74.7 | 81.2 | 87.9 | 0.641 |
| | LocalRetro (Chen & Jung, 2021) | **52.6** | **76.0** | **84.4** | **90.6** | **0.650** |
| Synthon | GraphRetro (Somnath et al., 2021) | **53.7** | 68.3 | 72.2 | 75.5 | 0.611 |
| | RetroDiff (Wang et al., 2023) | 52.6 | **71.2** | **81.0** | 83.3 | **0.629** |
| | MEGAN (Sacha et al., 2021) | 48.0 | 70.9 | 78.1 | **85.4** | 0.601 |
| | G2G (Shi et al., 2020a) | 48.9 | 67.6 | 72.5 | 75.5 | 0.582 |
| Template-free | SCROP (Zheng et al., 2019) | 43.7 | 60.0 | 65.2 | 68.7 | 0.521 |
| | Tied Transformer (Kim et al., 2021) | 47.1 | 67.1 | 73.1 | 76.3 | 0.572 |
| | Aug. Transformer (Tetko et al., 2020) | 48.3 | - | 73.4 | 77.4 | 0.569 |
| | Retrobridge (*) (Igashov et al., 2024) | 50.3 | **74.0** | **80.3** | **85.1** | 0.622 |
| | GTA_aug (Seo et al., 2021) | 51.1 | 67.6 | 74.8 | 81.6 | 0.605 |
| | Graph2SMILES (Tu & Coley, 2022) | 52.9 | 66.5 | 70.0 | 72.9 | 0.597 |
| | Retroformer (Wan et al., 2022) | 53.2 | 71.1 | 76.6 | 82.1 | 0.626 |
| | DualTF_aug (Sun et al., 2021) | 53.6 | 70.7 | 74.6 | 77.0 | 0.619 |
| Ours | Unaligned | 4.1 | 6.5 | 7.8 | 9.8 | 0.056 |
| | DiffAlign-input | 44.1 | 65.9 | 72.2 | 78.7 | 0.554 |
| | DiffAlign-PE | 49.0 | 70.7 | 76.6 | 81.8 | 0.601 |
| | DiffAlign-PE+skip | **54.7** | 73.3 | 77.8 | 81.1 | **0.639** |

### 4.3 BENEFITS OF DIFFUSION: GUIDED GENERATION AND INPAINTING

We study the use of an external function for guided generation and inpainting, thus demonstrating the advantages of graph-to-graph diffusion in the field of retrosynthesis as a concrete application domain. In retrosynthesis, an interesting use-case for posthoc-conditioning is to increase the probability of the generated reactants being synthesisable, using some pre-trained synthesisability model. To showcase the idea, we use a toy synthesisability model based on the total count of atoms in

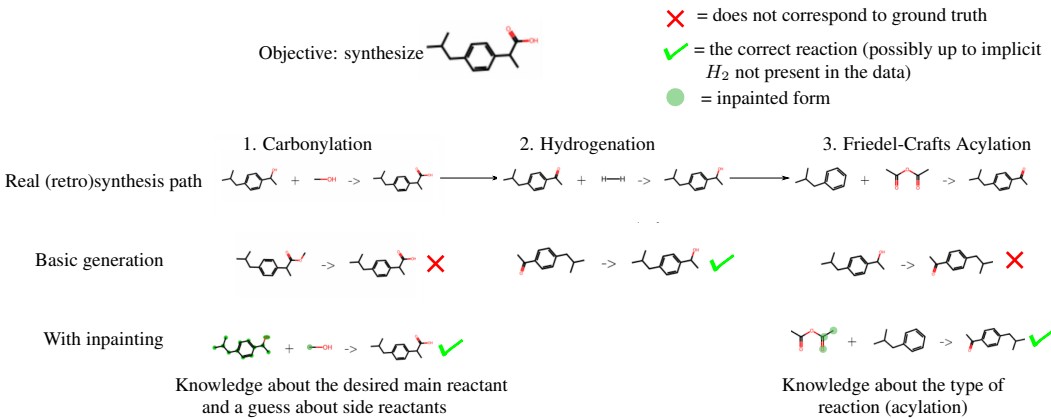

Figure 7: Replicating BHC's green synthesis of Ibuprofen using our model interactively, and comparing to the known synthesis path.

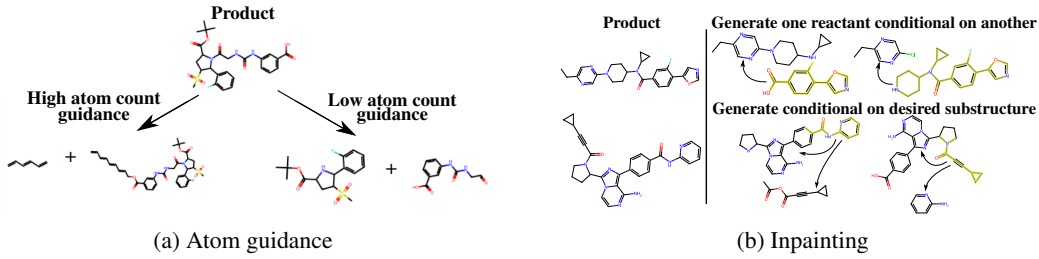

(a) Atom guidance
(b) Inpainting

Figure 8: (a) Atom count guidance lets us specify if reactants should have many or few atoms, controlling the *atom economy*. (b) Examples of inpainting with our model. Parts highlighted in yellow are fixed by a practitioner to reflect desired characteristics, and the diffusion model completes the reaction.

the reactants (Ertl & Schuffenhauer, 2009).Figure 8a shows an example where we nudge the model towards precursors with lower atom count with details of the procedure given in Alg. 3. More detailed comments on this procedure are available in App. F. To illustrate the benefits of inpainting, we present a hypothetical scenario of finding a known synthesis pathway, in particular, BHC's green synthesis of Ibuprofen (Cann & Connelly, 2000). Fig. 7 compares the output of our model to the ground truth synthesis. We write out the steps in detail in App. H.

## 5 CONCLUSION

In this work, we study an important aspect of conditional graph diffusion models: the equivariance of the denoiser. We show that a permutation equivariant model converges to a 'mean' distribution for all graph components—a fundamental limitation stemming from symmetry in the diffusion process. We propose aligned permutation equivariance as a solution, forcing the model to only consider permutations that maintain alignment between conditioning and generated graphs. Our aligned denoisers achieve state-of-the-art results among template-free methods, reaching a top-1 accuracy beyond template-based methods. This approach unlocks the benefits of graph diffusion in retrosynthesis, including flexible post-training conditioning and adjustable sampling steps during inference—valuable properties for interactive applications and multi-step retrosynthesis planners.

A limitation is the requirement of some mapping information between conditional and generated graphs, though not fully mapped graphs. The model handles mapping errors with additional denoising steps. Another limitation is the high computational demand of iterative denoising. Advances in accelerated diffusion sampling methods (Hoogeboom et al., 2022a; Karras et al., 2022; Lu et al., 2022; Song et al., 2023b; Sauer et al., 2023; Shih et al., 2023) are likely to improve this aspect.

ACKNOWLEDGMENTS

We acknowledge funding from the Research Council of Finland (grants 339730, 362408, 334600, 342077) and the Jane and Aatos Erkko Foundation (grant 7001703). We acknowledge CSC – IT Center for Science, Finland, for awarding this project access to the LUMI supercomputer, owned by the EuroHPC Joint Undertaking, hosted by CSC (Finland) and the LUMI consortium through CSC. We acknowledge the computational resources provided by the Aalto Science-IT project.

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

APPENDICES

This appendix is organized as follows.

App. A presents our theoretical results on aligned permutation equivariance and accompanying proofs. App. B provides additional details on the setup for conditional graph diffusion, including the transition matrices, noise schedule, and data encoding as graphs. App. C includes additional details to replicate our experimental setup. App. D provides detailed comparsion between our model and other retrosynthetic baselines. App. F develops a method to apply arbitrary post-training conditioning with discrete diffusion models, and presents case studies showcasing the usefulness of post-training conditional inference in applications relevant to retrosynthesis. This includes generating samples with desired properties and refining the generation interactively through inpainting.

## H  DETAILS FOR THE IBUPROFEN SYNTHESIS EXPERIMENT 37

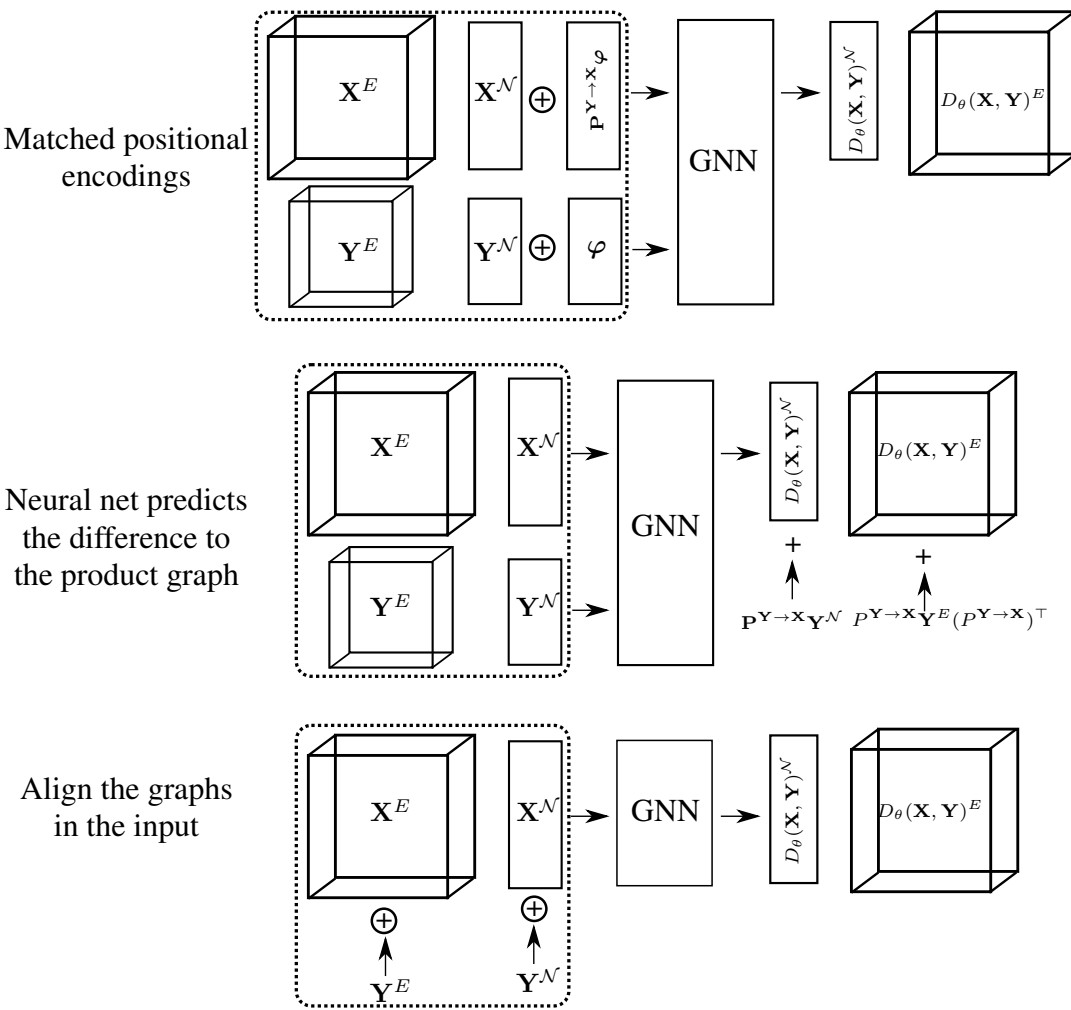

Figure A9: Different ways to align the graphs in the architecture. All of them can be combined. The $\oplus$ sign means concatenation along the feature dimension, and $+$ is the standard addition. All of the methods can be combined together.

## A  ALIGNED PERMUTATION EQUIVARIANCE

### A.1  VISUALIZING OUR ALIGNMENT METHODS

We visualize the various alignment methods we use in this work in Fig. A9.

### A.2  PROOF THAT PERMUTATION EQUIVARIANT DENOISERS DO NOT RECOVER THE IDENTITY DATA

**Definitions**    Let us consider a data set $\mathcal{D} = \{\mathbf{X}_n, \mathbf{Y}_n, \mathbf{P}_n^{\mathbf{Y}\to\mathbf{X}}\}_{n=1}^{N_{obs}}$, where for all data points, $\mathbf{X}_n = \mathbf{P}_n^{\mathbf{Y}\to\mathbf{X}}\mathbf{Y}_n$, that is, both sides of the reactions are equivalent, up to some permutation, as defined in the atom mapping matrix $\mathbf{P}_n^{\mathbf{Y}\to\mathbf{X}}$. It is always possible to preprocess the data such that the rows of $\mathbf{Y}_n$ are permuted with $\mathbf{Y}_n \leftarrow \mathbf{P}_n^{\mathbf{Y}\to\mathbf{X}}\mathbf{Y}_n$ so that the resulting atom mapping between $\mathbf{Y}_n$ and $\mathbf{X}_n$ is always identity. For simplicity, we assume such a preprocessed data set in this section.

Let us assume that the one-step denoiser probability, $p_\theta(\mathbf{X}_0 \mid \mathbf{X}_T, \mathbf{Y})$, is parameterized by the neural network $D_\theta(\mathbf{X}_T, \mathbf{Y}) \in \mathbb{R}^{N \times K}$ such that the probability factorises for the individual nodes and edges (so there is one output in the network for each node and each edge): $p_\theta(\mathbf{X}_0 \mid \mathbf{X}_T, \mathbf{Y}) = \prod_i \sum_k \mathbf{X}_{0,i,k}^N D_\theta(\mathbf{X}_T, \mathbf{Y})_{i,k}^N \prod_{i,j} \sum_k \mathbf{X}_{0,i,j,k}^E D_\theta(\mathbf{X}_T, \mathbf{Y})_{i,j,k}^E$.

**The ideal denoiser** The correct one-step denoiser $D_\theta(\mathbf{X}_T, \mathbf{Y}) = \mathbf{Y}$. This can be shown with the Bayes' rule by $q(\mathbf{X}_0 \mid \mathbf{X}_T, \mathbf{Y}) = \frac{q(\mathbf{X}_T \mid \mathbf{X}_0, \mathbf{Y}) q(\mathbf{X}_0 \mid \mathbf{Y})}{q(\mathbf{X}_T \mid \mathbf{Y})} = \frac{q(\mathbf{X}_T) q(\mathbf{X}_0 \mid \mathbf{Y})}{q(\mathbf{X}_T)} = q(\mathbf{X}_0 \mid \mathbf{Y})$. Because one $\mathbf{X}$ always matches with exactly one $\mathbf{Y}$ in the data, this is a delta distribution $q(\mathbf{X}_0 \mid \mathbf{Y}) = \prod_i \delta_{x_{0,i}, y_i}$, where we define $x_{0,i}$ and $y_i$ as the value of the $i$:th node / edge. It is also easy to see that $D_\theta(\mathbf{X}_T, \mathbf{Y}) = \mathbf{Y}$ is the optimal solution for the cross-entropy loss:

$$- \sum_{(\mathbf{X}_0, \mathbf{Y})} q(\mathbf{X}_0) \log p_\theta(\mathbf{X}_0 \mid \mathbf{X}_T, \mathbf{Y}) \propto - \sum_{\mathbf{Y}} \log p_\theta(\mathbf{Y} \mid \mathbf{X}_T, \mathbf{Y})$$

$$= - \sum_{\mathbf{Y}} \log \left[ \prod_i \sum_k \mathbf{Y}_{0,i,k}^N D_\theta(\mathbf{X}_T, \mathbf{Y})_{i,k}^{\mathcal{N}} \prod_{i,j} \sum_k \mathbf{Y}_{0,i,j,k}^E D_\theta(\mathbf{X}_T, \mathbf{Y})_{i,j,k}^E \right]. \tag{13}$$

All of the sums $\sum_k \mathbf{Y}_{0,i,k}^N D_\theta(\mathbf{X}_T, \mathbf{Y})_{i,k}^{\mathcal{N}}$ and $\sum_k \mathbf{Y}_{0,i,j,k}^E D_\theta(\mathbf{X}_T, \mathbf{Y})_{i,j,k}^E$ are maximized for each $i, j$ and $\mathbf{Y}$ if $D_\theta(\mathbf{X}_T, \mathbf{Y}) = \mathbf{Y}$. In this case, the loss goes to zero.

The following theorem states that if the neural net is permutation equivariant, it will converge to a 'mean' solution, where the output for each node and each edge is the marginal distribution of nodes and edges in the conditioning product molecule, instead of the global optimum $D_\theta(\mathbf{X}_T, \mathbf{Y}) = \mathbf{Y}$.

**Theorem 3.** *The optimal permutation equivariant denoiser Let $D_\theta(\mathbf{X}_T, \mathbf{Y})$ be permutation equivariant s.t. $D_\theta(\mathbf{P}\mathbf{X}_T, \mathbf{Y}) = \mathbf{P} D_\theta(\mathbf{X}_T, \mathbf{Y})$, and let $q(\mathbf{X}_T)$ be permutation invariant. The optimal solution with respect to the cross-entropy loss with the identity data is, for all nodes $i$ and $j$*

$$\begin{cases} D_\theta(\mathbf{X}_T, \mathbf{Y})_{i,:}^{\mathcal{N}} = \hat{\mathbf{y}}^{\mathcal{N}}, \hat{\mathbf{y}}_k^{\mathcal{N}} = \sum_i \mathbf{Y}_{i,k} / \sum_{i,k} \mathbf{Y}_{i,k}, \\ D_\theta(\mathbf{X}_T, \mathbf{Y})_{i,j,:}^E = \hat{\mathbf{y}}^E, \hat{\mathbf{y}}_k^E = \sum_{i,j} \mathbf{Y}_{i,j,k} / \sum_{i,j,k} \mathbf{Y}_{i,j,k}, \end{cases} \tag{14}$$

*where $\hat{\mathbf{y}}_k^{\mathcal{N}}$ and $\hat{\mathbf{y}}_k^E$ are the marginal distributions of node and edge values in $\mathbf{Y}$.*

*Proof.* **Nodes.** The cross-entropy denoising loss for the nodes can be written as

$$CE = - \sum_{(\mathbf{X}_0, \mathbf{Y})} \mathbb{E}_{q(\mathbf{X}_T \mid \mathbf{X}_0)} \sum_{i,k} \mathbf{X}_{0,i,k}^{\mathcal{N}} \log D_\theta(\mathbf{X}_T, \mathbf{Y})_{i,k} \tag{15}$$

$$= - \sum_{(\mathbf{X}_0, \mathbf{Y})} \mathbb{E}_{q(\mathbf{X}_T)} \sum_{i,k} \mathbf{X}_{0,i,k}^{\mathcal{N}} \log D_\theta(\mathbf{X}_T, \mathbf{Y})_{i,k} \tag{16}$$

$$= - \sum_{\mathbf{Y}} \mathbb{E}_{q(\mathbf{X}_T)} \sum_{i,k} \mathbf{Y}_{i,k}^{\mathcal{N}} \log D_\theta(\mathbf{X}_T, \mathbf{Y})_{i,k}, \tag{17}$$

where the first equality is due to $q(\mathbf{X}_T \mid \mathbf{X}_0)$ containing no information about $\mathbf{X}_0$ at the end of the forward process, and the second equality is due to $\mathbf{X}_0 = \mathbf{Y}$ in the data. Since $q(\mathbf{X}_T)$ is permutation invariant, that is, all permuted versions $\mathbf{P}\mathbf{X}_T$ of $\mathbf{X}_T$ are equally probable, we can split the expectation into two parts $\mathbb{E}_{q(\mathbf{X}_T)}[\cdot] \propto \mathbb{E}_{q(\mathbf{X}_T')} \sum_{\mathbf{P}}[\cdot]$, where $\mathbf{X}_T'$ contain only graphs in distinct isomorphism classes, and $\sum_{\mathbf{P}}$ sums over all permutation matrices of size $N \times N$,

$$CE \propto - \sum_{\mathbf{Y}} \mathbb{E}_{q(\mathbf{X}_T')} \sum_{\mathbf{P}} \sum_{i,k} \mathbf{Y}_{i,k}^{\mathcal{N}} \log D_\theta(\mathbf{P}\mathbf{X}_T', \mathbf{Y})_{i,k}^{\mathcal{N}}. \tag{18}$$

Due to the permutation equivariance, $D_\theta(\mathbf{P}\mathbf{X}_T', \mathbf{Y})^{\mathcal{N}} = \mathbf{P} D_\theta(\mathbf{X}_T', \mathbf{Y})^{\mathcal{N}}$, and $D_\theta(\mathbf{P}\mathbf{X}_T', \mathbf{Y})_{i,k}^{\mathcal{N}} = D_\theta(\mathbf{X}_T', \mathbf{Y})_{\pi(i),k}^{\mathcal{N}}$, where $\pi(i)$ denotes the index the index $i$ is mapped to in the permutation $\mathbf{P}$. Thus,

$$CE \propto - \sum_{\mathbf{Y}} \mathbb{E}_{q(\mathbf{X}_T')} \sum_\pi \sum_{i,k} \mathbf{Y}_{i,k}^{\mathcal{N}} \log D_\theta(\mathbf{X}_T', \mathbf{Y})_{\pi(i),k}^{\mathcal{N}} \tag{19}$$

$$= - \sum_{\mathbf{Y}} \mathbb{E}_{q(\mathbf{X}_T')} \sum_{i,k} \sum_\pi \mathbf{Y}_{\pi^{-1}(i),k}^{\mathcal{N}} \log D_\theta(\mathbf{X}_T', \mathbf{Y})_{i,k}^{\mathcal{N}}, \tag{20}$$

where the equality is due to all permutations being in a symmetric position: What matters is the relative permutation between $\mathbf{Y}^{\mathcal{N}}$ and $D_\theta(\mathbf{X}_T', \mathbf{Y})$. Now, $\sum_\pi \mathbf{Y}_{\pi^{-1}(i),k}^{\mathcal{N}} = \sum_\pi \mathbf{Y}_{\pi(i),k}^{\mathcal{N}} = C \sum_i \mathbf{Y}_{i,k}^{\mathcal{N}}$,

because for each node index $i$, all the nodes in $\mathbf{Y}$ are included equally often due to symmetry. This is proportional to the marginal distribution $\hat{\mathbf{y}}^{\mathcal{N}}$ up to some constant, and thus we have:

$$CE \propto -\sum_{\mathbf{Y}} \mathbb{E}_{q(\mathbf{X}'_T)} \sum_{i,k} \hat{\mathbf{y}}^{\mathcal{N}}_k \log D_\theta(\mathbf{X}'_T, \mathbf{Y})_{i,k}. \tag{21}$$

The optimal value for each node output $i$ is the empirical marginal distribution $D_\theta(\mathbf{X}'_T, \mathbf{Y})^N_{i,:} = (\hat{\mathbf{y}}^{\mathcal{N}})^\top$.

**Edges** With the exact same steps, we can get the equivalent of Eq. (18) for the edges:

$$CE \propto -\sum_{\mathbf{Y}} \mathbb{E}_{q(\mathbf{X}'_T)} \sum_{P} \sum_{i,j,k} \mathbf{Y}^E_{i,j,k} \log D_\theta(P\mathbf{X}'_T, \mathbf{Y})^E_{i,j,k}. \tag{22}$$

The permutation equivariance property for the edges is now written as $D_\theta(\mathbf{PX}'_T, \mathbf{Y})^E = \mathbf{P} D_\theta(\mathbf{X}'_T, \mathbf{Y})^E \mathbf{P}^\top$, and $D_\theta(\mathbf{PX}'_T, \mathbf{Y})^{\mathcal{N}}_{i,jk} = D_\theta(\mathbf{X}'_T, \mathbf{Y})^{\mathcal{N}}_{\pi(i),\pi(j),k}$. Thus,

$$CE \propto -\sum_{\mathbf{Y}} \mathbb{E}_{q(\mathbf{X}'_T)} \sum_{\pi} \sum_{i,j,k} \mathbf{Y}^E_{i,j,k} \log D_\theta(\mathbf{X}'_T, \mathbf{Y})^E_{\pi(i),\pi(j),k} \tag{23}$$

$$= -\sum_{\mathbf{Y}} \mathbb{E}_{q(\mathbf{X}'_T)} \sum_{i,j,k} \sum_{\pi} \mathbf{Y}^E_{\pi^{-1}(i),\pi^{-1}(j),k} \log D_\theta(\mathbf{X}'_T, \mathbf{Y})^E_{i,j,k}, \tag{24}$$

with the equality holding again due to symmetry. Now, for any pair of node indices $i$ and $j$, the set of all permutations contains all pairs of node indices $(\pi(i), \pi(j))$ equally often due to symmetry. These pairs correspond to edges in $\mathbf{Y}^E$, and thus $\sum_\pi \mathbf{Y}^E_{\pi^{-1}(i),\pi^{-1}(j),k} = \sum_\pi \mathbf{Y}^E_{\pi(i),\pi(j),k} = D\sum_{i,j} \mathbf{Y}^E_{i,j,k}$, where $D$ is a constant that counts how many times each edge pair appeared in the set of all permutations. This is again proportional to the marginal distribution over the edges $\hat{\mathbf{y}}^E$

$$CE \propto -\sum_{\mathbf{Y}} \mathbb{E}_{q(\mathbf{X}'_T)} \sum_{i,j,k} \hat{\mathbf{y}}^E_k \log D_\theta(\mathbf{X}'_T, \mathbf{Y})_{i,j,k}. \tag{25}$$

Again, the optimal value for each edge output $(i,j)$ is $D_\theta(\mathbf{X}'_T, \mathbf{Y})^E_{i,j,:} = (\hat{\mathbf{y}}^E)^\top$. $\qquad\qquad\square$

### A.3 PROOF THAT ALIGNED PERMUTATION EQUIVARIANT DENOISERS RECOVER THE IDENTITY DATA

Here, we detail the steps in the proof of Theorem 1, but with aligned permutation eqivariance intead of regular permutation equivariance, and highlight the principal differences in both calculations. We start with versions of Eq. (17) rewritten with atom mapping explicitly, and $P^{Y\to X}$ conditioning in the denoiser:

$$CE = -\sum_{(X_0,Y,P^{Y\to X})} \mathbb{E}_{q(X_T \mid X_0)} \sum_{i,k} X^N_{0,i,k} \log D_\theta(X_T, Y, P^{Y\to X})_{i,k} \tag{26}$$

$$= -\sum_{(Y,P^{Y\to X})} \mathbb{E}_{q(X_T)} \sum_{i,k} (P^{Y\to X}Y^N)_{i,k} \log D_\theta(X_T, Y, P^{Y\to X})_{i,k} \tag{27}$$

Versions of Eq. (18) to Eq. (20) with atom mapping conditioning and utilizing aligned permutation equivariance (instead of permutation equivariance):

$$CE \propto -\sum_{Y,P^{Y\to X}} \mathbb{E}_{q(X'_T)} \sum_{P} \sum_{i,k} (P^{Y\to X}Y^N)_{i,k} \log D_\theta(PX'_T, Y, P^{Y\to X})^N_{i,k} \tag{28}$$

$$= -\sum_{Y,P^{Y\to X}} \mathbb{E}_{q(X'_T)} \sum_{P} \sum_{i,k} (P^{Y\to X}Y^N)_{i,k} \log D_\theta(X'_T, Y, P^{-1}P^{Y\to X})^N_{\pi(i),k} \tag{29}$$

$$= -\sum_{Y,P^{Y\to X}} \mathbb{E}_{q(X'_T)} \sum_{i,k} \sum_{\pi} (P^{Y\to X}Y^N)_{\pi^{-1}(i),k} \log D_\theta(X'_T, Y, P^{-1}P^{Y\to X})^N_{i,k} \tag{30}$$

$$= -\sum_{Y,P^{Y\to X}} \mathbb{E}_{q(X'_T)} \sum_{i,k} \sum_{P} (P^{-1}P^{Y\to X}Y^N)_{i,k} \log D_\theta(X'_T, Y, P^{-1}P^{Y\to X})^N_{i,k} \tag{31}$$

We could assume again for notational simplicity (not necessary) that $P^{Y \to X} = I$ in the data. In any case, we have a cross-entropy loss where we try to predict a permutation of $Y$, $P^{-1} P^{Y \to X} Y$. Both $P^{-1} P^{Y \to X}$ and $Y$ are given as explicit information to the neural network, so for each $P$, the optimal output is $P^{-1} P^{Y \to X} Y$, which is a permutation of the original $Y$.

The primary difference in these two proofs:

- With the version with regular permutation equivariance, only the data term $Y$ is dependent on the permutation $P$ in Eq. (20). This makes it possible to factor out the sum $\sum_\pi Y^N_{\pi^{-1}(i),k}$, leading to Eq. (21). This factorization is also why the optimal neural net output is not dependent on $P$.

- With aligned permutation equivariance, the denoiser term retains dependence on $P^{-1}$, and the maximal factorization is $\sum_P (P^{-1} Y^N)_{i,k} \log D_\theta(X'_T, Y, P^{-1})^N_{i,k}$. Thus, for each combination of $P$ and $Y$, the neural net will have a different optimal output.

### A.4 Proof of the Generalized Distributional Invariance with Aligned Equivariance

We start by proving a useful lemma, and then continue and continue to the proof of the main theorem.

**Lemma 1.** *(An aligned denoiser induces aligned distribution equivariance for a single reverse step)*

*If the denoiser function $D_\theta$ has the aligned equivariance property $D_\theta(\mathbf{R X}, \mathbf{Q Y}, \mathbf{R P^{Y \to X} Q}^\top) = \mathbf{R} D_\theta(\mathbf{X}, \mathbf{Y}, \mathbf{P^{Y \to X}})$, then the conditional reverse distribution $p_\theta(\mathbf{X}_{t-1} \mid \mathbf{X}_t, \mathbf{Y}, \mathbf{P^{Y \to X}})$ has the property $p_\theta(\mathbf{R X}_{t-1} \mid \mathbf{R X}_t, \mathbf{Q Y}, \mathbf{R P^{Y \to X} Q}^\top) = p_\theta(\mathbf{X}_{t-1} \mid \mathbf{X}_t, \mathbf{Y}, \mathbf{P^{Y \to X}})$.*

*Proof.* First, let us denote the transition probabilities from $t$ to $t-1$ with $F_\theta(\mathbf{X}_t, \mathbf{Y}, \mathbf{P^{Y \to X}})$, where formally $F_\theta(\mathbf{X}_t, \mathbf{Y}, \mathbf{P^{Y \to X}})^{\mathcal{N}}_{i,k} = \sum_{k'} q(\mathbf{X}^{\mathcal{N}}_{t-1,i,k} \mid \mathbf{X}^{\mathcal{N}}_{t,i,k}, \mathbf{X}^{\mathcal{N}}_{0,i,k}) D_\theta(\mathbf{X}_t, \mathbf{Y}, \mathbf{P^{Y \to X}})^{\mathcal{N}}_{i,k'}$ and $F_\theta(\mathbf{X}_t, \mathbf{Y}, \mathbf{P^{Y \to X}})^E_{i,j,k} = \sum_{k'} q(\mathbf{X}^E_{t-1,i,j,k} \mid \mathbf{X}^E_{t,i,j,k}, \mathbf{X}^E_{0,i,j,k}) D_\theta(\mathbf{X}_t, \mathbf{Y}, \mathbf{P^{Y \to X}})^E_{i,j,k'}$. Since the values of $F_\theta$ depend only pointwise on the values of $D_\theta$, $F_\theta$ is aligned permutation equivariant as well.

We continue by directly deriving the connection:

$$p_\theta(\mathbf{R X}_{t-1} \mid \mathbf{R X}_t, \mathbf{Q Y}, \mathbf{R P^{Y \to X} Q}^\top) \tag{32}$$

$$= \prod_i \sum_k (\mathbf{R X}_{t-1})^{\mathcal{N}}_{i,k} F_\theta(\mathbf{R X}_t, \mathbf{Q Y}, \mathbf{R P^{Y \to X} Q}^\top)_{i,k}$$
$$\times \prod_{i,j} \sum_k (\mathbf{R X}_{t-1})^E_{i,j,k} F_\theta(\mathbf{R X}_t, \mathbf{Q Y}, \mathbf{R P^{Y \to X} Q}^\top)^E_{i,j,k} \tag{33}$$

$$= \prod_i \sum_k (\mathbf{X}_{t-1})^{\mathcal{N}}_{\pi(i),k} F_\theta(\mathbf{X}_t, \mathbf{Y}, \mathbf{P^{Y \to X}})_{\pi(i),k}$$
$$\times \prod_{i,j} \sum_k (\mathbf{X}_{t-1})^E_{\pi(i),\pi(j),k} F_\theta(\mathbf{X}_t, \mathbf{Y}, \mathbf{P^{Y \to X}})^E_{\pi(i),\pi(j),k}, \tag{34}$$

where in the last line we used the aligned permutation equivariance definition, and the the effect of the permutation matrix $\mathbf{R}$ on index $i$ was denoted as $\pi(i)$. Now, regardless of the permutation, the products contain all possible values $i$ and pairs $i, j$ exactly once. Thus, the expression remains equal if we replace $\pi(i)$ with just $i$:

$$p_\theta(\mathbf{R X}_{t-1} \mid \mathbf{R X}_t, \mathbf{Q Y}, \mathbf{R P^{Y \to X} Q}^\top) \tag{35}$$

$$= \prod_i \sum_k (\mathbf{X}_{t-1})^{\mathcal{N}}_{i,k} F_\theta(\mathbf{X}_t, \mathbf{Y}, \mathbf{P^{Y \to X}})_{i,k} \prod_{i,j} \sum_k (\mathbf{X}_{t-1})^E_{i,j,k} F_\theta(\mathbf{X}_t, \mathbf{Y}, \mathbf{P^{Y \to X}})^E_{i,j,k} \tag{36}$$

$$= p_\theta(\mathbf{X}_{t-1} \mid \mathbf{X}_t, \mathbf{Y}, \mathbf{P^{Y \to X}}), \tag{37}$$

which concludes the proof. $\square$

**Theorem 4.** *Aligned denoisers induce aligned permutation invariant distributions If the denoiser function $D_\theta$ has the aligned equivariance property and the prior $p(\mathbf{X}_T)$ is permutation invariant,*

*then the generative distribution $p_\theta(\mathbf{X}_0 \mid \mathbf{Y}, \mathbf{P^{Y \to X}})$ has the corresponding property for any permutation matrices $\mathbf{R}$ and $\mathbf{Q}$:*

$$p_\theta(\mathbf{R}\mathbf{X}_0 \mid \mathbf{Q}\mathbf{Y}, \mathbf{R}\mathbf{P^{Y \to X}}\mathbf{Q}^\top) = p_\theta(\mathbf{X}_0 \mid \mathbf{Y}, \mathbf{P^{Y \to X}}) \tag{38}$$

*Proof.* Let us assume that the result holds for some noisy data level $t$: $p_\theta(\mathbf{R}\mathbf{X}_t \mid \mathbf{Q}\mathbf{Y}, \mathbf{R}\mathbf{P^{Y \to X}}\mathbf{Q}^\top) = p_\theta(\mathbf{X}_t \mid \mathbf{Y}, \mathbf{P^{Y \to X}})$. We will then show that the same will hold for $\mathbf{X}_{t-1}$, which we can use to inductively show that the property holds for $\mathbf{X}_0$. We begin as follows:

$$p_\theta(\mathbf{R}\mathbf{X}_{t-1} \mid \mathbf{Q}\mathbf{Y}, \mathbf{R}\mathbf{P^{Y \to X}}\mathbf{Q}^\top) = \sum_{\mathbf{X}_t} p_\theta(\mathbf{R}\mathbf{X}_{t-1} \mid \mathbf{X}_t, \mathbf{Q}\mathbf{Y}, \mathbf{R}\mathbf{P^{Y \to X}}\mathbf{Q}^\top) p_\theta(\mathbf{X}_t \mid \mathbf{Q}\mathbf{Y}, \mathbf{R}\mathbf{P^{Y \to X}}\mathbf{Q}^\top) \tag{39}$$

$$= \sum_{\mathbf{X}_t} p_\theta(\mathbf{X}_{t-1} \mid \mathbf{R}^{-1}\mathbf{X}_t, \mathbf{Y}, \mathbf{P^{Y \to X}}) p_\theta(\mathbf{R}^{-1}\mathbf{X}_t \mid \mathbf{Y}, \mathbf{P^{Y \to X}}). \tag{40}$$

where on the second line we used Lem. 1 and the assumption that the result holds for noise level $t$. The sum over $\mathbf{X}_t$ contains all possible graphs and all of their permutations. Thus, the exact value of $\mathbf{R}^{-1}$ does not affect the value of the final sum, as we simply go through the same permutations in a different order, and aggregate the permutations with the sum. Thus,

$$p_\theta(\mathbf{R}\mathbf{X}_{t-1} \mid \mathbf{Q}\mathbf{Y}, \mathbf{R}\mathbf{P^{Y \to X}}\mathbf{Q}^\top) = \sum_{\mathbf{X}_t} p_\theta(\mathbf{X}_{t-1} \mid \mathbf{X}_t, \mathbf{Y}, \mathbf{P^{Y \to X}}) p_\theta(\mathbf{X}_t \mid \mathbf{Y}, \mathbf{P^{Y \to X}}) \tag{41}$$

$$= p_\theta(\mathbf{X}_{t-1} \mid \mathbf{Y}, \mathbf{P^{Y \to X}}) \tag{42}$$

showing that if the result holds for level $t$, then it also holds for level $t-1$. We only need to show that it holds for level $\mathbf{X}_{T-1}$ to start the inductive chain:

$$p_\theta(\mathbf{R}\mathbf{X}_{T-1} \mid \mathbf{Q}\mathbf{Y}, \mathbf{R}\mathbf{P^{Y \to X}}\mathbf{Q}^\top) = \sum_{\mathbf{X}_T} p_\theta(\mathbf{R}\mathbf{X}_{T-1} \mid \mathbf{X}_T, \mathbf{Q}\mathbf{Y}, \mathbf{R}\mathbf{P^{Y \to X}}\mathbf{Q}^\top) p(\mathbf{X}_T) \tag{43}$$

$$= \sum_{\mathbf{X}_T} p_\theta(\mathbf{X}_{T-1} \mid \mathbf{R}^{-1}\mathbf{X}_T, \mathbf{Y}, \mathbf{P^{Y \to X}}) p(\mathbf{R}^{-1}\mathbf{X}_T), \tag{44}$$

where on the second line we again used Lem. 1 and the permutation invariance of $p(\mathbf{X}_T)$. Again, the exact value of $\mathbf{R}^{-1}$ does not matter for the sum, since the sum goes through all possible permutations in any case. Thus we have

$$p_\theta(\mathbf{R}\mathbf{X}_{T-1} \mid \mathbf{Q}\mathbf{Y}, \mathbf{R}\mathbf{P^{Y \to X}}\mathbf{Q}^\top) = \sum_{\mathbf{X}_T} p_\theta(\mathbf{X}_{T-1} \mid \mathbf{X}_T, \mathbf{Y}, \mathbf{P^{Y \to X}}) p(\mathbf{X}_T) \tag{45}$$

$$= p_\theta(\mathbf{X}_{T-1} \mid \mathbf{Y}, \mathbf{P^{Y \to X}}). \tag{46}$$

Thus, since the property holds for $\mathbf{X}_{T-1}$, it also holds for $\mathbf{X}_{T-2}, \dots,$ until $\mathbf{X}_0$. This concludes the proof. $\square$

## A.5 Proofs that Our Denoisers Are Aligned Permutation Equivariant

In this section, we show for each of the three alignment methods that the corresponding denoisers do indeed fall within the aligned permutation equivariance function class. Fig. A9 summarizes the different alignment methods.

**Atom-mapped positional encodings**  We start by writing out one side of the aligned permutation equivariance condition, $D_\theta(\mathbf{R}\mathbf{X}_t, \mathbf{Q}\mathbf{Y}, \mathbf{R}\mathbf{P^{Y \to X}}\mathbf{Q}^\top)$ for this particular function class, and directly show that it equals $\mathbf{P}D_\theta(\mathbf{X}_t, \mathbf{Y}, \mathbf{P^{Y \to X}})$.

$$D_\theta(\mathbf{R}\mathbf{X}_t, \mathbf{Q}\mathbf{Y}, \mathbf{R}\mathbf{P^{Y \to X}}\mathbf{Q}^\top) = f_\theta^{\mathbf{X}}\left(\begin{bmatrix} \mathbf{R}\mathbf{X}_t^N & \mathbf{R}\mathbf{P^{Y \to X}}\mathbf{Q}^\top\mathbf{Q}\varphi \end{bmatrix}, \mathbf{R}\mathbf{X}^E\mathbf{R}^\top, \begin{bmatrix} \mathbf{Q}\mathbf{Y}^N & \mathbf{Q}\varphi \end{bmatrix}, \mathbf{Q}\mathbf{Y}^E\mathbf{Q}^\top\right) \tag{47}$$

$$= f_\theta^{\mathbf{X}}\left(\begin{bmatrix} \mathbf{R}\mathbf{X}_t^N & \mathbf{R}\mathbf{P^{Y \to X}}\varphi \end{bmatrix}, \mathbf{R}\mathbf{X}^E\mathbf{R}^\top, \begin{bmatrix} \mathbf{Q}\mathbf{Y}^N & \mathbf{Q}\varphi \end{bmatrix}, \mathbf{Q}\mathbf{Y}^E\mathbf{Q}^\top\right), \tag{48}$$

where $f_\theta$ itself is a function that is permutation equivariant for the combined $\mathbf{X}$ and $\mathbf{Y}$ graph as input. This means that the neural net itself gives an output for the entire combined graph, but we only consider the $\mathbf{X}$ subgraph as the denoiser output, denoted here as $f_\theta^{\mathbf{X}}$. For clarity, we can combine the reactant and product node features and adjacency matrices in the notation, and use the permutation equivariance property of our GNN:

$$f_\theta\left(\begin{bmatrix} \mathbf{R}\mathbf{X}_t^{\mathcal{N}} & \mathbf{R}\mathbf{P}^{\mathbf{Y}\to\mathbf{X}}\varphi \\ \mathbf{Q}\mathbf{Y}^{\mathcal{N}} & \mathbf{Q}\varphi \end{bmatrix}, \begin{bmatrix} \mathbf{R}\mathbf{X}_t^E\mathbf{R}^\top & 0 \\ 0 & \mathbf{Q}\mathbf{Y}^E\mathbf{Q}^\top \end{bmatrix}\right) \tag{49}$$

$$= \begin{bmatrix} \mathbf{R} & 0 \\ 0 & \mathbf{Q} \end{bmatrix} f_\theta\left(\begin{bmatrix} \mathbf{X}_t^{\mathcal{N}} & \mathbf{P}^{\mathbf{Y}\to\mathbf{X}}\varphi \\ \mathbf{Y}^{\mathcal{N}} & \varphi \end{bmatrix}, \begin{bmatrix} \mathbf{X}_t^E & 0 \\ 0 & \mathbf{Y}^E \end{bmatrix}\right) \quad \text{(permutation equivariance of base NN)} \tag{50}$$

Taking only the $\mathbf{X}$ part of the output and reverting to $D_\theta$ notation, we directly arrive at the result that $D_\theta(\mathbf{R}\mathbf{X}_t, \mathbf{Q}\mathbf{Y}, \mathbf{R}\mathbf{P}^{\mathbf{Y}\to\mathbf{X}}\mathbf{Q}^\top) = \mathbf{R}D_\theta(\mathbf{X}_t, \mathbf{Y}, \mathbf{P}^{\mathbf{Y}\to\mathbf{X}})$.

**Directly adding $\mathbf{Y}$ to the output** We again start by writing out an aligned equivariant input to the denoiser, with $f_\theta$ again denoting a network that is permutation equivariant with respect to the combined $\mathbf{X}$ and $\mathbf{Y}$ graph:

$$D_\theta(\mathbf{R}\mathbf{X}, \mathbf{Q}\mathbf{Y}, \mathbf{R}\mathbf{P}^{\mathbf{X}\to\mathbf{Y}}\mathbf{Q}^\top) = \text{softmax}(f_\theta^{\mathbf{X}}(\mathbf{R}\mathbf{X}, \mathbf{Q}\mathbf{Y}) + \mathbf{R}\mathbf{P}^{\mathbf{Y}\to\mathbf{X}}\mathbf{Q}^\top\mathbf{Q}\mathbf{Y}) \tag{51}$$

$$= \text{softmax}(\mathbf{R}f_\theta^{\mathbf{X}}(\mathbf{X}, \mathbf{Y}) + \mathbf{R}\mathbf{P}^{\mathbf{Y}\to\mathbf{X}}\mathbf{Y}) \quad \text{(permutation equivariance of base denoiser)} \tag{52}$$

$$= \mathbf{R}\,\text{softmax}(f_\theta^{\mathbf{X}}(\mathbf{X}, \mathbf{Y}) + \mathbf{P}^{\mathbf{Y}\to\mathbf{X}}\mathbf{Y}) \tag{53}$$

$$= \mathbf{R}D_\theta(\mathbf{X}, \mathbf{Y}, \mathbf{P}^{\mathbf{X}\to\mathbf{Y}}) \tag{54}$$

where we were able to move the permutation outside the softmax since the softmax is applied on each node and edge separately.

**Aligning $\mathbf{Y}$ and $\mathbf{X}$ at the input to the model** Let's denote by $[\mathbf{X}\ \ \mathbf{P}^{\mathbf{Y}\to\mathbf{X}}\mathbf{Y}]$ concatenation along the feature dimension for both the nodes and edges of the graphs. Recall then that the definition of aligning the graphs in the input is $D_\theta(\mathbf{X}, \mathbf{Y}, \mathbf{P}^{\mathbf{Y}\to\mathbf{X}}) = f_\theta([\mathbf{X}\ \ \mathbf{P}^{\mathbf{Y}\to\mathbf{X}}\mathbf{Y}])$, where $f_\theta$ is a permutation equivariant denoiser. Writing out the aligned equivariance condition

$$D_\theta(\mathbf{R}\mathbf{X}, \mathbf{Q}\mathbf{Y}, \mathbf{R}\mathbf{P}^{\mathbf{Y}\to\mathbf{X}}\mathbf{Q}^\top) = f_\theta([\mathbf{R}\mathbf{X}\ \ \mathbf{R}\mathbf{P}^{\mathbf{Y}\to\mathbf{X}}\mathbf{Q}^\top\mathbf{Q}\mathbf{Y}]) = f_\theta([\mathbf{R}\mathbf{X}\ \ \mathbf{R}\mathbf{P}^{\mathbf{Y}\to\mathbf{X}}\mathbf{Y}]) \tag{55}$$

$$= f_\theta(\mathbf{R}[\mathbf{X}\ \ \mathbf{P}^{\mathbf{Y}\to\mathbf{X}}\mathbf{Y}]) = \mathbf{R}f_\theta([\mathbf{X}\ \ \mathbf{P}^{\mathbf{Y}\to\mathbf{X}}\mathbf{Y}]) \tag{56}$$

$$= D_\theta(\mathbf{X}, \mathbf{Y}, \mathbf{P}^{\mathbf{Y}\to\mathbf{X}}) \tag{57}$$

which shows that this method results in aligned equivariance as well.

## A.6 A Single-layer Graph Transformer with Orthogonal Atom-mapped Positional Encodings is Able to Implement the Identity Data Solution for Nodes

Here, we show that a single-layer Graph Transformer neural net can model the identity data for the nodes given orthogonal atom-mapped positional encodings (e.g., the graph Laplacian eigenvector-based ones). In particular it is possible to find a $\theta$ such that $D_\theta(\mathbf{X}_t, \mathbf{Y}, \mathbf{P}^{\mathbf{Y}\to\mathbf{X}})^{\mathcal{N}} = \mathbf{Y}^{\mathcal{N}}$. We hope that this section can serve as an intuitive motivation for why matched positional encodings help in copying over the structure from the product to the reactant side.

Recall that we have $N$ atoms on both sides of the reaction. Let us map the atom mapping indices to basis vectors in an orthogonal basis $\varphi = [\varphi_1, \varphi_2, \ldots, \varphi_N]^\top$. In practice, the node input to the neural net on the reactant side is now $\mathbf{X}_t^* = [\mathbf{X}_t^{\mathcal{N}}, \varphi]$, and the node input on the product side is $\mathbf{Y}^* = [\mathbf{Y}^{\mathcal{N}}, \varphi]$.

Now, a single-layer Graph Transformer looks as follows (as defined in the Vignac et al. (2023) codebase):

$$\mathbf{N} = [(\mathbf{X}_t^{*\mathcal{N}})^\top, (\mathbf{Y}^{*\mathcal{N}})^\top]^\top \quad \text{(Concatenate rows)} \tag{58}$$

$$\mathbf{E} = \begin{bmatrix} \mathbf{X}_t^E & 0 \\ 0 & \mathbf{Y}^E \end{bmatrix} \quad \text{(Create joined graph)} \tag{59}$$

$$\mathbf{N}_1 = MLP_N(\mathbf{N}) \quad \text{(Applied with respect to last dimension)} \tag{60}$$

$$\mathbf{E}_1 = \frac{1}{2}(MLP_E(\mathbf{E}) + MLP_E(\mathbf{E})^\top) \quad \text{(Symmetrize the input)} \tag{61}$$

$$t_1 = MLP_t(t) \tag{62}$$

$$\mathbf{Q}, \mathbf{K}, \mathbf{V} = \mathbf{W}_Q \mathbf{N}_1, \mathbf{W}_K \mathbf{N}_1, \mathbf{W}_V \mathbf{N}_1 \quad \text{(One attention head for simplicity)} \tag{63}$$

$$\mathbf{A}_1 = \mathbf{Q}\mathbf{K}^\top \tag{64}$$

$$\mathbf{A}_2 = \mathbf{A}_1 * (\mathbf{W}_{\mathbf{E}_2}\mathbf{E}_1 + 1) + \mathbf{W}_{\mathbf{E}_3}\mathbf{E}_1 \tag{65}$$

$$\mathbf{A}_3 = \text{softmax}(\mathbf{A}_3) \tag{66}$$

$$\mathbf{N}_2 = \mathbf{A}\mathbf{V}/\sqrt{d_f} \quad (d_f \text{ is the embedding dim of } \mathbf{V}) \tag{67}$$

$$\mathbf{N}_3 = \mathbf{N}_2 * (\mathbf{W}_{t_2}t_1 + 1) + \mathbf{W}_{t_3}t_1 \tag{68}$$

$$\mathbf{E}_2 = \mathbf{W}(\mathbf{A}_2 * (\mathbf{W}_{t_4}t_1 + 1) + W_{t_5}t_1) \tag{69}$$

$$t_4 = MLP_{t_4}(\mathbf{W}t_2 + \mathbf{W}[\min(\mathbf{N}_3), \max(\mathbf{N}_3), \text{mean}(\mathbf{N}_3), \text{std}(\mathbf{N}_3)]$$
$$+ \mathbf{W}[\min(\mathbf{N}_3), \max(\mathbf{N}_3), \text{mean}(\mathbf{N}_3), \text{std}(\mathbf{N}_3)] \tag{70}$$

$$\mathbf{E}_3 = MLP_{\mathbf{E}_2}(\mathbf{E}_2) + \mathbf{E} \tag{71}$$

$$\mathbf{E}_{out} = \frac{1}{2}(\mathbf{E}_3 + \mathbf{E}_3^\top) \quad \text{(Symmetrize the output)} \tag{72}$$

$$\mathbf{N}_{out} = MLP_{N_3}(\mathbf{N}_3) + \mathbf{N} \tag{73}$$

$$t_{out} = t_4 + t \tag{74}$$

Now, for purposes of illustration, we can define most linear layers to be zero layers: $MLP_E, MLP_t, \mathbf{W}_{E_2}, \mathbf{W}_{E_3}, \mathbf{W}_{t_2}, \mathbf{W}_{t_3}, \mathbf{W}_{t_4}, \mathbf{W}_{t_5} = 0$. In addition to this, we define $MLP_N$ to be an identity transform. $\mathbf{W}_Q$ and $\mathbf{W}_K$ are both chosen as picking out the $\mathbf{U}$ columns of $\mathbf{N}_1$, with additional overall scaling by some constant $\alpha$. $\mathbf{W}_V$ is chosen to pick out the product node labels: $\mathbf{W}_V \mathbf{N}_1 = \begin{bmatrix} 0 \\ \mathbf{Y}^{\mathcal{N}} \end{bmatrix}$. Now, we can easily see how the Graph Transformer can obtain the optimal denoising solution for the nodes. Consider an input $\mathbf{N} = [(\mathbf{P}\mathbf{X}_t^*)^\top, (\mathbf{Y})^\top]^\top$, where the reactant side is permuted. The output of the network should be $\mathbf{P}\mathbf{Y}$. Focusing on the parts of the network that compute the node features:

$$\mathbf{A}_1 = \alpha^2 \begin{bmatrix} \mathbf{P}\varphi\varphi^\top\mathbf{P}^\top & \mathbf{P}\varphi\varphi^\top \\ \varphi\varphi^\top\mathbf{P}^\top & \varphi\varphi^\top \end{bmatrix} = \alpha^2 \begin{bmatrix} \mathbf{I} & \mathbf{P} \\ \mathbf{P}^\top & \mathbf{I} \end{bmatrix}, \tag{75}$$

$$\mathbf{A}_2 = \mathbf{A}_1, \tag{76}$$

$$\mathbf{A}_3 \approx \text{softmax}(\mathbf{A}_2) = \frac{1}{2}\begin{bmatrix} \mathbf{I} & \mathbf{P} \\ \mathbf{P}^\top & \mathbf{I} \end{bmatrix} \quad \text{(If } \alpha \gg 1\text{)}, \tag{77}$$

$$\mathbf{N}_2 = \frac{\mathbf{A}_3\mathbf{V}}{\sqrt{d_F}} = \begin{bmatrix} \mathbf{I} & \mathbf{P} \\ \mathbf{P}^\top & \mathbf{I} \end{bmatrix}\begin{bmatrix} 0 \\ Y_N \end{bmatrix}/(2\sqrt{d_f}) = \begin{bmatrix} \mathbf{P}Y_N \\ Y_N \end{bmatrix}/(2\sqrt{d_f}). \tag{78}$$

Here, we used the fact that we chose the positional embeddings to be an orthogonal basis, and $\varphi\varphi^\top = \mathbf{I}$, as well as the fact that $\mathbf{P}\mathbf{P}^\top = \mathbf{I}$ for any permutation matrices. The term $\frac{1}{2}$ in the third equation came from the fact that each row of $\mathbf{A}_1$ contains two non-zero values that are also equal. The probability gets divided between the two of them in the softmax if the logits are scaled large enough, and the approximation becomes arbitrarily accurate.

From now on, since we are interested in the denoising output only for the reactant side, we drop out the reactant side $\mathbf{Y}^{\mathcal{N}}$ and only focus on the $\mathbf{P}\mathbf{Y}^{\mathcal{N}}$ part. We choose the final $MLP_{\mathbf{N}_2}$ to scale the output by some factor $\beta \gg 1$:

$$\mathbf{N}_{out} = \beta\mathbf{N}_2 + \mathbf{N} \tag{79}$$

$$n_\theta(\mathbf{P}\mathbf{X}_t^*, \mathbf{Y}^*)^{\mathcal{N}} = \text{softmax}(\beta\mathbf{P}\mathbf{Y}^{\mathcal{N}} + \mathbf{N}) \approx \mathbf{P}\mathbf{Y}^{\mathcal{N}}. \tag{80}$$

Here, the approximation can be made arbitrarily accurate by scaling $\beta$ to a higher value, since the logits will become more and more peaked towards the values where $P\mathbf{Y}^{\mathcal{N}}$ equals one instead of zero. This showcases how the attention mechanism in the Graph Transformer pairs with atom mapping-based orthogonal positional encodings to achieve the identity function from products to reactants.

### A.7 EXTENDING THEOREM 1 TO SMALLER TIME STEPS

We prove the following result characterizing the optimal denoiser output for a more restricted setup of absorbing-state diffusion and on an identity data with no edges (i.e., just a set of nodes):

$$D_\theta(X_t, Y)_{i,k} = \begin{cases} Y_{i,k}, & \text{if } X_{t,i} \text{ is not in the absorbing state} \\ \hat{y}_k^M, & \text{if } X_{t,i} \text{ is in the absorbing state} \end{cases} \quad \text{where } \hat{y}_k^M = \frac{\sum_{j \in M_t} Y_{j,k}}{\sum_{j \in M_t} \sum_k Y_{j,k}}$$

where $M_t$ refers to the set of nodes in the mask state. We will include the proof as an additional comment below. In words, the optimal output for the masked nodes is the marginal distribution of the nodes that have not been de-masked yet, that is, the mean of all possible placements of the remaining nodes.

The analysis for graphs with edges becomes more involved, but we postulate that a similar analysis would show that the output would converge to a mean of all possible permutations of $Y$ that are in agreement with the current set of subgraphs $X_t$.

To recap, we have the simplified scenario where we generate a set of nodes $X$ conditioned on another set of nodes $Y$, without edges. The goal is to characterize the optimal permutation equivariant denoiser for this process at any noise level.

**Definitions.** Let $X_0 \in \{0,1\}^{N \times K}$ be the initial set of $N$ nodes, each represented by a one-hot vector of dimension $K$. We condition on $Y \in \{0,1\}^{N \times K}$, where $X_0 = Y$ in our data. The forward process is an absorbing state diffusion, defined as follows for $t \in 0, 1, \ldots, T$:

$$q(X_{t,i} = M | X_{0,i}) = t/T \tag{81}$$
$$q(X_{t,i} = X_{0,i} | X_{0,i}) = 1 - t/T \tag{82}$$

where $M$ represents the additional "mask" state. We denote the denoiser as $D_\theta(X_t, Y)$, which is assumed to be permutation equivariant with respect to $X_t$.

**Notation** For a given $X_t$, we define: $M(X_t) \subset 1, \ldots, N$: Set of indices of masked nodes $U(X_t) = 1, \ldots, N \setminus M(X_t)$: Set of indices of non-masked nodes $\hat{y}_k^{M(X_t)} = \frac{1}{|M(X_t)|} \sum_{i \in M(X_t)} Y_{i,k}$: Marginal distribution of node values in $Y$ for the node indices that are masked in $X_t$

**Result**

**Theorem 5.** *The optimal permutation equivariant denoiser $D_\theta(X_t, Y)$ for the absorbing state diffusion process at any time $t$ and for any particular $X_t$ is given by: $D_\theta(X_t, Y)_{i,k} = Y_{i,k}, \text{if } i \in U(X_t)$ $D_\theta(X_t, Y)_{i,k} = \hat{y}_k^{M(X_t)}, \text{if } i \in M(X_t)$*

*Proof.* The cross-entropy loss for the denoiser at time $t$ is:

$$CE_t = -\mathbb{E}_{q(X_t|X_0)} \sum_{i=1}^N \sum_k X_{0,i,k} \log D_\theta(X_t, Y)_{i,k} \tag{83}$$

$$= -\mathbb{E}_{q(X_t|X_0)} \left[ \sum_{i \in U(X_t)} \sum_k X_{0,i,k} \log D_\theta(X_t, Y)_{i,k} + \sum_{i \in M(X_t)} \sum_k X_{0,i,k} \log D_\theta(X_t, Y)_{i,k} \right] \tag{84}$$

We consider the optimal denoiser output for non-masked and masked nodes separately:

**Non-masked nodes** ($i \in U(X_t)$) For these nodes, the optimal output is $D_\theta(X_t, Y)_{i,k} = Y_{i,k}$, since $Y_{i,k} = X_{0,i,k}$.

**Masked nodes ($i \in M(X_t)$)** Let's focus on a particular instantiation of $X_t$. Due to the permutation equivariance of $D_\theta$, for any permutation $P$ that only permutes indices within $M(X_t)$, we have: $D_\theta(PX_t, Y) = PD_\theta(X_t, Y)$. Moreover, because all nodes in $M(X_t)$ are in the same mask state, all such permutations of $X_t$ are identical: $D_\theta(PX_t, Y) = D_\theta(X_t, Y)$ Combining these facts, we conclude that for any $i, j \in M(X_t)$: $D_\theta(X_t, Y)_i = D_\theta(X_t, Y)_j$ Thus, the denoiser must output the same value for all masked nodes in a given $X_t$. Now, consider the part of the cross-entropy loss corresponding to the masked nodes for this particular $X_t$:

$$CE_t^M(X_t) = - \sum_{i \in M(X_t)} \sum_k X_{0,i,k} \log D_\theta(X_t, Y)_{i,k} \tag{85}$$

$$= - \sum_k \left( \sum_{i \in M(X_t)} X_{0,i,k} \right) \log D_\theta(X_t, Y)_{m,k} \tag{86}$$

where $m$ is any index in $M(X_t)$. Observe that $\sum_{i \in M(X_t)} X_{0,i,k}$ is proportional to the marginal distribution of the masked nodes in $X_0$ (which is equal to $Y$). Using our defined notation, we can rewrite the cross-entropy as: $CE_t^M(X_t) \propto -|M(X_t)| \sum_k \hat{y}_k^{M(X_t)} \log D_\theta(X_t, Y)_{m,k}$ The optimal $D_\theta(X_t, Y)_{m,k}$ that minimizes this expression is $\hat{y}_k^{M(X_t)}$. Combining the results for masked and non-masked nodes yields the stated optimal denoiser. $\square$

## A.8 ELABORATION ON WHY WE CAN CHOOSE ANY NODE MAPPING MATRIX DURING INFERENCE

We note that all node mapping matrices can be characterized by a base permutation matrix $\mathbf{P}^{\mathbf{Y} \to \mathbf{X}}$ left multiplied by different permutations $\mathbf{R}$. Applying Theorem 2: $p_\theta(\mathbf{X}_0 \,|\, \mathbf{Y}, \mathbf{R}\mathbf{P}^{\mathbf{Y} \to \mathbf{X}}) = p_\theta(\mathbf{R}^{-1}\mathbf{X}_0 \,|\, \mathbf{Y}, \mathbf{P}^{\mathbf{Y} \to \mathbf{X}})$, where $\mathbf{R}^{-1}$ is just another permutation matrix, shows that the distribution equals $p_\theta(\mathbf{X}_0 \,|\, \mathbf{Y}, \mathbf{P}^{\mathbf{Y} \to \mathbf{X}})$, up to some permutation. In fact, by sampling the node mapping matrix randomly, the distribution $p_\theta(\mathbf{X}_0 \,|\, \mathbf{Y}) = \sum_{\mathbf{P}^{\mathbf{Y} \to \mathbf{X}}} p_\theta(\mathbf{X}_0 \,|\, \mathbf{Y}, \mathbf{P}^{\mathbf{Y} \to \mathbf{X}}) p(\mathbf{P}^{\mathbf{Y} \to \mathbf{X}})$ becomes invariant to any permutation of $\mathbf{X}_0$ and $\mathbf{Y}$, although this is not strictly necessary in our context.

## B DETAILS ON CONDITIONAL GRAPH DIFFUSION

**Our transition matrices** To define $\mathbf{Q}_t^{\mathcal{N}}$ and $\mathbf{Q}_t^E$, we adopt the absorbing-state formulation from Austin et al. (2021), where nodes and edges gradually transfer to the *absorbing state* $\perp$. Formally, we give the generic form of the transition matrix $\mathbf{Q}_t$ for node input $\mathbf{X}^{\mathcal{N}} \in \mathbb{R}^{N_X \times N_X}$ [1]

$$\mathbf{Q}_t = (1 - \beta_t)\mathbf{I} + \beta_t \mathbb{1} e_\perp^\top, \tag{87}$$

where $\beta_t$ defines the *diffusion schedule* and $e_\perp$ is one-hot on the absorbing state $\perp$. For completeness, we list the other two common transitions relevant to our application. The first is the uniform transition as proposed by Hoogeboom et al. (2021)

$$\mathbf{Q}_t = (1 - \beta_t)\mathbf{I} + \beta_t \frac{\mathbb{1}\mathbb{1}^\top}{K} \tag{88}$$

where $\beta_t$, $I$ are as before and $K$ is the number of element (edge or node) types, i.e., the number of input features for both nodes and edges. Vignac et al. (2023) also proposed a marginal transition matrix

$$\mathbf{Q}_t^{\mathcal{N}} = (1 - \beta^t)\mathbf{I} + \beta^t \mathbb{1}(m^{\mathcal{N}})^\top \quad \text{and} \quad \mathbf{Q}_t^E = (1 - \beta^t)\mathbf{I} + \beta^t \mathbb{1}(m^E)^\top \tag{89}$$

which they argued leads to faster convergence. In this case, $m^{\mathcal{N}} \in \mathbb{R}^{K_a}$ and $m^E \in \mathbb{R}^{K_b}$ are row vectors representing the marginal distributions for node and edge types respectively. We tested all three types of transition matrices in early experiments and noted the absorbing state model to be slightly better than the others. The marginal $q(\mathbf{X}_t \,|\, \mathbf{X}_0)$ and conditional posterior $q(\mathbf{X}_{t-1} \,|\, \mathbf{X}_t, \mathbf{X}_0)$ also have a closed form for all of these transition matrices.

---

[1]The only difference between $\mathbf{Q}_t^{\mathcal{N}}$ and $\mathbf{Q}_t^E$ for the absorbing-state and uniform transitions is the dimensions of $I$, $e$, $\mathbb{1}$ and the value of $K$. We therefore give a generic form for both and imply choosing the right dimensions for each case.

**Noise schedule** We use the mutual information noise schedule proposed by Austin et al. (2021), which leads to

$$\frac{t}{T} = 1 - \frac{I(\mathbf{X}_t; \mathbf{X}_0)}{H(\mathbf{X}_0)} = \frac{H(\mathbf{X}_0, \mathbf{X}_t) - H(\mathbf{X}_t)}{H(\mathbf{X}_0)} = \frac{\sum_{\mathbf{X}_0, \mathbf{x}_t} p(\mathbf{X}_0) q(\mathbf{X}_t \mid \mathbf{X}_0) \log \frac{q(\mathbf{X}_t \mid \mathbf{X}_0)}{\sum_{\mathbf{X}_0'} p(\mathbf{X}_0') q(\mathbf{X}_t \mid \mathbf{X}_0')}}{\sum_{\mathbf{X}_0} p(\mathbf{X}_0) \log p(\mathbf{X}_0)}$$

(90)

For absorbing state diffusion, these equations lead to $\beta_t = \frac{1}{T-t+1}$. Similarly, the total transition probability to the absorbing state at time $t$ has a simple form: $q(\mathbf{X}_t = \perp \mid \mathbf{X}_0) = \frac{t}{T}$.

**Forward process posterior** For transition matrices that factorize over dimensions, we have

$$q(\mathbf{X}_{t-1,i,:} \mid \mathbf{X}_{t,i,:}, \mathbf{X}_{0,i,:}) \sim \frac{\mathbf{X}_t \mathbf{Q}_t^\top \cdot \mathbf{X}_{0,i,:} \bar{\mathbf{Q}}_{t-1}}{\mathbf{X}_{0,i,:} \bar{\mathbf{Q}}_t \mathbf{X}^\top}$$

(91)

where $\mathbf{X}_{i,:}$ is the one-hot encoding of $i^{\text{th}}$ node/edge of the graph, in row vector format.

**Variational lower-bound loss** Diffusion models are commonly trained by minimizing the negative variational lower-bound on the model's likelihood (Ho et al., 2020). Austin et al. (2021) discuss the difference between optimizing the ELBO and cross-entropy losses and show that the two losses are equivalent for the absorbing-state transition. We choose to use cross-entropy, similar to Vignac et al. (2023), due to faster convergence during training. We include the formula for the ELBO in Eq. (92) for completeness.

$$\mathcal{L}_{\text{vb}} = \mathbb{E}_{\mathbf{X}_0 \sim q(\mathbf{X}_0)} \Bigg[ \underbrace{\text{KL}\big[ q(\mathbf{X}_T \mid \mathbf{X}_0) \,\|\, p(\mathbf{X}_T) \big]}_{\mathcal{L}_T}$$

$$+ \underbrace{\sum_{t=2}^{T} \mathbb{E}_{\mathbf{X}_t \sim q(\mathbf{X}_t \mid \mathbf{X}_0)} \text{KL}\big[ q(\mathbf{X}_{t-1} \mid \mathbf{X}_t, \mathbf{X}_0) \,\|\, p_\theta(\mathbf{X}_{t-1} \mid \mathbf{X}_t) \big]}_{\mathcal{L}_{t-1}}$$

$$- \underbrace{\mathbb{E}_{\mathbf{X}_1 \sim q(\mathbf{X}_1 \mid \mathbf{X}_0)} \log p_\theta(\mathbf{X}_0 \mid \mathbf{X}_1)}_{\mathcal{L}_0} \Bigg]$$

(92)

We also note that we use this quantity as part of the scoring function mentioned in Sec. 4.

**Data encoding and atom-mapping** We illustrate our graph encoding using atom-mapping and permutation matrices in Fig. A10.

**Connection between the denoising parameterization** $\tilde{p}(\mathbf{X}_0 \mid \mathbf{X}_t, \mathbf{Y})$ **and** $p(\mathbf{X}_{t-1} \mid \mathbf{X}_t, \mathbf{Y})$. Here, we elaborate on how does $\tilde{p}(\mathbf{X}_0 \mid \mathbf{X}_t, \mathbf{Y})$ connect with $p(\mathbf{X}_{t-1} \mid \mathbf{X}_t, \mathbf{Y})$.

Recall that the reverse transition is factorized with respect to nodes and edges:

$$p_\theta(\mathbf{X}_{t-1} \mid \mathbf{X}_t, \mathbf{Y}) = \prod_{i=1}^{N_X} p_\theta(\mathbf{X}_{t-1}^{\mathcal{N},i} \mid \mathbf{X}_t, \mathbf{Y}) \prod_{i,j}^{N_X} p_\theta(\mathbf{X}_{t-1}^{E,i,j} \mid \mathbf{X}_t, \mathbf{Y}).$$

(93)

We parameterize the transition as follows:

$$p_\theta(\mathbf{X}_{t-1} \mid \mathbf{X}_t, \mathbf{Y}) = \sum_{\mathbf{X}_0} q(\mathbf{X}_{t-1} \mid \mathbf{X}_t, \mathbf{X}_0) \tilde{p}_\theta(\mathbf{X}_0 \mid \mathbf{X}_t, \mathbf{Y}).$$

(94)

The connection between this parameterization and the probabilities $p_\theta(\mathbf{X}_{t-1}^{\mathcal{N},i} \mid \mathbf{X}_t, \mathbf{Y})$ is obtained by noting that both $q(\mathbf{X}_{t-1} \mid \mathbf{X}_t, \mathbf{X}_0)$ and $\tilde{p}_\theta(\mathbf{X}_0 \mid \mathbf{X}_t, \mathbf{Y})$ factorize over dimensions:

$$q(\mathbf{X}_{t-1} \mid \mathbf{X}_t, \mathbf{X}_0) = \prod_i^{N_\mathbf{X}} q(\mathbf{X}_{t-1}^{\mathcal{N},i} \mid \mathbf{X}_t^{\mathcal{N},i}, \mathbf{X}_0^{\mathcal{N},i}) \prod_{i,j}^{N_\mathbf{X}} q(\mathbf{X}_{t-1}^{E,i,j} \mid \mathbf{X}_t^{E,i,j}, \mathbf{X}_0^{E,i,j})$$

(95)

$$\tilde{p}_\theta(\mathbf{X}_0 \mid \mathbf{X}_t, \mathbf{Y}) = \prod_i^{N_\mathbf{X}} \tilde{p}_\theta(\mathbf{X}_0^{\mathcal{N},i} \mid \mathbf{X}_t, \mathbf{Y}) \prod_{i,j}^{N_\mathbf{X}} \tilde{p}_\theta(\mathbf{X}_0^{E,i,j} \mid \mathbf{X}_t, \mathbf{Y})$$

(96)

Figure A10: Illustrating graph encoding using atom-mapping.

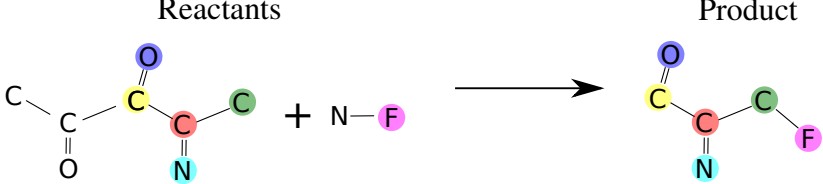

Atom mapping from the nodes of $\mathbf{Y}$ to the nodes of $\mathbf{X}$

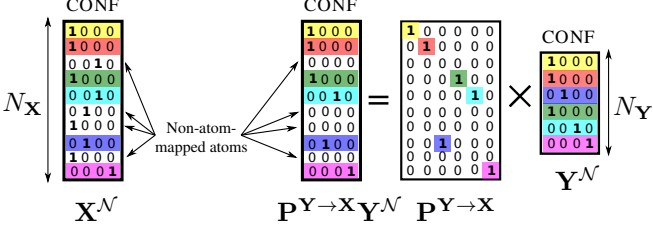

Atom mapping from the edges of $\mathbf{Y}$ to the edges of $\mathbf{X}$

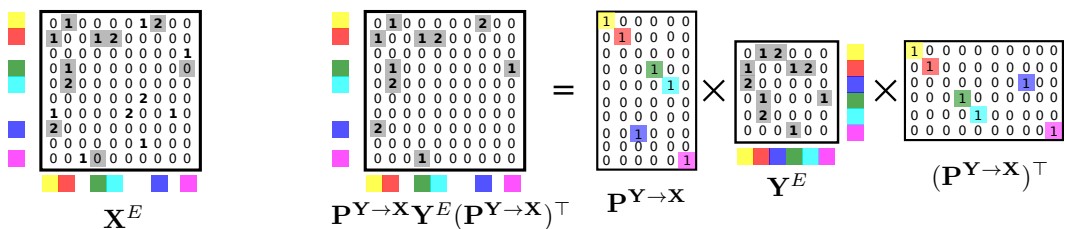

Plugging these in to Eq. (94) and expanding the sum, we get

$$
p_\theta(\mathbf{X}_{t-1} \,|\, \mathbf{X}_t, \mathbf{Y}) = \sum_{\mathbf{X}_0^{\mathcal{N},0}} \sum_{\mathbf{X}_0^{\mathcal{N},1}} \cdots \sum_{\mathbf{X}_0^{E,N_\mathbf{X},N_\mathbf{X}}} \Bigg( \prod_i^{N_\mathbf{X}} q\big(\mathbf{X}_{t-1}^{\mathcal{N},i} \,|\, \mathbf{X}_t^{\mathcal{N},i}, \mathbf{X}_0^{\mathcal{N},i}\big) \tilde{p}_\theta\big(\mathbf{X}_0^{\mathcal{N},i} \,|\, \mathbf{X}_t, \mathbf{Y}\big)
$$
$$
\prod_{i,j}^{N_\mathbf{X}} q\big(\mathbf{X}_{t-1}^{E,i,j} \,|\, \mathbf{X}_t^{E,i,j}, \mathbf{X}_0^{E,i,j}\big) \tilde{p}_\theta\big(\mathbf{X}_0^{E,i,j} \,|\, \mathbf{X}_t, \mathbf{Y}\big) \Bigg)
$$
$$
(97)
$$

Gathering terms together, we get

$$= \left( \sum_{\mathbf{X}_0^{\mathcal{N},0}} q\big(\mathbf{X}_{t-1}^{\mathcal{N},0} \mid \mathbf{X}_t^{\mathcal{N},0}, \mathbf{X}_0^{\mathcal{N},0}\big) \tilde{p}_\theta\big(\mathbf{X}_0^{\mathcal{N},0} \mid \mathbf{X}_t, \mathbf{Y}\big) \right) \cdot \tag{98}$$
$$\underbrace{\phantom{\sum_{\mathbf{X}_0^{\mathcal{N},0}}}}_{=p_\theta(\mathbf{X}_{t-1}^{\mathcal{N},0} \mid \mathbf{X}_t, \mathbf{Y})}$$

$$\left( \sum_{\mathbf{X}_0^{\mathcal{N},1}} q\big(\mathbf{X}_{t-1}^{\mathcal{N},1} \mid \mathbf{X}_t^{\mathcal{N},1}, \mathbf{X}_0^{\mathcal{N},1}\big) \tilde{p}_\theta\big(\mathbf{X}_0^{\mathcal{N},1} \mid \mathbf{X}_t, \mathbf{Y}\big) \right) \cdot \tag{99}$$
$$\underbrace{\phantom{\sum_{\mathbf{X}_0^{\mathcal{N},1}}}}_{=p_\theta(\mathbf{X}_{t-1}^{\mathcal{N},1} \mid \mathbf{X}_t, \mathbf{Y})}$$

$$\ldots \tag{100}$$

$$\left( \sum_{\mathbf{X}_0^{E,N_\mathbf{x},N_\mathbf{x}}} q\big(\mathbf{X}_{t-1}^{E,N_\mathbf{x},N_\mathbf{x}} \mid \mathbf{X}_t^{E,N_\mathbf{x},N_\mathbf{x}}, \mathbf{X}_0^{E,N_\mathbf{x},N_\mathbf{x}}\big) \tilde{p}_\theta\big(\mathbf{X}_0^{E,N_\mathbf{x},N_\mathbf{x}} \mid \mathbf{X}_t, \mathbf{Y}\big) \right) \tag{101}$$
$$\underbrace{\phantom{\sum_{\mathbf{X}_0^{E,N_\mathbf{x},N_\mathbf{x}}}}}_{=p_\theta(\mathbf{X}_{t-1}^{E,N_\mathbf{x},N_\mathbf{x}} \mid \mathbf{X}_t, \mathbf{Y})}$$

$$= \prod_{i=1}^{N_X} p_\theta(\mathbf{X}_{t-1}^{\mathcal{N},i} \mid \mathbf{X}_t, \mathbf{Y}) \prod_{i,j}^{N_X} p_\theta(\mathbf{X}_{t-1}^{E,ij} \mid \mathbf{X}_t, \mathbf{Y}). \tag{102}$$

## C EXPERIMENTAL SETUP

### C.1 COPYING TASK ON THE GRID DATASET

**Data generation** We generate $5 \times 5$ fully connected grids then noise them by flipping a fixed portion of edges chosen at random. We set the portion of edges to be flipped to $5\%$ of the total number of edges (i.e., for $5 \times 5$ grids, 31 edges get flipped).

**Neural network** We use the same neural network architecture for both the aligned and unaligned denoisers. We use a graph transformer Dwivedi & Bresson (2021) with 2 layers, 1 attention head, and dimension 16 for the hidden layer. The activation function used is ReLu, and dropout rate of 0.1.

**Training hyperparameters** We train both models for 10 epochs, using Adam optimizer with a learning rate of $0.01$. We set the batch size to 32.

### C.2 DATA: USPTO DATA SETS

All open-source data sets available for reaction modelling are derived in some form from the patent mining work of Lowe (2012). We distinguish 5 subsets used in previous work: 15k, 50k, MIT, Stereo, and full (original data set). Table A3 provides key information about the subsets.

Table A3: UPSTO-50K subsets used in retrosynthesis

| Subset | Introduced by | # of reactions | Preprocessing & Data split (script) |
|--------|---------------|----------------|-------------------------------------|
| Full | Lowe (2012) | 1 808 938 | Dai et al. (2019) |
| Stereo | Schwaller et al. (2018) | 1 002 970 | Schwaller et al. (2019b) |
| MIT | Jin et al. (2017) | 479 035 | - |
| 50k | Schneider et al. (2016) | 50 016 | Dai et al. (2019) |

**15k** is proposed by Coley et al. (2017a). The subset includes reactions covered by the 1.7k most common templates. All molecules appearing in the reaction are included to model the involvement of reagents and solvents despite not contributing with atoms to the product.

**50k** is preprocessed by Schneider et al. (2016). The goal of the analysis is to assign roles (reactant, reagent, solvent) to different participants in a reaction through atom mapping. This effort led to the

creation of an atom-mapped and classified subset of around 50k reactions, which is used nowadays as a benchmark for retrosynthesis tasks. It is not clear how said subset was selected.

**MIT** is used by Jin et al. (2017). The preprocessing is described as 'removing duplicate and erroneous reactions' with no further explanation of what qualifies as an erroneous reaction. The output of this filtering is a data set of 49k reactions (from an original set of 1.8M reactions).

**Stereo** is proposed by Schwaller et al. (2018). The authors apply a more flexible filtering strategy compared to USPTO-MIT. Their data set only discards 800k reactions from the original data set because they are duplicates or they could not be canonicalized by RDKit. In addition, the data set only considers single-product reactions (92% of the full data set), as opposed to splitting multi-product reactions. The preprocessing steps include removing reagents (molecules with no atoms appearing in the product), removing hydrogen atoms from molecules, discarding atom-mapping information and canonicalizing molecules. In addition, since the original method applied to this subset is a language model, tokenization is performed on the atoms.

**Full** is preprocessed by Dai et al. (2019). The processing includes removing duplicate reactions, splitting reactions with multiple products into multiple reactions with one product, removing reactant molecules appearing unchanged on the product side, removing all reactions with bad atom-mapping (i.e., when the sorted mapping between products and reactants is not one-to-one), and removing bad products (missing mapping, or not parsed by Rdkit).

**Our choice** Similar to many other works on retrosynthesis, we use 50k as the main data set to evaluate our method.

### C.3 Notes on Our Sampling and Ranking Procedures

**Duplicate removal** Removing duplicates from the set of generated precursors is a common methodology in retrosynthesis, albeit often not discussed explicitly in papers. The benefit of duplicate removal is to ensure that an incorrect molecule that is nevertheless judged as the best one according to the ranking scheme does not fill up all of the top-$k$ positions after ranking. While this does not affect top-1 scores, not removing the duplicates would degrade the other top-$k$ scores significantly.

**Choice of scoring function** Specifically, we use the following formula for approximating the likelihood of the sample under the model.

$$s(\mathbf{X}) = (1 - \lambda) \frac{\text{count}(\mathbf{X})}{\sum_{\mathbf{X}'} \text{count}(\mathbf{X}')} + \lambda \cdot \frac{e^{\text{elbo}(\mathbf{X})}}{\sum_{\mathbf{X}'} e^{\text{elbo}(\mathbf{X}')}}, \tag{103}$$

where count(.) returns the number of occurrences of sample X in the set of generated samples (by default, 100), elbo(.) computes the variational lower bound of the specific sample under the model, and $\lambda$ is a weighting hyperparameter. The sums are taken over the set of generated samples. Intuitively, the idea is to provide an estimate of the likelihood from two routes. The ELBO is an estimate (a lower bound) of $\log p_\theta(\mathbf{X})$, and exponentiating and normalizing gives an estimate of the probability distribution. Since the same reactants are often repeated in a set of 100 samples, the counts can be used as a more direct proxy, although they inherently require a relatively large amount of samples to limit the variance of this estimate.

We set the value of $\lambda$ to be 0.9, although we find that the top-$k$ scores are not at all sensitive to variation in the exact value, as long as it is below 1, and the count information is used. Thus, the counts seem more important than the ELBO, which may be due to the lower bound nature of the ELBO or stochasticity in estimating its value. More accurate likelihood estimation schemes for diffusion models, such as exact likelihood values using the probability flow ODE (Song et al., 2021), could be a valuable direction for future research in the context of retrosynthetic diffusion models.

### C.4 Details on Stereochemistry

Our model does not explicitly consider changes in stereochemistry in the reaction, but instead, we use the atom mappings implicitly assigned to the samples by the model to transfer the chiral tags from the products to the reactants. The initial choice of $\mathbf{P}^{\mathbf{Y} \to \mathbf{X}}$ at the start of sampling can be considered to be the atom mapping of the generated reactants, given that the model has been trained on correct atom mappings.

For the chiral tags, we take the ground-truth SMILES for the product molecules from the dataset and assign the corresponding chiral tag to the corresponding atom mapping on the generated reactants. For cis/trans isomerism, we use the `Chem.rdchem.BondDir` bond field in rdkit molecules and transfer them to the reactant side based on the atom mapping of the pair of atoms at the start and end of the bond.

Note that when using rdkit, transferring chirality requires some special care: The chiral tags `Chem.ChiralType.CHI_TETRAHEDRAL_CCW` and `Chem.ChiralType.CHI_TETRAHEDRAL_CW` are defined in the context of the order in which the bonds are attached to the chiral atom in the molecule data structure. Thus, the chiral tag sometimes has to be flipped to retain the correct stereochemistry, based on whether the order of the bonds is different on the reactant molecule data structure and the product molecule data structure.

## C.5 DETAILS OF THE EVALUATION PROCEDURE

**Top-$k$ scores**  We evaluate the top-$k$ scores by ranking the list of generated and deduplicated samples and calculating the percentage of products for which the ground-truth reactants are in the first k elements in the list.

**Mean reciprocal rank**  We formally define the MRR as $\text{MRR} = \mathbb{E}_{p(r)}[r^{-1}]$. Verbally, it is the expected value of the inverse of the amount of reactant suggestions that the model makes before encountering the ground truth, and as such measures how early on is the correct reactant encountered in the ranked samples. It also incorporates the intuition that the difference between obtaining the correct reactants in, say, the 9th or the 10th position, is not as significant as the difference between the 1st and 2nd positions.

While we do not have direct access to the entire $p(r)$ just based on the top-$k$ scores, we can estimate it with a uniform distribution assumption on $r$ within the different top-$k$ ranges. Formally, we define four sets $S_1, S_2, S_3, S_4 = \{1\}, \{2, 3\}, \{4, 5\}, \{6, 7, 8, 9, 10\}$ in which $p(r)$ is assumed to be uniform, and $s(r) \in \{1, 2, 3, 4\}$ denotes the group that rank $r$ belongs to. Top-$k$ is denoted as $\text{top}_k$, where $k \in \{1, 3, 5, 10\}$ in our case. Note that they are also equal to the cumulative distribution of $p(r)$ until $k$. We thus define $\hat{p}(r) = \frac{\text{top}_{\max(G_{s(r)})} - \text{top}_{\max(G_{s(r)-1})}}{|G_{s(r)}|}$ and $\hat{p}(1) = \text{top}_1$. For the case where the ground truth was not in the top-10, we assume it is not recovered and place the rest of the probability mass on $p(\infty)$. Our MRR estimate is then defined as

$$\widehat{\text{MRR}} = \sum_{r=1}^{10} \hat{p}(r) \frac{1}{r}. \tag{104}$$

In cases where we do not have top-$k$ values for all {1,3,5,10} (such as the Augmented Transformer in Table 2 for top-3), we assume that $\hat{p}(r)$ is constant in the wider interval between the preceding and following top-$k$s (2–5 in the case of the top-3 missing).

## C.6 NEURAL NETWORK ARCHITECTURE, HYPERPARAMETERS, AND COMPUTE RESOURCES

We discuss here the neural network architecture and hyper-parameters we choose. Our denoiser is implemented as a Graph Transformer (Dwivedi & Bresson, 2021), based on the implementation of Vignac et al. (2023) with additional graph-level level features added to the input of the model. See Vignac et al. (2023) for an in-depth discussion of the neural network.

In all of our models, we use 9 Graph Transformer layers. When using Laplacian positional encodings, we get the 20 eigenvectors of the Graph Laplacian matrix with the largest eigenvalues and assign to each node a 20-dimensional feature vector.

We use a maximum of 15 'blank' nodes, in practice meaning that the models have the capacity to add 15 additional atoms on the reactant side. In another detail, following Vignac et al. (2023), we weigh the edge components in the cross-entropy loss by a factor of 5 compared to the node components.

We used a batch size of 16 for the models where the expanded graph containing X and Y as subgraphs is given as input. These models were trained for approximately 600 epochs with a single A100/V100/AMD MI250x GPU. For the model where alignment is done by concatenating **Y** along the feature dimension in the input, the attention map sizes were smaller and we could fit a larger

Figure A11: Left: An example reaction from USPTO-50k where the reactant is not possible to predict with a synthon-based model. Right: example reactants generated by our model (includes the ground-truth one). The SMILES-string for the reaction is `ON=C1CCc2oc3ccccc3c21»O=C1NCCc2oc3ccccc3c21`.

batch of 32 with a single V100 GPU. This model was trained for 600 epochs. The training time for all of our models was approximately three days. In early experiments and developing the model, we trained or partially trained multiple models that did not make it to the main paper. Sampling 100 samples for one product with $T = 100$ from the model takes roughly 60 seconds with the current version of our code with an AMD MI250x GPU, and 100 samples with $T = 10$ takes correspondingly about 6 seconds. It is likely that the inference could be optimized, increasing the sample throughput.

The reported models were chosen based on evaluating different checkpoints with 10 diffusion steps on the validation set for different checkpoints and chose the best checkpoint based on the MRR score.

## D  COMPARISION TO RETROSYNTHETIC BASELINES

**Overview of retrosynthetic baselines**  There are three main types of retrosynthesis models (Liu et al., 2023). Template-based models depend on the availability and quality of hard-coded chemical rules (Segler & Waller, 2017; Xie et al., 2023). Synthon-based models are limited by their definition of a reaction center, which does not necessarily hold for complex reactions (Yan et al., 2020; Shi et al., 2020a; Wang et al., 2023). See Fig. A11 for an example of a reaction impossible for synthon-based models but which our model gets correctly. Template-free methods are the most scalable since they do not use any chemical assumptions in their design but perform suboptimally compared to template-based methods on benchmarks like UPSTO-50k (Wan et al., 2022; Seo et al., 2021). Efforts to bridge the gap between the template-based and template-free paradigms include methods investigating pretraining (Zhong et al., 2022; Jiang et al., 2023). Note that our method is in between template-free and template-based methods, since it uses atom-mapping information but not an explicit (and thus limited) set of templates. We put our models in their own category in the results table.

**Methods with a different evaluation procedure**  Despite Igashov et al. (2024)'s RetroBridge being a closely related to a diffusion model, we cannot include it in a straightforward comparison because it discards atom charges from the ground truth smiles during evaluation. Specifically, the model uses only atom types as node features and compares the generated samples to the smiles reconstructed from the ground truth data through the same encoding (i.e., without charges too) [2]. To compare our model to RetroBridge fairly, we trained our DiffAlign-PE+Skip on uncharged atom-labels, and evaluated it without considering atom charge nor stereochemistry. The results are shown in App. D. Many recently published methods for retrosynthesis have capped out their top-1 scores in the 49%-53% range (see our Table 2), indicating that the increase we achieve over RetroBridge's top-1 score (of over 5%) is quite significant. Furthermore, the results we achieve are with 5 times less denoising steps (100 instead of 500). For a closer look at the performance at different step counts, we invite the reader to compare Figure 5 in (Igashov et al., 2024) to our Figure Fig. 6. The comparison shows that Retrobridge's top-1 score decreases to almost 0 at a step count more than 10. In contrast, our top-1 is close to 50% even at 5 steps, and higher than Retrobridge at 100 steps. We attribute this improvement to our more careful analysis of alignment, and utilising multiple alignment techniques.

---

[2]This can be seen in the code shared by Igashov et al. (2024): `https://github.com/igashov/RetroBridge`

Table A4: Comparison of top-$k$ accuracy and MRR on the USPTO-50k test set without charges and without stereochemical information, following the evaluation setup of Igashov et al. (2024)

| Model | k = 1 ↑ | k = 3 ↑ | k = 5 ↑ | k = 10 ↑ | $\widehat{\text{MRR}}$ ↑ |
|---|---|---|---|---|---|
| RetroBridge (T=500) | 50.8 | 74.1 | **80.6** | **85.6** | 0.622 |
| DiffAlign-PE+Skip (T=100) | **56.3** | **75.2** | 80.4 | 84.0 | **0.658** |

Table A5: Round-trip coverage and accuracy for different baselines. The methods are categorized as: template-based (TB), non-template-based (NTB), and models performing their evaluation without taking into account formal charges (NC) nor stereochemistry (NS). We achieve the highest coverage and top-1 accuracy among non-template-based methods. Our lower top-k>1 accuracy may be due to our model generating a higher number of unique predictions compared to competitors, visualized in Figure Fig. A12.

| | Model | Coverage (↑) | | | Accuracy (↑) | | |
|---|---|---|---|---|---|---|---|
| | | $k = 1$ | $k = 3$ | $k = 5$ | $k = 1$ | $k = 3$ | $k = 5$ |
| TB | GLN (Dai et al., 2019) | **82.5** | 92.0 | 94.0 | **82.5** | **71.0** | 66.2 |
| NTB | MEGAN (Sacha et al., 2021) | 78.1 | 88.6 | 91.3 | 78.1 | 67.3 | 61.7 |
| | Graph2SMILES (Tu & Coley, 2022) | — | — | — | 76.7 | 56.0 | 46.4 |
| | Retroformer$_{\text{aug}}$ (Wan et al., 2022) | — | — | — | 78.6 | **71.8** | **67.1** |
| | DiffAlign-PE+skip (ours) | **81.6** | **90.0** | **91.8** | **81.6** | 62.8 | 53.3 |
| NS | LocalRetro (Chen & Jung, 2021) | 82.1 | 92.3 | 94.7 | 82.1 | 71.0 | 66.7 |
| NS-NC | RetroBridge (Igashov et al., 2024) | 85.1 | 95.7 | 97.1 | 85.1 | **73.6** | **67.8** |
| | DiffAlign-PE+skip-NSNC (ours) | **87.6** | **96.4** | **97.6** | **87.6** | 69.3 | 59.1 |

**Methods with pretraining** Zhong et al. (2022) and Jiang et al. (2023) pre-train their models on the USPTO-Full and Pistachio data sets, respectively, and as such the results are not directly comparable to models trained on the standard USPTO-50k benchmark. Pretraining with diffusion models is an interesting direction for future research, but we consider it outside the scope of our work. Furthermore, comparison between models with different pretraining datasets and pretraining strategies has the danger of complicating comparisons, given that relative increases in performance could be explained by the model, the pretraining strategy, or the pretraining dataset. As such, we believe that standardized benchmarks like USPTO-50k are necessary when researching modelling strategies.

Another commonly used metric used to evaluate retrosynthesis model is round-trip accuracy, in which a forward prediction model is used to evaluate whether our samples can indeed produce the input product. *Coverage* considers a sample correct if it matches exactly the ground truth or is deemed suitable by the forward prediction oracle. *Accuracy* counts the percentage of valid precursors over all generated samples. We use the same prediction model as previous work, i.e., Molecular Transformer (Schwaller et al., 2019a). As can be seen from Table A5 our model outperforms all non-template-based baselines on all thresholds. We also highlight a known tradeoff between diversity and accuracy of generation in Fig. A12, which explains partly why our accuracy is lower than other baselines (we compare to samples from RetroBridge (Igashov et al., 2024) in particular as an illustration).

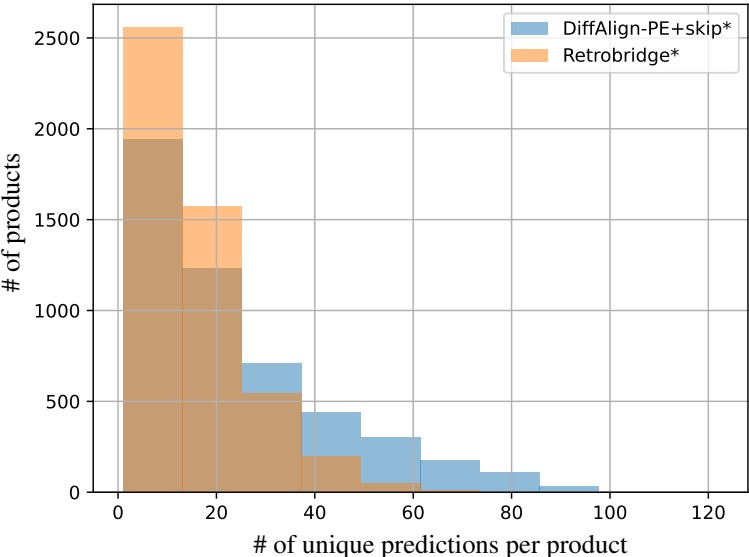

Figure A12: We compare the diversity (number of unique predictions per product) of our samples to RetroBridge (Igashov et al., 2024). It is important for a retrosynthetic model to generate diverse precursors, thus offering the practitioner with multiple synthesis strategies, especially given the limited accuracy of forward prediction models.

## E    ADDITIONAL ABLATIONS

**Disentangling the effect of the inductive biases due to graph positional encodings vs. the inductive bias of alignment.**    To disentangle the effect of the graph inductive bias brought by the graph Laplacian positional encodings and the alignment inductive bias brought by matching the positional encodings for the input and output graph, we trained two models:

1. A model where the only alignment method is the matched PE where the PE is generated from the graph Laplacian eigenvectors of the conditioning graph $Y$ (our default PE method).

2. A model where the only alignment method is matched PEs where the PE is generated by sampling random Gaussian noise vectors on the conditioning graph nodes $Y$ and placing these same vectors on the $X$ side.

Both contain alignment information, but the latter does not contain the additional inductive bias brought by the graph positional encoding. We show the results at roughly 10% of the full training budget for the results in Table 2. The results are listed in Table A6. The results are not significantly different, indicating that the inductive bias brought by the graph positional encodings is not as significant as the inductive bias of alignment.

**Different transitions.**    (Vignac et al., 2023) did not originally consider the absorbing state transitions we use. Instead, they used the uniform and marginal transitions. With the uniform transitions, each step provides a non-zero chance of transitioning from one state to any of the others. This converges to an uniform distribution over the node and edge states at $\mathbf{X}_T$. With the marginal transitions, these transitions are biased such that the distribution of nodes and edges in $p(\mathbf{X}_T)$ is the marginal distribution of the node and edge types in the original data. Since there are much more no-edges than edges in the data, this encodes a sparsity to the graphs in $p(\mathbf{X}_T)$.

To provide a fair comparison, we compare our absorbing state model and these models on roughly 10% of the full training budget (80 epochs), and list the results for 512 samples from the test set in

| Model | Top-1 | Top-3 | Top-5 | Top-10 | Top-50 | MRR |
|---|---|---|---|---|---|---|
| DiffAlign-PE-Gaussian | 45.51 | 68.55 | 73.24 | 75.78 | 83.20 | 0.5649 |
| DiffAlign-PE-Laplacian | 46.09 | 68.75 | 72.85 | 75.98 | 83.98 | 0.5686 |

Table A6: Performance comparison of the model with the only alignment method being the matched Gaussian noise encoding versus a model with the only alignment method being the matched Laplacian eigenvector-based positional encoding. Both contain alignment information, but the first does not contain additional inductive biases about the structure of the conditioning graph $\mathbf{Y}$. Results are obtained with 512 samples form the USPTO-50k test set and 10% of the full training budget used in Table 2.

| Model | Top-1 | Top-3 | Top-5 | Top-10 | Top-50 | MRR |
|---|---|---|---|---|---|---|
| Uniform-PE+Skip | 43.75 | 58.98 | 62.89 | 67.38 | 72.66 | 0.5156 |
| Marginal-PE+Skip | 44.73 | 60.94 | 63.28 | 66.60 | 69.92 | 0.5244 |
| Absorbing-PE+Skip | **49.22** | **70.90** | **74.61** | **77.93** | **84.96** | **0.5952** |

Table A7: Performance comparison of the PE+Skip models with different transition matrices, evaluated with 512 samples form the USPTO-50k test set and 10% of the full training budget.

Table A7. The absorbing-state model performs the best, whereas the marginal and uniform models perform slightly worse. The marginal transitions somewhat outperform the uniform transitions.

**Comparison to only using the skip connections.** Here, we provide a comparison to a model that only uses the skip connections, and no other alignment method. The results are again shown for roughly 10% of the training budget (epoch (80) for the PE+Skip model and for the Skip-only model in Table A8. It is evident that only having the skip connection does not work well, although it is better than an entirely unaligned model (see Table 2). This is likely due to the skip-only model having limited expressivity: The neural net does not get the alignment information in the input at all, and thus is unable to use it apart from the skip connection, which enables literally copying the $Y$ graph from input to output.

# F ADDING POST-TRAINING CONDITIONING TO DISCRETE DIFFUSION MODELS

In this section, we show a method to add additional controls and conditions on the used discrete diffusion model post-training. Note that this approach differs from (Vignac et al., 2023)'s in that our method is an adaptation of reconstruction guidance (Ho et al., 2022; Chung et al., 2023; Song et al., 2023a), while they adapt classifier-guidance (Sohl-Dickstein et al., 2015; Dhariwal & Nichol, 2021; Song et al., 2021) for their conditional model. While the notation is from the point of view of our graph-to-graph translation, the method here applies in general to any discrete diffusion model. We start out by following (Vignac et al., 2023) and (Sohl-Dickstein et al., 2015; Dhariwal & Nichol, 2021), and then make the novel connection to reconstruction guidance. We write Bayes' rule for an additional condition $y$ (e.g., a specified level of drug-likeness or synthesizability, or an inpainting mask)

$$p_\theta(\mathbf{X}_{t-1} \mid \mathbf{X}_t, \mathbf{Y}, y) \propto p(y \mid \mathbf{X}_{t-1}, \mathbf{X}_t, \mathbf{Y})p_\theta(\mathbf{X}_{t-1} \mid \mathbf{X}_t, \mathbf{Y}) \tag{105}$$
$$= p(y \mid \mathbf{X}_{t-1}, \mathbf{Y})p_\theta(\mathbf{X}_{t-1} \mid \mathbf{X}_t, \mathbf{Y}) \tag{106}$$

where the second equation was due to the Markovian structure of the generative process ($\mathbf{X}_{t-1}$ d-separates $y$ and $\mathbf{X}_t$). Now, we can take the log and interpret the probabilities as tensors

| Model | Top-1 | Top-3 | Top-5 | Top-10 | Top-50 | MRR |
|---|---|---|---|---|---|---|
| DiffAlign-Skip | 9.77 | 17.38 | 24.61 | 29.30 | 32.23 | 0.1517 |
| DiffAlign-PE+Skip (Laplacian) | **49.22** | **70.90** | **74.61** | **77.93** | **84.96** | **0.5952** |

Table A8: Performance comparison for a model with the only alignment method being skip connections and a model where we also use the Laplacian positional encodings, evaluated with 512 samples form the USPTO-50k test set and 10% of the full training budget

$\mathbf{P}_\theta(y\,|\,\mathbf{X}_{t-1},\mathbf{Y})$ and $\mathbf{P}_\theta(\mathbf{X}_{t-1}\,|\,\mathbf{X}_t,\mathbf{Y})$ defined in the same space as the one-hot valued tensors $\mathbf{X}_{t-1}$ and $\mathbf{X}_t$. We get:

$$\log\mathbf{P}_\theta(\mathbf{X}_{t-1}\,|\,\mathbf{X}_t,\mathbf{Y},y) \propto \log\mathbf{P}(y\,|\,\mathbf{X}_{t-1},\mathbf{Y}) + \log\mathbf{P}_\theta(\mathbf{X}_{t-1}\,|\,\mathbf{X}_t,\mathbf{Y}) \tag{107}$$

Similarly to (Vignac et al., 2023), we can now Taylor expand $\log\mathbf{P}_\theta(y\,|\,\mathbf{X}_{t-1},\mathbf{Y})$ around $\mathbf{X}_t$ with

$$\log\mathbf{P}(y\,|\,\mathbf{X}_{t-1},\mathbf{Y}) \approx \log\mathbf{P}(y\,|\,\mathbf{X}_t,\mathbf{Y}) + \nabla_{\mathbf{X}'_t}\log\mathbf{P}(y\,|\,\mathbf{X}'_t,\mathbf{Y})|_{\mathbf{X}'_t=\mathbf{X}_t}(\mathbf{X}_{t-1}-\mathbf{X}_t) \tag{108}$$

Given that we are interested in the distribution w.r.t. $\mathbf{X}_{t-1}$, the $\mathbf{X}_t$ terms are constant when we plug them in to Eq. (107), resulting in

$$\log\mathbf{P}_\theta(\mathbf{X}_{t-1}\,|\,\mathbf{X}_t,\mathbf{Y},y) \propto \nabla_{\mathbf{X}'_t}\log\mathbf{P}(y\,|\,\mathbf{X}'_t,\mathbf{Y})|_{\mathbf{X}'_t=\mathbf{X}_t}\mathbf{X}_{t-1} + \log\mathbf{P}_\theta(\mathbf{X}_{t-1}|\mathbf{X}_t,\mathbf{Y}) \tag{109}$$

Simplifying the notation to assume taking the gradient at $\mathbf{X}_t$, we can also write

$$\log\mathbf{P}_\theta(\mathbf{X}_{t-1}\,|\,\mathbf{X}_t,\mathbf{Y},y) \propto \nabla_{\mathbf{X}_t}\log\mathbf{P}_\theta(y\,|\,\mathbf{X}_t,\mathbf{Y})\mathbf{X}_{t-1} + \log\mathbf{P}_\theta(\mathbf{X}_{t-1}\,|\,\mathbf{X}_t,\mathbf{Y}) \tag{110}$$

In practice, the equation means that given $\log p_\theta(y\,|\,\mathbf{X}_t,\mathbf{Y})$, we get $p_\theta(\mathbf{X}_{t-1}\,|\,\mathbf{X}_t,\mathbf{Y},y)$ by adding the input gradient of $\log p_\theta(y\,|\,\mathbf{X}_t,\mathbf{Y})$ to the logits given by the regular reverse transition and re-normalizing.

It would be possible to approximate $\log p_\theta(y\,|\,\mathbf{X}_t,\mathbf{Y})$ by training an additional classifier, leading to classifier guidance (Sohl-Dickstein et al., 2015; Dhariwal & Nichol, 2021; Song et al., 2021) and the exact method presented in (Vignac et al., 2023). We go further by adapting these formulas to reconstruction guidance (Ho et al., 2022; Chung et al., 2023; Song et al., 2023a). These methods and more advanced versions (Dou & Song, 2024; Finzi et al., 2023; Boys et al., 2023; Wu et al., 2023; Peng et al., 2024) provide different levels of approximations of the true conditional distribution. Here, we show an approach particularly similar to (Chung et al., 2023), by approximating the true denoising distribution $p(\mathbf{X}_0|\mathbf{X}_t,\mathbf{Y})$ directly with the denoiser output

$$p(y\,|\,\mathbf{X}_t,\mathbf{Y}) = \sum_{\mathbf{X}_0} p(\mathbf{X}_0\,|\,\mathbf{X}_t,\mathbf{Y})p(y\,|\,\mathbf{X}_0) \tag{111}$$

$$\approx \sum_{\mathbf{X}_0} \tilde{p}_\theta(\mathbf{X}_0\,|\,\mathbf{X}_t,\mathbf{Y})p(y\,|\,\mathbf{X}_0). \tag{112}$$

Here $p(\mathbf{X}_0\,|\,\mathbf{X}_t,\mathbf{Y})$ is the true, intractable distribution given by going through the entire sampling process and $\tilde{p}(\mathbf{X}_0\,|\,\mathbf{X}_t,\mathbf{Y})$ is the factorized distribution that is given by the denoiser output, and 'jumps to' $X_0$ directly. This results in the following update step:

$$\log\mathbf{P}_\theta(\mathbf{X}_{t-1}\,|\,\mathbf{X}_t,\mathbf{Y},y) \propto \nabla_{\mathbf{X}_t}\log\left(\mathbb{E}_{\tilde{p}_\theta(\mathbf{X}_0\,|\,\mathbf{X}_t,\mathbf{Y})}p(y\,|\,\mathbf{X}_0)\right)\mathbf{X}_{t-1} + \log\mathbf{P}_\theta(\mathbf{X}_{t-1}\,|\,\mathbf{X}_t,\mathbf{Y}). \tag{113}$$

Summing over all possible graphs $\mathbf{X}_0$ is prohibitive, however. Instead, we could sample $\mathbf{X}_0$ from $\tilde{p}_\theta(\mathbf{X}_0\,|\,\mathbf{X}_t,\mathbf{Y})$ with the Gumbel-Softmax trick (Jang et al., 2016) and evaluate $\log p(y\,|\,\mathbf{X}_0)$. As long as $\log p(y\,|\,\mathbf{X}_0)$ is differentiable, we can then just use automatic differentiation to get our estimate of $\nabla_{\mathbf{X}_t}\log\mathbf{P}_\theta(y\,|\,\mathbf{X}_t,\mathbf{Y})$. Another, more simplified approach that avoids sampling from $\tilde{p}_\theta(\mathbf{X}_0\,|\,\mathbf{X}_t,\mathbf{Y})$ is to relax the definition of the likelihood function to directly condition on the continuous-valued probability vector $\tilde{\mathbf{P}}_\theta(\mathbf{X}_0\,|\,\mathbf{X}_t,\mathbf{Y})$ instead of the discrete-valued $\mathbf{X}_0$. For simplicity, we adopted this approach, but the full method with Gumbel-Softmax is not significantly more difficult to implement. The algorithm for sampling is shown in Alg. 3.

**Toy Synthesisability Model: Controlling Atom Economy**

The model of synthesizability that we use is

$$\tilde{\mathbf{X}}_0 = p_\theta(\mathbf{X}_0\,|\,\mathbf{X}_t,\mathbf{Y}), \tag{114}$$

$$p(y = \text{synthesizable}\,|\,\tilde{\mathbf{X}}_0) = \sigma\left(\frac{\sum_{i\in S}\tilde{\mathbf{X}}_{0,i,d} - a}{b}\right)^\gamma, \tag{115}$$

where $S$ is the set of non-atom-mapped nodes, $a$, $b$ and $\gamma$ are constants that define the synthesizability model and $d$ refers to the dummy node index. The intuition is that the more nodes are classified as dummy nodes (non-atoms), the fewer atoms we have in total, leaving the atom economy higher. Note that $\sum_{i\in S}\tilde{\mathbf{X}}_{0,i,d}$ is the expected amount of dummy nodes from $p_\theta(\mathbf{X}_0\,|\,\mathbf{X}_t,\mathbf{Y})$. We set $a$

---

**Algorithm 3** Sampling with atom-count guidance

---

**Input:** Product $\mathbf{Y}$
**Choose:** $P^{\mathbf{Y} \to \mathbf{X}} \in \mathbb{R}^{N_{\mathbf{X}} \times N_{\mathbf{Y}}}$
$\mathbf{X}_T \propto p(\mathbf{X}_T)$
**for** $t = T$ **to** $1$ **do**
  $\tilde{\mathbf{X}}_0 = D_\theta(\mathbf{X}_t, \mathbf{Y}, P^{\mathbf{Y} \to \mathbf{X}})$                                    ▷ *Denoising output*
  $\tilde{\mathbf{X}}_{t-1}^i = \sum_k q(\mathbf{X}_{t-1}^i \mid \mathbf{X}_t^i, \mathbf{X}_0^i)\tilde{\mathbf{X}}_0^i$                  ▷ *Regular reverse transition probabilities*
  $\mathbf{X}_{t-1}^i \sim \text{softmax}(\log \tilde{\mathbf{X}}_{t-1} + \gamma \nabla_{\mathbf{X}_t} \log \sigma(\frac{\sum_{i \in S} \tilde{\mathbf{X}}_{0,i,d} - a}{b}))$         ▷ *Renormalize*
**Return** $\mathbf{X}_0$

---

to half the amount of dummy nodes and $b$ to one-quarter of the amount of dummy nodes. It turns out that this leaves $\gamma$ as a useful parameter to tune the sharpness of the conditioning. The gradient estimate is then given by

$$\nabla_{\mathbf{X}_t} \log \mathbf{P}_\theta(y \mid \mathbf{X}_t, \mathbf{Y}) = \gamma \nabla_{\mathbf{X}_t} \log \sigma(\frac{\sum_{i \in S} \tilde{\mathbf{X}}_{0,i,d} - a}{b}), \tag{116}$$

which can be directly calculated with automatic differentiation.

## G    HANDLING NOISY NODE MAPPINGS FOR THE TOY DATA

Sec. 4.1 introduced the graph copying as a toy data set. Here, we extend it to analyse the effect of imperfect node mapping during training. In particular, we consider a scheme for adding 'noise' to the node mapping by swapping them with each other on either the conditioning, or equivalently, the other side. We do it as follows:

1. Sample the number of pair swaps: $f \sim \text{Poisson}(\lambda)$

2. For each of the $f$ swaps, randomly select two nodes $i, j$ in the graph $A$ and swap their mappings

The Poisson parameter $\lambda$ controls the expected number of swaps, allowing us to tune the level of noise in the node mapping. We trained small models for a 1000 steps of training, and show the results with different noise levels in Fig. A13. Small errors in the node mapping are evidently not a significant issue, but larger $\lambda$ does start affecting performance, at least in this early training phase. We also include a measure of similarity to the target graph: The precision is the fraction of edges that were correctly inferred, compared to the ground-truth target.

## H    DETAILS FOR THE IBUPROFEN SYNTHESIS EXPERIMENT

Below we explain the synthesis steps visualized in Fig. 7 in detail:

1. The retrosynthesis begins with carbonylation, adding a 'CO' structure. Initial basic generation yields unpromising results, so the practitioner suggests a partial reactant structure, leading to a more viable path.

2. The model then proposes hydrogenation, a logical next step. While H2 molecules aren't explicitly represented in the data, they're inferred from the context of the reaction.

3. For the third step, the practitioner identifies an opportunity for acylation (involving the `C(=O)C` group), potentially leading to readily available reactants. Given this `C(=O)C` structure, the model successfully completes the reaction. These steps align with the synthesis plan proposed by BHC.

4. The retrosynthesis path starts with carbonylation, a well-known synthetic reaction which adds a 'CO' structure to a compound. The practitioner tries basic generation but then notices that the suggested reactant is not promising. They then suggest a partial structure of the reactants which leads to a more sensible path.

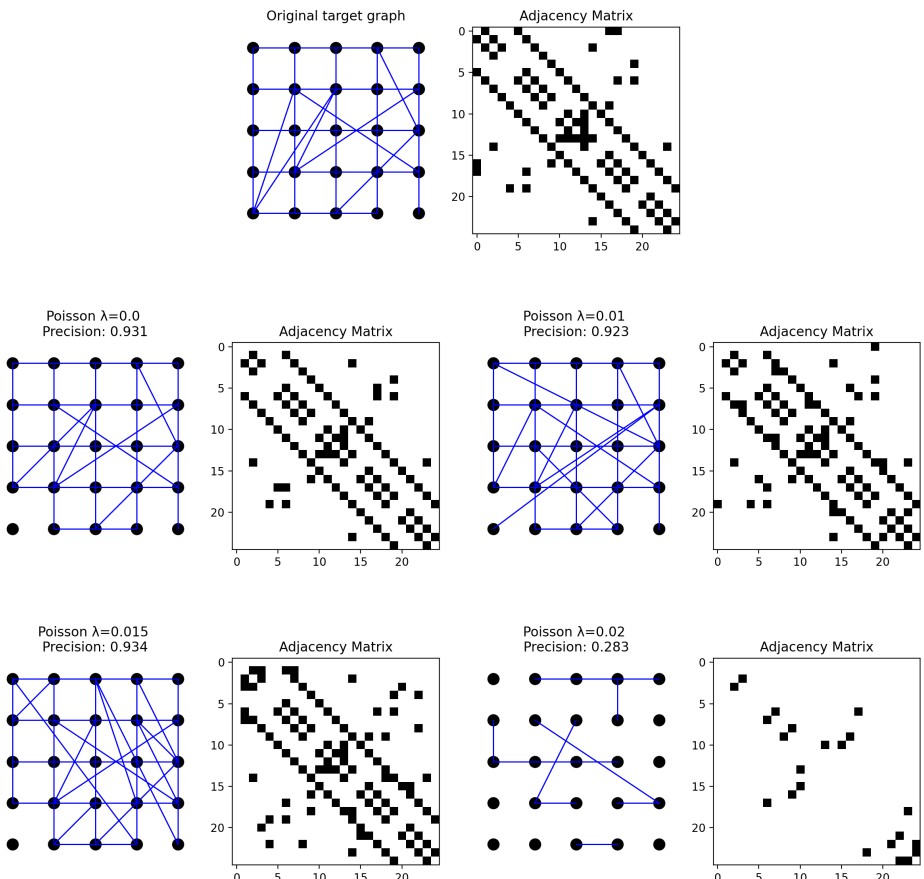

Figure A13: Results from training an aligned model for a 1000 steps in the graph copying task presented in Sec. 4.1, with different levels of noise in the atom mappings. $\lambda$ parameterizes a Poisson distribution that defines the level of mistakes in the node mapping. On lower noise levels, the results do not significantly change. However, there seems to be a limit in which the neural net training dynamics changes to produce suboptimal results.

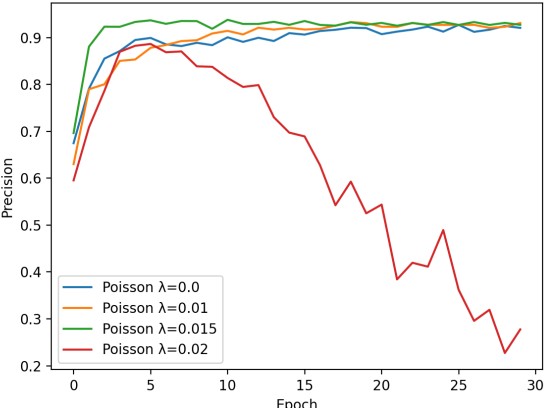

Figure A14: The precision of the adjacency matrix estimates from an aligned model across training, with different levels of node mapping noise $\lambda$. Going above $\lambda > 0.02$ nudges the training dynamics to perform poorly on this task.

5. Next, our model proposes hydrogenation, which is a sensible suggestion in this case. The data does not handle explicitly individual `H2` molecules, but they are inferred from the context.

6. In the third step, the practitioner notices that an acylation reaction (a reaction with the `C(=O)C` group) might lead to reactants that are readily available. The model is able to complete the rest of the reaction after knowing that C(=O)C is present. These steps match the synthesis plan proposed by BHC.

