# OpenReview forum: "Equivariant Denoisers Cannot Copy Graphs: Align Your Graph Diffusion Models"
_ICLR.cc/2025/Conference — ICLR 2025 Poster_

### Official Review · Reviewer_HVWS · 2024-11-03

**Soundness:** 3
**Presentation:** 3
**Contribution:** 2
**Rating:** 6
**Confidence:** 4

**Summary:**

This paper proposes an aligned permutation equivariant denoiser for a discrete diffusion model to address the graph translation problem. To align the initial state and target, several techniques, including nodal positional encoding, skip connections, and input alignment, are introduced. The model was validated on retrosynthesis, guided generation, and inpainting tasks.

**Strengths:**

Aligning initial and target to generate efficiently makes sense and the results are promising compared to unaligned version.

**Weaknesses:**

Since alignment itself is an empirical approach that has been used previously, the contribution may be limited to explaining why alignment is necessary specifically in a discrete diffusion model with an absorbing state as the prior. When compared to RetroBridge in the retrosynthesis task, it is difficult to find a significant performance gap between the proposed model and RetroBridge. In the cases of guided diffusion and inpainting, only a few figures are provided without numerical comparisons, making precise validation difficult.

**Questions:**

1. The paper assumes that the prior distribution is absorbing state. Does the proposed model still effective even on the uniform distribution prior?

2. What happens if the node-mapping matrix has uncertainty and is not a binary matrix?

3. How does the performance compare when using only direct skip connections without positional encoding?

4. How does the performance differ from graph translation methods in Related Works, such as G2gt?

---

> ### Author Response · Authors · 2024-11-21
> **Response to reviewer HVWS (1/3)**
>
> We thank the reviewer for the effort put in reading our manuscript and suggesting points for improvements. We would like to start highlighting the following points regarding the raised weaknesses (elaborated in more detail later):
> - Our **Theorem 1 is applicable to all standard discrete diffusion priors**, including uniform and marginal (from Digress [1]). Further, it is not difficult to extend it to Gaussian diffusion models. We argue that the theoretical understanding is important, because it enabled us to propose new types of alignment that lead to improved results.
> - **We achieve significant improvements over Retrobridge**. Retrobridge used a non-standard evaluation scheme, and the difference becomes larger with when adopting RetroBridge's evaluation scheme. Furthermore, our model achieves much better results with much less diffusion steps.
> - **We have now added numerical results for inpainting**, showing that side information from a chemist interacting with the system can improve the accuracy significantly.
>
> **Detailed answers:**
>
> **Alignment has been used previously..., Is the contribution limited to "why alignment is necessary for a discrete diffusion model with an absorbing-state prior"**
>
> As noted in Section 2, alignment has been used in a diffusion-like approach previously in the code of Retrobridge, which we noticed while preparing the first draft of our paper. This design choice was neither analysed nor explicitly discussed in the paper. In contrast, a main contribution of our work is the theoretical understanding of the problem and formalizing the solution through aligned equivariance. **Understanding the core of the issue through the theory enabled us to propose new types of alignment methods that result in significant improvements over Retrobridge** (see comparisons in next section). This showcases how the formalization of aligned equivariance is a useful conceptual tool in developing new models. Aside from Retrobridge, we are not aware of work that overlaps with our discussion of alignment for graph diffusion.
>
> Further, our Theorem 1 **applies to all standard discrete diffusion priors**, not just the absorbing state diffusion. It would not be difficult to extend it to Gaussian diffusion models [1] as well or even flow matching [2], since the core insight is that the nodes in the noisy graph $X_T$ are in a symmetric position, and the equivariant denoiser is unable to break these symmetries.
>
> **"When compared to RetroBridge in the retrosynthesis task, it is difficult to find a significant performance gap between the proposed model and RetroBridge."**
>
> Although this was not as apparent from the initial submission, we do achieve a significant improvement over Retrobridge when following their evaluation scheme. Retrobridge was evaluated in a non-standard manner, where charges and stereochemistry were not taken into account when calculating the top-k scores, inflating the results and making a direct comparison difficult. We trained another model using the same training data as Retrobridge (not including charges), and conducted the evaluation in the same way. The results are in the following table:
>
> |Model|k=1|k=3|k=5|k=10|MRR|
> |-----|---|---|---|----|---|
> |RetroBridge (T=500)|50.8|74.1|**80.6**|**85.6**|0.622|
> |Ours-uncharged (T=100)|**56.3**|**75.2**|80.4|84.0|**0.658**|
>
> Many recently published methods for retrosynthesis have capped out their top-1 scores in the 49%-53% range (see our Table 2), indicating that an increase of over 5% is quite significant. Furthermore, we achieve this with 5 times less denoising steps (100 instead of 500). For a closer look at the performance at different step counts, see also their Figure 5 compared to our Figure 5b, **showing that Retrobridge top-1 score decreases to almost 0 at a step count of more than 10. In contrast, our top-1 is close to 50% even at 5 steps, and higher than Retrobridge at 100 steps**. We attribute this improvement to our more careful analysis of alignment, and utilising multiple alignment techniques.

---

> ### Author Response · Authors · 2024-11-21
> **Response to reviewer (2/3)**
>
> **Numerical comparisons on inpainting.**
> In order to quantify the benefits of inpainting, we conducted the following experiment: fix one of the reactants in multi-molecule reactant sets, and generate the other reactants based on the fixed reactant+product. This scenario corresponds to a chemist selecting one precursor (based on availability for example) and wanting to explore synthesis routes incorporating their choice. **Below we report our results, which we also added to Appendix E**.
>
> |Model|k=1|k=3|k=5|k=10|MRR|
> |-----|---|---|---|----|---|
> |DiffAlign-1reactant-inpainting (T=10)|**68.8**|**82.56**|**84.74**|**86.82**|**0.753**|
>
> Our best model reached a top-1 accuracy of 68.88 and an MRR of 0.753 with just 10 diffusion steps, showcasing the potential for high accuracy and speed in an interactive setup. **Also note that we are able to achieve these results without a need to retrain a model specifically on this task, which most of the baselines we compare to are not able to do at all.**
>
> **Questions**
>
> **The paper assumes that the prior distribution is absorbing state. Does the proposed model still effective even on the uniform distribution prior?**
> Yes, our model is effective with other priors. In fact, changing the prior distribution (or transition matrix), while an important design choice for diffusion models, is irrelevant to our study of the limitations of equivariant denoisers. In our early experiments, we tested the uniform, marginal, and absorbing state transitions, and noticed the marginal and absorbing state models to perform similarly while uniform was slightly worse. The absorbing state model is somewhat simpler and is the preferred choice in most modern discrete diffusion literature [3,4,5], motivating our choice. **We have now made this point more clear at the end of the diffusion formulation section.**
>
> **What happens if the node-mapping matrix has uncertainty and is not a binary matrix?**
>
> Our method requires node-mapping information to align the source and target graphs. While a complete and correct mapping matrix is ideal, **the method is robust to noisy mappings** up to a limit. Therefore, one way would be to convert a nonbinary mapping matrix to a binary one via argmax (or potentially even sampling) works for our purposes. To follow up on this, we performed a small toy experiment on the graph copying task and noisy node mappings, and added the results in Appendix G. For low levels of noise, results remain the same, but for a higher node mapping noise level, the performance can drop.
>
> **How does the performance compare when using only direct skip connections without positional encoding?**
> We thank the reviewer for an interesting suggestion. We are currently training such a model and will try to get the results for the rebuttal period. Likely the results will not be as good since the skip connection does not add a lot of expressivity aside from the ability to copy graphs effectively. The positional encodings, on the other hand, are given as input to the neural network and as such may allow for more expressivity.
>
> **How does the performance differ from graph translation methods in Related Works, such as G2gt?**
>
> We list our top-k scores and mean reciprocal rank compared to the G2G [6] graph translation based retrosynthesis method below:
>
> | Method | k = 1 | k = 3 | k = 5 | k = 10 | MRR |
> |--------|-------|-------|-------|--------|-----|
> | G2G | 48.9 | 67.6 | 72.5 | 75.5 | 0.582 |
> | DiffAlign-PE+skip | **54.7** | **73.3** | **77.8** | **81.1** | **0.639** |
>
> We outperform G2G by a clear margin across all top-k metrics. We have now also moved a full comparison to other methods as Table 2 in the main text. Other aspects to keep in mind when comparing against G2G is that
> 1) It is particularly designed for retrosynthesis, with a multi-stage procedure for identifying reaction centers, breaking the product into reactants and filling out missing parts with an autoregressive model
> 2) This approach inherently limits the model to chemical reactions where a reaction center exists. We added a figure Figure A11 in Appendix D demonstrating a type of reaction which cannot be handled by methods like G2G, but for which our model returns correct predictions. This also applies to other previous work such as [7] and [8].

---

> ### Author Response · Authors · 2024-11-21
> **Response to reviewer (3/3)**
>
> **References**
>
> [1] Niu et al, "Permutation invariant graph generation via score-based generative modeling", AISTATS 2020
>
> [2] Eijkelboom et al., "Variational Flow Matching for Graph Generation", NeurIPS 2024
>
> [3] Lou et al., "Discrete Diffusion Modeling by Estimating the Ratios of the Data Distribution", ICML 2024
>
> [4] Sahoo et al., "Simple and Effective Masked Diffusion Language Models", NeurIPS 2024
>
> [5] Shi et al., "Simplified and Generalized Masked Diffusion for Discrete Data", NeurIPS 2024
>
> [6] Shi et al., "A Graph to Graphs Framework for Retrosynthesis Prediction", ICML 2020
>
> [7] Somnath et al., "Learning Graph Models for Retrosynthesis Prediction", NeurIPS 2021
>
> [8] Liu et al. "MARS: a motif-based autoregressive model for retrosynthesis prediction." Bioinformatics, 2024

---

> ### Author Response · Authors · 2024-12-01
> **Discussion period ending soon**
>
> Thank you once again for taking time to review our paper and provide valuable suggestions. This is a kind reminder that tomorrow is the last day reviewers can post replies on openreview. We would love to know if our answers so far have addressed your concerns and if there is anything else we can do to clarify matters further.

---

> > ### Author Response · Authors · 2024-12-01
> > **Recap on new ablations relevant to the questions**
> >
> > Additionally, we would like to raise here the new results we have prepared during the rebuttal that also directly answer questions 1 and 3 with experiments (questions 2 and 4 had experimental results in the previous answers). These are also written in the common response.
> >
> > **The ablation on transition matrices** shows that while the uniform transitions perform the least well and the absorbing transitions perform the best, all of them can work with a reasonable accuracy.
> >
> > |Model|Top-1|top-3|top-5|top-10|top-50|MRR|
> > |-|-|-|-|-|-|-|
> > |Uniform-PE+Skip|43.75%|58.98%|62.89%|67.38%|72.66%|0.5156
> > |Marginal-PE+Skip|44.73%|60.94%|63.28%|66.60%|69.92%|0.5244
> > |Absorbing-PE+Skip|**49.22%**|**70.90%**|**74.61%**|**77.93%**|**84.96%**|**0.5952**
> >
> > **The ablation on using only the skip connection alignment method.** Below, we see that the model with only the skip connections as alignment does not perform well, although it is better than the entirely unaligned model (4% top-1 in our Table 1). This is likely due to limited expressivity: the neural network does not get the alignment information at all as input in this case.
> >
> > |Model|Top-1|top-3|top-5|top-10|top-50|MRR|
> > |-|-|-|-|-|-|-|
> > |DiffAlign-Skip|9.77%|17.38%|24.61%|29.30%|32.23%|0.1517|
> > |DiffAlign-PE+Skip (Laplacian)|**49.22%**|**70.90%**|**74.61%**|**77.93%**|**84.96%**|**0.5952**|

---

> > > ### Comment · Reviewer_HVWS · 2024-12-01
> > >
> > > Thank you for the further experiments and explanation. I updated the score accordingly.

---

### Official Review · Reviewer_gPTC · 2024-11-03

**Soundness:** 3
**Presentation:** 2
**Contribution:** 3
**Rating:** 6
**Confidence:** 3

**Summary:**

The paper presents a novel method to perform graph-to-graph translation using graph diffusion models. The authors show why equivariant models are inherently limited in their ability to perform this task when nodes from the prior distribution need to be aligned with nodes from the output. To that purpose, they demonstrate that an ideal equivariant denoiser cannot correctly copy a graph. Then, they propose several ways to break equivariance for the nodes that should be aligned between the input and the output, and empirically validate their method on the retrosynthesis task.

**Strengths:**

- To the best of my knowledge, the topic of breaking symmetries to perform graph-to-graph translation has not yet been addressed in the graph diffusion literature.
- The problem of using equivariant models is clearly identified and backed by theoritical evidences.
- The paper proposed several simple ways to solve this problem.

**Weaknesses:**

- Why using the aborbing state distribution and not the marginal distributions introduced in DiGress (Vignac et al., 2022) ?

- I would avoid statements like "it is easy to see" l304 or "Clearly" l242 and provide instead clear motivations and explanations

- Some claims or design choices lack motivation or justification :
- For example, the claims in section 3.2 "a model that is capable of copying graphs from one side of the reaction
to the other should also be extendable to modulations of this task". Do you mean that it's a prerequisite or that is implies that the model has the ability to peform "the modoluations". Overall, I fell that this subsection conveys very little information.
- The proposed aligning methods are insufficiently motivated.

**Questions:**

- What is the difference between $D$ and $\tilde p_{\theta}$ ?
- How is $\psi$ generated in section 3.5 ?
- How is Y passed to the denoiser ? I feel it's not clearly outlined anywhere except in the section "Aligning Y in the input"
- I don't get how you build the dataset for retrosynthesis. If it is standard benchmark, could you please provide more explanations or point out to a reference

---

> ### Author Response · Authors · 2024-11-21
> **Response to reviewer gPTC (1/2)**
>
> We thank the reviewer for the insightful questions. Answers in order:
>
> **Why using the aborbing state distribution and not the marginal distributions introduced in DiGress (Vignac et al., 2022) ?** While the transition matrix is an important design choice for the performance of diffusion models, this choice is less relevant when studying the properties of an equivariant denoiser. The absorbing state model [1] is somewhat simpler than alternatives, and has been empirically found to outperform other transitions [1,2,3,4], making it a good testing ground for our theory. In our early experiments, we tested the uniform, marginal, and absorbing state transitions and did not notice substantial difference in performance. The absorbing state transition gave marginally better results which motivated our final choice. We will make this point clearer in the text.
>
> **Redundant statements like clearly, easy to see, etc** We thank the reviewer for pointing this out. This prompted us to go through our manuscript to streamline the language and replace all occurrences of these words with more precise descriptions. We also went through many similarly redundant phrases, like "should be possible".
>
> **Do you mean that it's a prerequisite or that is implies that the model has the ability to peform "the modulations".** We agree this sentence is confusing. What we meant to say is that for many graph translation tasks, such as retrosynthesis, the optimal denoising function is close to the identity denoising function, motivating the choice of copying as a toy model. We have changed the text accordingly.
>
> **Overall, I felt that this subsection [3.2] conveys very little information.** We agree that the writing should be improved here, and have merged Section 3.2 with Section 3.3 and removed superfluous parts. The point in the text of Section 3.2 was to: 1) present the graph copying task as a theoretical tool to study the limitations of equivariant models, and 2) explain that graph copying is a real bottleneck in graph translation.
>
> **The proposed aligning methods are insufficiently motivated.** We agree that a clearer exposition of the underlying motivations would strengthen the paper, and **we have expanded this section in the revision.** Our choice of alignment methods follows a principled progression, building upon established concepts in deep learning and graph neural networks:
> - **Graph positional encodings** are a standard tool to break permutation symmetry in GNNs, prompting us to examine their use in inducing aligned permutation equivariance. This turns out to be possible using node-mapped positional encodings. This enables leveraging all the existing graph positional encoding methods for graph translation.
> - **The 'skip connection'** denoiser is motivated by two insights: (a) its ability to represent identity data minimally through a single parameter λ, and (b) the well-documented success of residual connections across deep learning applications, suggesting their potential effectiveness in our domain.
> - **The input alignment** is a natural next step from the 'skip connection' alignment. Instead of aligning the graphs at the output, what if we align them at the input? This way, the neural net representations corresponding to the target graph edges and nodes are directly given all of the information about the corresponding conditioning edges and nodes at the very beginning of the neural network.
>
> It is likely that many other methods exist, and we believe that proposing some of the simplest approaches is a good starting point for future research to develop even more sophisticated methods.

---

> ### Author Response · Authors · 2024-11-21
> **Response to reviewer gPTC (2/2)**
>
> **Questions**
>
> **Difference between $D_{\theta}$ and $\tilde{p}_{\theta}$:** $D_{\theta}$ is the output of the neural network, which consists of vectors of probabilities corresponding to each component in the graph (node or edge). $D_{\theta}(X) = (D_{\theta}(X)^N, D_{\theta}(X)^E)$ is a tuple of matrices. $\tilde{p}_{\theta}$ is the probability of a single graph $X_0$ under our model, which is parameterized by the probabilities $D_{\theta}$. Since the probability of the graph factorizes over the nodes and edges independently, we can formally write the connection between $D_{\theta}$ and $\tilde{p}_{\theta}$ for a single graph as:
> $\tilde{p}_{\theta}(X_0|X_t) = \prod_i (D_{\theta}(X_t)_i^N)^T \cdot X_{0,i}^N * \prod_{i,j} (D_{\theta}(X_t)_{ij}^E)^T \cdot X^E_{0,i,j}$
> where $D_{\theta}(X_t)_i^N$ is the probability vector for a single node $i$ (resp. for edge $(i,j)$), $X^N_{0,i}$ is a onehot-encoded vector representing the ground truth label for node $i$ (resp. for $X^E_{0,i,j}$ and edge $(i,j)$), and we denote the transpose of a matrix $M$ and $M^T$. **We have included this clarification, and an explanation regarding the output space of $D_\theta$ (as suggested by reviewer Gt4F) in our main text when introducing $D_{\theta}$.**
>
> **How is $\varphi$ generated?** $\varphi$, $\varphi$ can be any vector of numbers as long at the vectors are distinct between pairs of mapped nodes, per the description in Section 3.5 paragraph "Node-mapped positional encodings". For our experiments, we specifically use Laplacian eigenvector-based positional encodings [5]. These vectors are obtained by taking the eigendecomposition of the graph Laplacian matrix, and using these eigenvector values as node features. **We have included this information in the section "Methods for Alignment in GNNs".**
>
> **How is Y passed to the denoiser? I feel it's not clearly outlined anywhere except in the section "Aligning Y in the input"** Thank you for pointing this out. We pass the node feature matrix $Y^N$ by concatenating $Y^N$ to $X^N$ along the node dimension, creating a node matrix of size $(N_X+N_Y, K_a)$ ($K_a$ is the node feature dimension). Correspondingly, we form an edge tensor of size $(N_X+N_Y,N_X+N_Y,K_b)$, where the edges between the conditioning graph and target graph are zero, and the edges within the conditioning and target graphs are the original $Y^E$ and $X^E$. We considered this to be the simplest way to include the conditioning graph information without breaking permutation equivariance. **We have now explained this in Section 5, "Permutation equivariant denoisers cannot learn the identity function".**
>
> **I don't get how you build the dataset for retrosynthesis. If it is standard benchmark, could you please provide more explanations or point out to a reference** We use the standard USPTO-50k benchmark [8]. We added the following description to our "Experimental Setup" paragraph:
>
> "We use the benchmark dataset USPTO-50k for our experiments. The dataset consists of 50000 chemical reactions, in SMILES format [7], carefully curated by [3] from an original 2 million reactions extracted through text mining by [6]. More information on the benchmark dataset USPTO and its various subsets can be found in \cref{app:uspto-data}.""
>
> **References:**
>
> [1] Austin et al., "Structured Denoising Diffusion Models in Discrete State-Spaces", NeurIPS 2021
>
> [2] Lou et al., "Discrete Diffusion Modeling by Estimating the Ratios of the Data Distribution", ICML 2024
>
> [3] Sahoo et al., "Simple and Effective Masked Diffusion Language Models", NeurIPS 2024
>
> [4] Shi et al., "Simplified and Generalized Masked Diffusion for Discrete Data", NeurIPS 2024
>
> [5] Dwivedi et al., "Benchmarking Graph Neural Networks", JMLR
>
> [6] Lowe et al., "Extraction of Chemical Structures and Reactions from the Literature"
>
> [7] Weininger, David, "SMILES, a chemical language and information system."
>
> [8] Schneider et al., "What’s what: The (nearly) definitive guide to reaction role assignment"

---

> ### Comment · Reviewer_gPTC · 2024-11-26
> **Answer to rebuttal**
>
> Thanks for writting a detailed rebuttal.
>
> There are two points that I consider require further clarification :
> - Absorbing transitions perform better than marginal : even though it is true that most discrete diffusion models employ absorbing transitions, most *graph* discrete diffusion models employ marginal transitions successfully. It does not surprise me that absorbing transitions work well, but I'd like to see an ablation if you have the experimental results at hand. It's not a big deal, but it would make your work more complete.
>
> - Concerning Section 8 : it looks a lot like the DiGress' conditional generation method. In what way is it novel ?

---

> > ### Author Response · Authors · 2024-11-28
> > **Results on different transitions and clarification on the conditional generation method novelty**
> >
> > Thank you for the chance to provide additional clarification. The discussion has really helped improve the overall quality of the work, on both theoretical exposition as well as empirical consolidation. We list our answers below:
> >
> > ### Results for ablation on absorbing state
> >
> > Here, we provide initial results on an ablation on the noise type. Note that our marginal noise model is still training, so for a fair (and quick) comparison we evaluated this model and our best model DiffAlign-PE+Skip at roughly 10% of the full training budget (80 epochs), with 512 samples drawn from the test set. The table below includes our results:
> >
> > |Model|Top-1|top-3|top-5|top-10|top-50|MRR|
> > |-|-|-|-|-|-|-|
> > |Uniform-PE+Skip|43.75%|58.98%|62.89%|67.38%|72.66%|0.5156
> > |Marginal-PE+Skip|44.73%|60.94%|63.28%|66.60%|69.92%|0.5244
> > |Absorbing-PE+Skip|**49.22%**|**70.90%**|**74.61%**|**77.93%**|**84.96%**|**0.5952**
> >
> > We also added these results to our latest manuscript, Appendix E. Upon the request of reviewer Gt4F, we conducted additional ablations on the positional encoding/alignment methods. You can see the results in the general comment ([link](https://openreview.net/forum?id=onIro14tHv&noteId=OFlcEufT6K)).
> >
> >
> > ### Section 8 & similarity to DiGress guidance
> > The key difference is that **DiGress adapts classifier guidance to discrete diffusion, which requires training, while our method adapts so-called 'reconstruction guidance', which does not require training.** Reconstruction guidance was introduced concurrently in [1,2,3]. In reconstruction guidance, we only need access to a user-defined likelihood for *clean data*, $p(y|X_0)$. We then modify the sampling process such that the final generated $X_0$ has a high likelihood according to the model. If $p(y|X_0)$ is given, this does not require training, and is a purely inference-time procedure. In contrast, classifier guidance [5,6] requires training a noise-conditional classifier $p(y|X_t)$. **Thus, the benefit is more flexibility and the ability to incorporate the exact form of the constraint $p(y|X_0)$ in the method in case it is available.**
> >
> > **The way it works in continuous models:** Because the correct conditioning formula $\nabla_{X_t}\log p(X_t|Y) = \nabla_{X_t}\log p(Y|X_t) + \nabla_{X_t}\log p(X_t)$ references $p(Y|X_t)$ instead of $p(Y|X_0)$, we need to approximate
> >
> > $$\nabla_{X_t}\log p(y|X_t) = \nabla_{X_t}\log\int p(y|X_0)p(X_0|X_t)dX_0 \approx \nabla_{X_t}\log p(y|E[X_0|X_t]) \approx \nabla_{X_t}\log p(y|D_\theta(X_t))$$
> > where $D_\theta(X_t)$ is the neural net denoiser that outputs an $X_0$ estimate. This guidance term is then used to nudge the generative process such that the final generated sample $X_0$ matches better with the likelihoods $p(y|X_0)$.
> >
> > **We adapt this idea to discrete diffusion.** To do this, we start with similar mathematics as DiGress [4] (and also [5,6]) does for classifier guidance. We then continue to derive the connection to reconstruction guidance, which is novel. We have modified our text (Section 8 and Appendix F) to make this distinction explicit, with changes highlighted. We also rewrote some parts to clarify further how the method works in practice.

---

> ### Author Response · Authors · 2024-11-28
> **more details on conditional generation novelty**
>
> **Reiterating the mathematics from Appendix E.** Below we reiterate the similarities and differences between the two methods for completeness. This is included in Appendix E. We use our own notation throughout but point to the equivalent results in DiGress where appropriate. A quick reminder that $X_{t}$ is the noisy input graph at step $t$, $Y$ is the condition graph, and $y$ is the property we are using to guide generation.
>
> $\newcommand{\X}{\mathbf{X}}
> \newcommand{\Y}{\mathbf{Y}}
> \newcommand{\P}{\mathbf{P}}$
>
> **The similar part**: We start by using Bayes' rule:
> $$p_\theta(\X_{t-1}| \X_t, \Y, y) \propto p_\theta(\X_{t-1}| \X_t, \Y) p(y | \X_{t-1}, \X_t, \Y) $$$$ = p_\theta(\X_{t-1} | \X_t, \Y) p(y \mid \X_{t-1}, \Y)$$
> where the second equality is due to $\X_t$ and $\X_{t-1}$ being independent (Markov property). By interpreting the probabilities as tensors in the same space as $\X_{t-1}$ and $\X_{t}$, taking the log and applying a first-order Taylor expansion of $p(y \mid \X_{t-1}, \Y)$ around $\X_t$, we get:
> $$\log \P_\theta(\X_{t-1}\mid\X_t, \Y, y) \propto \log\P_\theta(\X_{t-1}\mid\X_t, \Y) + \log \P(y \mid\X_{t-1}, \Y) $$$$ \approx \log\P_\theta(\X_{t-1}\mid\X_t, \Y) + \log \P(y \mid\X_{t}, \Y)+ \nabla_{\X_t} \log \P(y \mid\X_{t}, \Y)(\X_{t-1} - \X_t)$$
> In DiGress, this result is given as the first equation after Lemma 5.1, while you can see it in our work in Appendix E eq 103 to 108. From here on, DiGress uses classifier guidance to approximate $\P(y \mid\X_{t}, \Y)$, while we further derive this to a reconstruction guidance-like method to incorporate arbitrary guiding signals at inference time.
>
> **Classifier guidance (Digress)**: DiGress assumes $\P(y \mid\X_{t}, \Y) = \mathcal{N}(g_{n}(\X_t), \sigma_y I)$, where $g_n$ is a regressor estimating $y$ for a noisy graph $\X_{t}$. The gradients $\nabla_{\X_t'} \log \P(y \mid\X_{t}', \Y)|_{\X_t'=\X_t}$ can thus be computed using automatic differentation and the assumption of normality means we can replace the log with a squared difference (see Algorithm 3 in DiGress).
>
> **Reconstruction guidance (Us)** The idea here is that we can compute $p(y\mid\X_t, \Y)$ directly if we have access to a property predictor $p(y\mid\X_0)$ from a clean graph.
> $$p(y\mid\X_t, \Y) = \sum_{\X_0} p(\X_0\mid\X_t, \Y) p(y\mid\X_0)$$$$\approx \sum_{\X_0} \tilde p_\theta(\X_0\mid\X_t, \Y) p(y\mid\X_0),$$
> Here $p(\X_0\mid\X_t, \Y)$ is the true, intractable distribution given by going through the entire sampling process and $\tilde p(\X_0\mid\X_t, \Y)$ is the factorized distribution that is given by the denoiser output, and 'jumps to' $X_0$ directly. This leads to the following update step:
>
> $$\log \P_\theta(\X_{t-1}\mid\X_t, \Y, y) \propto \nabla_{\X_t} \log \left(\sum_{\X_0} \tilde p_\theta(\X_0\mid\X_t, \Y)p(y\mid\X_0)\right)\X_{t-1} + \log\P_\theta(\X_{t-1}\mid\X_t, \Y).$$
>
> where we can bypass the sum over all possible $\X_0$ (i.e.,in the expectation) by sampling from $\tilde p_\theta(\X_0\mid\X_t, \Y)$ using the Gumbel-softmax trick, or by relaxing the definition of the likelihood to take a continuous vector $\tilde p_\theta(\X_0\mid\X_t, \Y)$ instead of a discrete value $\X_0$.
>
> We thank again the help in improving the paper, and are open to further discussion.
>
> **References**
>
> [1] Chung et al. "Diffusion Posterior Sampling for General Noisy Inverse Problems." ICLR 2023
>
> [2] Ho et al. "Video diffusion models." NeurIPS 2022
>
> [3] Song et al. "Pseudoinverse-guided diffusion models for inverse problems." ICLR 2023
>
> [4] Vignac, Clement, et al. "DiGress: Discrete Denoising diffusion for graph generation." ICLR 2023
>
> [5] Sohl-Dickstein et al, "Deep unsupervised learning using nonequilibrium thermodynamics", ICML 2015
>
> [6] Dhariwal and Nichol, "Diffusion Models Beat GANs on Image Synthesis", NeurIPS 2021

---

> ### Author Response · Authors · 2024-12-01
> **Discussion period ending soon**
>
> Thank you once again for your detailed feedback and for taking time to discuss our answers. This is a kind reminder that tomorrow is the last day reviewers can post replies on openreview. We would love to know if our answers so far have addressed your concerns and if there is anything else we can do to clarify matters further.

---

> > ### Comment · Reviewer_gPTC · 2024-12-01
> >
> > Thanks for writing a detailed response and answering all my concerns. I have updated my score accordingly.

---

### Official Review · Reviewer_Gt4F · 2024-11-03

**Soundness:** 3
**Presentation:** 3
**Contribution:** 3
**Rating:** 6
**Confidence:** 4

**Summary:**

The central objective of this paper is to design a discrete denoising diffusion model for graph-to-graph translation, where the task is to train a model to predict a target graph given a source graph and, perhaps surprisingly, a matrix of node mappings between them. The authors use [1] as their design template, though the state transition matrix of the forward process relies on the absorbing state formulation from [2]. In this model, the backward process relies on the denoiser network realized by an equivariant transformer architecture, which, on its own, can take only the target graph as input. The authors' main challenge is augmenting the denoiser by also taking the source graph as input. The authors demonstrate theoretically and empirically that a simple augmentation of the denoiser's input by the source graph is not sufficient even for simply copying graphs, claiming that the equivariance of the denoiser is the main reason for this issue. Therefore, the authors propose to partially relax the equivariance by further extending the denoiser's input with the node mapping matrix, proposing the aligned permutation equivariance. The key idea is to design the underlying transformer architecture such that if the source and target graphs are permuted (by two separate permutation matrices) and the node mapping matrix is permuted accordingly, then the output of the denoiser is equivariant only under the permutations of the target graph. In other words, the output remains sensitive only to the permutation of the target graph. The authors compare the performance of their model with many baseline methods on the task of chemical retrosynthesis, achieving state-of-the-art top-1 accuracy and mean reciprocal rank on the USPTO-50k dataset. Finally, they also show the suitability of their model in the context of the guided and conditional generation of molecular graphs.

**Strengths:**

The paper presents a comprehensive set of experiments that demonstrate the benefits and the practical utility of the proposed model in the context of guided and conditional generation of molecular graphs. The model achieves state-of-the-art results in chemical retrosynthesis.

The paper concerns an interesting problem, and its central message is important to the community.

Nice and clear illustrations accompany the ideas introduced in the paper.

**Weaknesses:**

The writing deserves improvement. Certain parts of the paper need to be communicated more clearly. The notation sometimes needs to be clarified. These concerns are detailed in the following comments.

Major:

The content from the start of Section 3 up to Section 3.2 qualifies more like a background section. The material is heavily inspired by [1]. Please note that [1] also uses conditioning, though it is not presented in the equations in such an explicit way. It takes more work to recognize the details behind the authors' contributions in the current form. Can the authors clearly delineate which parts are background and which are novel contributions?

In line 124 the authors define $X=P^{Y\rightarrow X}Y$ as $(X^N,X^E)=(P^{Y\rightarrow X}Y^N,P^{Y\rightarrow X}Y^E(P^{Y\rightarrow X})^T)$, and in line 151 define $D_\theta(X,Y)=(D_\theta(X,Y)^N,D_\theta(X,Y)^E)$, saying that $D_\theta(X,Y)^N$ and $D_\theta(X,Y)^E$ output a probability vector for each node and edge, respectively. Now, in line 107, the authors say that $X^N$ and $X^E$ are one-hot encoded (again, it is not suitable to use $\mathbb{R}$ in lines 104 and 105 to define this), then how is it possible that $D_\theta(X,Y)=P^{Y\rightarrow X}Y$? There is no mention that the output of $D_\theta(X,Y)$ is implicitly one-hot encoded. This confusion repeats throughout the text, and clear definitions should stabilize it. The authors should

- clarify when and where $X^N$ and $X^E$ are one-hot encoded or real-valued matrices/tensors,
- explicitly define the output space of $D_\theta(X,Y)$,
- explain how the equality $D_\theta(X,Y)=P^{Y\rightarrow X}Y$ is possible given the different encodings,
- use consistent notation throughout the paper for the spaces these variables belong to.

Minor:

The introduction should better motivate why graph-to-graph translation is important in practical applications.

Line 50: *``a solution aligned''* -> *``a solution: aligned''*

Figure 1: *``w.r.t to''* -> *``w.r.t.''*

In Section 3.3, the authors use phrases such as *``should be possible''*, *``should have learned''*, *``should be the same''*. Why do not use a more certain language? Can the authors please clarify what is or is not true instead of something that should be?

The definition of graph-to-graph translation task deserves improvements. Specifically, if the authors define that the node feature matrix and edge feature tensor belong to the set of reals, why do they redefine them as one-hot vectors in line 107? If they are one-hot, then they are only a subset of reals.

There should be $Y$ on the left side of (2). Consequently, how does this density differ from (3)?

Line 153: *``such that we have a probability vector for each node and edge.''* Probability is not the whole set of reals. It only ranges from 0 to 1.

Line 217: The use of *`respectively'* does not fit the sentence's construction.

Line 243: $\mathbf{X}=\mathbf{P}^{\mathbf{Y}\rightarrow \mathbf{X}}Y$ -> $\mathbf{X}=\mathbf{P}^{\mathbf{Y}\rightarrow \mathbf{X}}\mathbf{Y}$

Line 323: What is DPS?

Line 374: What is MRR? It is first defined in line 402.

The explanation of retrosynthesis in lines 337 and 358 is nearly identical.

Line 337: *``3 parts"* -> *``three parts''*

Table A3 should be in the main text (e.g., Figure 8 can be moved to the appendix).

Figure 3 demonstrates the case for $X_T$, yet there is $D_\theta(X_t,Y)$.

**Questions:**

Do the authors need the node mapping matrix for this task?

Can the authors explain why the left part of (6) should hold (in light of the above concerns with the output of the denoising network $D_\theta$ and the one-hot encoding of $X$)?

Why do the authors present Theorem 1 for fixed t=T? Is the result for $X_T$ not rather obvious (by definition)?

If the authors use $P^{Y\rightarrow X}$ and $Y$ as input into the denoiser, then these two terms already form the full information about $X$. The authors seem to train the denoiser that always has direct access to $X\_0$ through $P^{Y\rightarrow X}Y$. The consequence of this is that $\tilde{p}\_\theta$ in (3) contains $P^{Y\rightarrow X}$ in the condition, i.e., $\tilde{p}\_\theta(X\_0|X\_t,Y,P^{Y\rightarrow X})=\tilde{p}_\theta(X\_0|X\_t,X\_0)$. This design choice is unsound. Please, can the authors clarify?

Can the authors please provide a more comprehensive explanation of the following comment (starting in line 227): *``This makes it impossible for the model ... some random permutation of $Y$.''*?

---

> ### Author Response · Authors · 2024-11-21
> **Response to Reviewer Gt4F (1/2)**
>
> We thank the reviewer for taking the time to write a detailed review and for pointing out places where our text could be improved. Below we answer their questions, clarify minor misunderstandings, and implement the improvements they suggested.
>
> Note that the reviewer mentions references [1] and [2] but their text is missing the corresponding citations. We deduced from the context the following references. Let us know if they are not correct.
>
> [1] Vignac et al. "DiGress: Discrete Denoising diffusion for graph generation", ICLR 2023.
>
> [2] Austin et al. "Structured Denoising Diffusion Models in Discrete State-Spaces", NeurIPS 2021.
>
> **Major concerns**
>
> **...which parts are background and which are novel contributions?**
> We do not claim any significant novelty in the discrete diffusion framework used, and agree that the discussion on diffusion models is better organised into a separate background section. The methods section now contains the novel contributions, including the theoretical analysis on limitations of equivariant denoisers, the concept of aligned permutation equivariance, and the methods for inducing aligned equivariance.
>
> **The output space of $D_{\theta}$ not matching with the one-hot encoding of $X^N$ and $X^E$ in $D_{\theta}= P^{Y \rightarrow X} Y$**
>
> We agree that a clarification is in place, and we have made the following changes to the presentation of the paper:
>
> - Spaces of $X^N$ and $X^E$: We have now defined the node and edge vectors to be in the vertices of the probability simplex $\Delta^{K_a-1}$ and $\Delta^{K_b-1}$, where $K_a$ and $K_b$ are the amount of node and edge types, respectively. Formally, for the entire tensor $X^N\in$ (vert$(\Delta^{K_a-1}))^{N_X}$ and $X^E\in$ (vert$(\Delta^{K_b-1}))^{(N_X\times N_X)}$.
> - The output space of $D_\theta(X,Y)=(D_\theta(X,Y)^N,D_\theta(X,Y)^E)$ is the probability simplex for each node and edge, that is, $D_\theta(X,Y)^N\in (\Delta^{K_a-1})^{N_X}$ and $D_\theta(X,Y)^E\in (\Delta^{K_b-1})^{(N_X\times N_X)}$.
> - Thus, the output space of $D_\theta(X,Y)^N$ is a strict superset of the space of $X^N$, and the denoiser can output one-hot values as well. $D_\theta(X,Y)=P^{Y\to X}Y$ means that the denoiser outputs probability vectors that are entirely peaked at a specific permutation of $Y$, and the entropy of the distribution is zero. This would also mean that the denoiser has learned to copy the graph, up to a permutation.
>
> We have now gone through the text and clarified the notation everywhere. We are happy to address any further comments on this point.
>
> **Minor concerns**
>
> **why graph-to-graph translation is important in practical applications.** We agree that more motivation would be helpful, and have now added a few sentences to the introduction to strengthen it. The new sentences are highlighted in teal. To summarise, application areas are molecular editing and property optimisation, chemical reaction modeling, and tasks in which the objective is to predict a future state of a graph.
>
> **Use of uncertain phrases** We have now revised our text to replace all uncertain language with clear and concise descriptors. For instance:
> - In Section 4.1: "a model that is capable of copying graphs from one side of the reaction to the other should also be extendable to modulations..."" -> we removed the sentence and gave a more descriptive motivation
> - In Section 4.1: "while the copy-paste task should be solvable..." -> "The optimal unconstrained denoiser in this situation is always $D_\theta(X_t, Y) = P^{Y\to X}Y$"
>
> **Typos, and minor improvements** We fixed the following typos and minor mistakes per the reviewer's request: Line 50, Figure 1, respectively taken out of line 217, bolding in line 244, DPS -> Diffusion Posterior Sampling, MMR in line 374 -> included a brief description of the metric earlier, changed the definition of retrosynthesis in line 374, swapped table a3 with figure 8.
>
> **Y on the left side of (2). Consequently, how does this density differ from (3)?** Thank you for pointing out the missing $Y$. Eqs. (2) and (3) refer to the same density, but the point of (3) is to provide a method to calculate the single-step transition given a network that has been trained to predict $X_0$.

---

> ### Author Response · Authors · 2024-11-21
> **Response to Reviewer Gt4F (2/2)**
>
> **Questions**
>
> **DPS, MRR** These refer to the diffusion posterior sampling [1] guidance method and the mean reciprocal rank. We have added the descriptions in the text.
>
> **Do the authors need the node mapping matrix for this task?** We need some kind of mapping information between the nodes in the input and output graphs. Such information is often available (such as the atom mapping information in chemical reaction modelling), or could be obtained with graph matching algorithms. We formally represent this information in the node mapping matrix to enable mathematical analysis and to allow us to define the concept of aligned equivariance. We expect alignment to be useful when the node mapping is informative of the structural similarities in the two graphs: If there is an edge between two mapped nodes in the initial graph, there should be a high probability of an edge between the corresponding nodes in the target graph.
>
> **Why do the authors present Theorem 1 for fixed t=T?** Theorem 1 uses a fixed t to provide intuitive understanding and highlight the main issue of equivariant denoisers. Note that it does apply to other forward processes aside from the absorbing state transition as well, although it is easiest to see for the absorbing state diffusion. **We extend the results to other t<T in Appendix A.6. for a simplified setup.** We have revised the main text in Methods to clarify this point.
>
> **If the authors use P and Y as input into the denoiser, then these two terms already form the full information about X.**
> In the identity data case, the conditioning graph $Y$ in itself already contains full information about the graph $X$, up to permutation. While this is restrictive, our aim in the section was to then argue the following:
> - Although one could expect the task to be trivial even without node mapping, we show that a permutation equivariant diffusion model is unable to represent the graph-copying operation effectively.
> - We then show that the class of aligned equivariant functions is able to represent the copying operation
> - Importantly, **it does not matter which $P^{Y\to X}$ matrix we use during sampling**. We can choose a random mapping matrix, which will result in the model to output a correspondingly random permutation of the conditioning graph $Y$. Thus, the effect of $P^{Y\to X}$ is to break symmetries, not to provide additional information.
>
> During training, however, we need actual mapping information so that the model will learn to associate structurally similar nodes from one graph to the other.
>
> **A more comprehensive explanation of "This makes it impossible for the model ... some random permutation of $Y$"**
> An expanded version of the same comment goes as follows:
> - Because the equivariant denoiser outputs a factorized marginal distribution of nodes and edges in the graph, sampling from it will result in a highly noisy graph where the edges and nodes are uncorrelated.
> - However, with multiple steps, correlations can emerge. The extreme case is that if $T$ is very large, only a single dimension of $X$ will change at once. The factorized output distribution does not cause issues here, since we are only changing one dimension at a time, turning the model into an autoregressive model that generates the dimensions in a random order. This is very slow, however.
>
> We have updated the description in the submission to be in line with this explanation, as we believe it is more precise. We are happy to answer any further concerns.
>
> **References**
>
> [1] Chung et al., "Diffusion Posterior Sampling for General Noisy Inverse Problems"

---

> > ### Comment · Reviewer_Gt4F · 2024-11-25
> >
> > I want to thank the authors for a detailed response, and I appreciate all the effort you have put into its preparation. However, I still have concerns that require further clarification.
> >
> > Regarding your note on the references, I apologize for not including them in the review. Those you are mentioning are correct.
> >
> > I especially appreciate that (i) the separation between the authors' original contributions and the background is now transparent, and (ii) the introduction of the probabilistic simplex and its vertices now makes a clear distinction between the outputs of $\mathbf{D}_\theta$ and $\mathbf{X}$.
> >
> > You can find a few references to strengthen your example applications of g2g translation.
> >
> > *``Thank you for pointing out the missing $\mathbf{Y}$. Eqs. (2) and (3) refer to the same density, but the point of (3) is to provide a method to calculate the single-step transition given a network that has been trained to predict $\mathbf{X}_{0}$.''*
> >
> > I disagree that (2) and (3) are the same density. There is a clear relationship between the two.
> >
> > *``We have revised our text to replace all uncertain language with clear and concise descriptors.''*
> >
> > I can no longer see Section 4.1 in the paper nor find the phrases you refer to. I believe that the authors mean the first two paragraphs of Section 5.
> >
> > Sometimes, the authors use the comma behind `i.e.' and sometimes not. Please make it consistent throughout the text.
> >
> > *``We need some kind of mapping information between the nodes in the input and output graphs.''* *``During training, however, we need actual mapping information so that the model will learn to associate structurally similar nodes from one graph to the other.''*
> >
> > The g2g translation task is defined without the knowledge of the node mapping matrix. I understand why the authors need this matrix. However, I need help finding the experiments convincing to what extent the performance of the proposed model is given by the proposed alignment or the use of the Laplacian positional encoding. The ablation study does not cover this case (i.e., Unaligned-PE). Would it be possible to see an experiment only with the Laplacian positional encoding? Section 2 covers this as a related work, justifying it as a valid ablation case. It is a quick modification of the authors' model. There should also be a standalone case using only the skip connections. These essential ablations are crucial for supporting the paper's key message. Why did the authors not include these experiments in the paper?
> >
> > I am still concerned about the authors' answers regarding *`$P$ and $Y$ from the full information about $X$*'. Let us, for example, consider the case in line 333, where $f_\theta([X^N_t \hspace{4pt} P^{Y\rightarrow X}Y^N], [X^E_t \hspace{4pt} P^{Y\rightarrow X}Y^E(P^{Y\rightarrow X})^T])=f_\theta([X^N_t \hspace{4pt} X^N], [X^E_t \hspace{4pt} X^E])$. If the denoiser uses $X^N$ as an input, then what is the purpose of having its noisy version $X^N_t$? Is it not the case that the denoiser will learn to ignore the noisy input? Is there any purpose of using the diffusion model if the transformer has access to $X$ at each time step?

---

> > > ### Author Response · Authors · 2024-11-28
> > > **Answers to follow-up questions and suggestions**
> > >
> > > We thank the reviewer for the further suggestions, and the opportunity to further clarify our work. The discussion has really improved the overall quality of the work, on both theoretical and empirical aspects. Answers/discussion points to the further concerns below:
> > >
> > >  **"You can find a few references to strengthen your example applications of g2g translation.""** We have now added references to molecular property optimization [1], retrosynthesis [2] and dynamic graphs [3].
> > >
> > >  **"I can no longer see Section 4.1...""** Thank you for the catch, this was referring to the old section numbering. Indeed, we meant the new beginning of Section 5.
> > >
> > >  **"Sometimes, the authors use the comma behind i.e. and sometimes not.""** We corrected this to now always use the comma.
> > >
> > > **"I disagree that (2) and (3) are the same density. There is a clear relationship between the two.""**
> > > We have now added a much more thorough exposition on what we mean in **Appendix B, "Connection between the denoising parameterization $\tilde p(X_0\mid X_t, Y)$ and $p(X_{t-1}\mid X_t, Y)$."** In short, what we mean is that we parameterize the reverse transition in Eq.2 using Eq.3. In particular, in Eq.2
> > >
> > > $p_{\theta}(X_{t-1} \mid X_t, Y) = \prod_{i=1}^{N_X} p_{\theta}(X_{t-1}^{N,i} \mid X_t, Y) \prod_{i,j}^{N_X} p_{\theta}(X^{E,i,j}_{t-1} \mid X_t, Y)$,
> > >
> > > we define the individual node transition probabilities as (similarly for edges)
> > >
> > > $p_{\theta}(X_{t-1}^{N,i} \mid X_t, Y)=\sum_{X_0^{N,i}} q\left(X_{t-1}^{N,i} \mid X_t^{N,i}, X_0^{N,i} \right) \tilde p_{\theta}\left(X_0^{N,i} \mid X_t, Y\right)$.
> > >
> > > Thus, the probability for an individual node/edge in $X_{t-1}$ is given by summing over all values of $X_0$ for that particular node/edge. Let us know if there is further disagreement.
> > >
> > >  **Extent of performance improvement due to Laplacian positional encoding (PE) or alignment?** To disentangle the effect of the graph inductive bias brought by the graph Laplacian positional encodings and the alignment inductive bias brought by matching the positional encodings for the input and output graph, we trained two models:
> > >  1) A model where the only alignment method is the matched PE where the PE is generated from the graph Laplacian eigenvectors of the conditioning graph $Y$ (our default PE method). This contains the alignment information, but also inductive biases about the graph structure of $Y$.
> > >  2) A model where the only alignment method is matched PEs where the PE is generated by sampling random Gaussian noise vectors on the conditioning graph nodes $Y$ and placing these same vectors on the $X$ side. This only contains the alignment information, and no inductive biases about the graph structure.
> > >
> > >  To provide a quick and fair comparison, we evaluated both models at 10% of our full training budget (epoch 80). The results below are on 512 samples from the test set.
> > >
> > > |Model|Top-1|top-3|top-5|top-10|top-50|MRR|
> > > |-|-|-|-|-|-|-|
> > > |DiffAlign-PE-Gaussian|45.51%|68.55%|73.24%|75.78%|83.20%|0.5649|
> > > |DiffAlign-PE-Laplacian|46.09%|68.75%|72.85%|75.98%|83.98%|0.5686|
> > >
> > > The results are very similar, implying that the inductive bias brought by the Laplacian PE is not very significant.
> > >
> > > **Standalone results only using the skip connections.** We similarly trained a model using skip-connection only as an alignment method, and compare its results to the standard DiffAlign-PE+Skip following the same protocol detailed above (both at 10% of full training)
> > >
> > > |Model|Top-1|top-3|top-5|top-10|top-50|MRR|
> > > |-|-|-|-|-|-|-|
> > > |DiffAlign-Skip|9.77%|17.38%|24.61%|29.30%|32.23%|0.1517|
> > > |DiffAlign-PE+Skip (Laplacian)|**49.22%**|**70.90%**|**74.61%**|**77.93%**|**84.96%**|**0.5952**|
> > >
> > >  It is evident that only having the skip connection does not work well, although it is better than an entirely unaligned model (4% top-1 after full training). This is likely due to the skip-only model having limited expressivity: The neural net does not get the alignment information in the input at all, and thus is unable to use it apart from the skip connection, which enables literally copying the $Y$ graph from input to output. For tasks other than the identity data, this is not good enough.
> > >
> > >  We have updated the results in a new Appendix E.

---

> > > > ### Author Response · Authors · 2024-11-28
> > > > **Continuing, on the use of X_t as input**
> > > >
> > > > **"If the denoiser uses as an input, then what is the purpose of having its noisy version $X_t^N$? Is it not the case that the denoiser will learn to ignore the noisy input?"**
> > > >
> > > > We believe the concern may stem from a miscommunication regarding the use of node-mapping in the *identity data context* and for *non-identity data context*. To facilitate discussion, we explain our intent in more length in the two contexts below. In case there is a disagreement, perhaps you can point to the part where we disagree and we can continue from there?
> > > >
> > > > **In the case of the identity data**, the model indeed gets $X^N$ as input, and as such it can ignore the noisy version $X_t^N$. In particular, if it has the node mapping information, the model can output $X^N$ in a single step. The point of Theorem 1 is that with a permutation equivariant denoiser, however, this is not possible. We want this trivial task to be possible in a single step, because the identity data is similar to more realistic types of data.
> > > >
> > > > **For other types of data, $Y^N\neq X^N$**. However, $Y$ may contain a significant amount of information about $X$ in the sense that if there is an edge between two mapped nodes in $Y$, then there is a high probability of an edge between the corresponding mapped nodes in $X$. With alignment information, the model effectively **learns to copy a subgraph of $Y$ to $X$, and generate the other parts.** At inference time, it does not have access to information about which edges are copied and which edges are not. This is a task for the neural network to infer based on the training data. *With a permutation equivariant denoiser,* this subgraph copying is inherently hindered due to the same inability to break symmetries that is behind Theorem 1.
> > > >
> > > > We thank again for the effort in improving the paper, and are open to more discussion.
> > > >
> > > > **References.**
> > > >
> > > > [1] Lin et al., "Learning Multimodal Graph-to-Graph Translation for Molecular Optimization", ICLR 2019
> > > >
> > > > [2] Shi et al., "A Graph to Graphs Framework for Retrosynthesis Prediction", ICML 2020
> > > >
> > > > [3] Rossi et al., "Temporal Graph Networks for Deep Learning on Dynamic Graphs"

---

> > > > > ### Comment · Reviewer_Gt4F · 2024-11-29
> > > > >
> > > > > I want to thank the authors for further clarifications. Again, I must appreciate the authors' engagement and effort in improving the paper.
> > > > >
> > > > > **The relation between (2) and (3).**
> > > > >
> > > > > It is perhaps my mistake. I did not mean there was any mathematical disagreement between the two. However, (2) and (3) are not the same densities from the symbolical perspective. My concerns resulted from the need for more notational distinction in the l.h.s. of (2) and (3). Following the style of the authors' notation, e.g., (4) uses the superscript $ N $ in the first argument, I expect the same superscript (or perhaps something more universal) to hold in (3) as well. This overloading of symbols should be made clear when the authors first use it, or when introducing the notation, or (less preferably) using the symbol such as in (4) throughout the text.
> > > > >
> > > > > **Disentangling graph inductive biases from alignment.**
> > > > >
> > > > > The authors' ablation is meaningful, especially the one with the standalone skip connection. Thanks for producing the experiments so quickly. I understand that, under the time constraints, it is not possible to use the full training budget. However, as seen from my previous reply, I was more concerned about the case where there is no alignment and only the Laplacian positional encoding is used, i.e., Unaligned-PE. Would it still be possible to see this case?
> > > > >
> > > > > **On the use of X_t as input.**
> > > > >
> > > > > Indeed, the two cases the authors clarify are described in the paper. I am rather concerned with that if the authors use $P^{Y\rightarrow X}$ and $Y$ as inputs into the denoiser, then what if the denoiser only learns to multiply these two terms and learns only from the resulting $X$, ignoring the noisy version $X_t$ in the process? How does the performance change when the authors keep $X_t=X_T$ during the training? In this case, the authors' Algorithm 1 no longer samples from $q$, and their model becomes **similar** to, e.g., the NADE model (https://arxiv.org/pdf/1605.02226). Under this setting, we have a neural network (it can also be a transformer network), which takes $X$ as input, yet it is trained with the cross entropy, which also takes $X$ as input. This case of ignoring the noisy input while having full access to $X$ (directly or via the product of $P^{Y\rightarrow X}$ and $Y$) concerns me the most.

---

> ### Author Response · Authors · 2024-11-29
> **Preliminary answers to the follow-up questions**
>
> We thank the reviewer for the further clarifications. It seems that we are now getting towards a shared understanding in most questions, with perhaps room for some minor clarifications.
>
> **On the relation between (2) and (3)**. We now understand better what the reviewer meant in the previous message, and agree that a clarification is useful here. We have now added new Equations (4) and (5) in the text that clearly show the factorization over dimensions, mirroring equations (1) and (2).We list the new equations below:
>
> $q\big(X_{t-1} \mid X_t, X_0 \big) = \prod_i^{N_X} q\big(X_{t-1}^{N,i} \mid X_t^{N,i}, X_0^{N,i} \big) \prod_{i,j}^{N_X} q\big(X_{t-1}^{E,i,j} \mid X_t^{E,i,j}, X_0^{E,i,j} \big)$
>
> $\tilde p_{\theta}\big(X_0  \mid X_t, Y\big) = \prod_i^{N_X} \tilde p_{\theta}\big(X_0^{N,i}  \mid X_t, Y\big) \prod_{i,j}^{N_X} \tilde p_{\theta}\big(X_0^{E,i,j}  \mid X_t, Y\big)$
>
> EDIT:  It is unfortunately not possible to edit the paper at the moment, but we can add these two lines later for a camera-ready version. Of course, it does not make a significant difference for the main content of the paper.
>
> **No alignment and only Laplacian positional encodings.** We are glad that the reviewer is happy with the additional experiments, but also better understand the remaining suggestion. We understood the intent of the question to be about what is the comparative effect of Laplacian PEs vs. alignment, but now realise that the intent is to understand whether Laplacian PEs without alignment can be useful. We can see a couple of ways to formalise this, but perhaps the following procedure is the most meaningful:
> 1. Create the Laplacian PE for $Y$ in the same manner as before.
> 2. Instead of copying the $Y$ positional encodings to $X$, we create new Laplacian PEs based on the noisy graph $X_t$. The caveat that this does not contain any structural information in the beginning of generation when the graph is in the absorbing state.
>
> We are now trying to run this experiment, and will get back to you soon. We are sending this response now to leave more time for discussing the latter point.
>
> **Ignoring the $X_t$ input in the denoiser.**
>
> We thank the reviewer for the clarification. It seems that the miscommunication is easier to spot now. We list subquestions / comments that had the most tangible opportunities for clarification, and our answers to them below. Perhaps this resolves some of the issues? Let us know also if there is an additional experiment that you think would be useful to run regarding this.
> - "How does the performance change when the authors keep $X_t=X_T$ during the training?"
>     - In fact, this experiment is implicitly included in Figure 6b, which shows the top-k scores for different sampling step counts. In the case where of a single sampling step, the model only gets as input $X_T$ and $Y$ and $P^{Y\to X}$, and directly outputs the estimate of $X_0$. Although the model has technically been trained on other $t$ values as well, they are not used here and the model is mathematically equivalent to one that was trained solely on $X_t=X_T$ (assuming that the other $t$ do not interfere significantly with the training). The results are clearly worse, although it does perform better than the 100-step unaligned model.
> - "This case of ignoring the noisy input while having full access to $X$ (directly or via the product of $P^{Y\rightarrow X}$ and $Y$) concerns me the most."
>     - We would like to stress that the model never has full access to $X$, aside from the identity data experiment. As such, this scenario can not occur. It only has access to $Y$, which can significantly overlap with $X$, but not entirely.
>
> Let us know if this sheds light on the matter.

---

> > ### Author Response · Authors · 2024-12-01
> > **Discussion deadline ending soon**
> >
> > We thank reviewer Gt4F once again for a detailed discussion that has improved the quality of our paper considerably. This is a kind reminder that tomorrow is the last day reviewers can post replies on openreview. We would love to know if our answers so far have addressed your concerns and if there is anything else we can do to clarify matters further.

---

> > > ### Author Response · Authors · 2024-12-01
> > > **Ablation on an unaligned model with Laplacian PEs**
> > >
> > > We now have the results for the model with graph Laplacian positional encodings, but without aligning the positional encodings across sides. The procedure was the same as we did for the previous ablations (10% of full training). For comparison, we use to the aligned Laplacian positional encodings from the previous ablations. The only practical difference between the runs is a few lines of code that specify how the positional encodings are assigned to the nodes. The top-k results are as follows:
> > >
> > > |Model|Top-1|top-3|top-5|top-10|top-50|MRR|
> > > |-|-|-|-|-|-|-|
> > > |Unaligned-Laplacian-PE|0.2%|0.2%|0.2%|0.8%|0.8%|0.003
> > > |DiffAlign-PE-Laplacian|46.09%|68.75%|72.85%|75.98%|83.98%|0.5686|
> > >
> > > The results from the models with only the positional encodings without alignment are very limited. It could be that by training for longer, we could achieve something similar to our unaligned model in Table 2 (top-1=4%). The suboptimal results are not surprising given our Theorem 1: The unaligned model is unable to break symmetries in $X_t$ (regardless of the positional encoding), inhibiting performance.
> > >
> > > Please, let us know it you have further concerns. This has been a very fruitful discussion, and we think it has improved the paper considerably.

---

> > > > ### Comment · Reviewer_Gt4F · 2024-12-01
> > > >
> > > > Again, I would like to thank the authors for quickly producing the experiments for the Unaligned-PE model. Admittedly, I expected a slightly better performance, though not a better one than the DiffAlign-PE model. This case completes the ablations nicely.
> > > >
> > > > Regarding the experiments in Figure 6. In the paper, the authors state the following: *``While we use a relatively large value for $T$ during training in our best models, we also highlight that the performance of the aligned model does not degrade significantly when reducing the count of sampling steps to a fraction of $T = 100$.''* In the previous reply, they also say: *``Although the model has technically been trained on other $t$ values as well, they are not used here and the model is mathematically equivalent to one that was trained solely on $X_t=X_T$ (assuming that the other do not interfere significantly with the training).''* From these two claims, I gather that the model was not trained with $X_t=X_T$, and Figure 6 shows the performance only for different numbers of sampling steps after the training. I guess the training with $X_t=X_T$ is not feasible in the time that remains till the end of the discussion deadline.
> > > >
> > > > Would it also be possible to provide a few comments on the computational complexity of graph matching algorithms, as they are needed to obtain $P^{Y\rightarrow X}$?

---

> > > > > ### Author Response · Authors · 2024-12-01
> > > > > **Answers to remaining questions**
> > > > >
> > > > > We are glad to hear that the reviewer is happy with the additional ablations, and again thank them for the engagement in improving the paper.
> > > > >
> > > > > **The situation of $X_t=X_T.$** Indeed, it is difficult to ensure that we would get new results on only training on $X_t=X_T$ by the end of the discussion period. We would, however, like to clarify what we mean by the equivalence of the model in Figure 6 and training on $X_t=X_T$:
> > > > > 1. Denote the optimal denoiser for a model trained with $T$ timesteps as $D_T^{opt}(X_t,t,Y)$ (optimal with respect to our cross-entropy denoising loss). There are T time-indexed functions, and we learn all of them using a single neural network that takes time t as an input.
> > > > > 2. If a model is trained only on the highest noise level $X_t=X_T$, it does not matter what $T$ is since we always sample $X_t$ from the exact same fully noisy distribution.
> > > > > 3. From this, it follows that $D_1^{opt}(X_1,t=1,Y) = D_{100}^{opt}(X_{100},t=100,Y)$. To reiterate, the distribution of $X_1$ with $T=1$ is exactly the same as the distribution of $X_{100}$ with $T=100$.
> > > > > 4. The accelerated sampling process with a single step for the $T=100$ model (left side of  Figure 6b) is defined to be the sampling process that we would have if $T=1$, but we use the trained $D_{100}(X_{100},t=100,Y)$ as the denoiser, instead of $D_{1}(X_1,t=1,Y)$. If we have the optimal value for $D_{100}(X_{100},t=100,Y)$, then the output of the sampling process is exactly the same as with $T=1$.
> > > > >
> > > > > We hope that Figure 6 is a strong enough indication that a model trained with $T=1$ is likely worse than a model trained with $T=100$ and using the full 100 generation steps. It is also very common in diffusion models that using more steps improves performance.
> > > > >
> > > > > **Computational complexity.** This is an excellent question. The graph matching/alignment problem is, in its full generality, NP-Hard [1], but efficient polynomial-time approximations exist. One class are spectral methods [1,2,3], which have complexity $O(n^3)$, where $n$ is the amount of nodes (although this can be reduced by using truncated eigendecompositions). Other methods include graduated assignment [4], which relaxes the discrete matching into a continuous problem with $O(ml)$ complexity, where $m$ and $l$ are the amount of links in the two graphs, respectively. Other polynomial-time methods include, e.g., random-walk-based approaches [5]. For chemical reaction modelling, the problem is particularly well studied (called *atom-mapping*) and many fast algorithms exist [6,7,8,9]. For instance, [7] is based on Transformer networks, and as such have computational complexity $O(n^2)$, and is parallelizable on GPUs (also worth noting that linear-complexity Transformers exist [10]). The dataset and atom mapping we use for the chemical reaction experiments was originally derived from [11], where they used the NameRxn-atom mapping software, but the computational complexity of the method is not publicly documented.
> > > > >
> > > > > We will include this discussion and the citations in the Appendix (with a reference in the main text) in a potential camera-ready version.
> > > > >
> > > > > **References.**
> > > > >
> > > > > [1] Feizi et al., "Spectral Alignment of Graphs"
> > > > >
> > > > > [2] Knossow et al., "Inexact Matching of Large and Sparse Graphs Using Laplacian Eigenvectors"
> > > > >
> > > > > [3] Fan et al., "Spectral Graph Matching and Regularized Quadratic Relaxations: Algorithm and Theory"
> > > > >
> > > > > [4] Gold et al., "A Graduated Assignment Algorithm for Graph matching"
> > > > >
> > > > > [5] Gori et al., "Graph matching using Random Walks"
> > > > >
> > > > > [6] Lin et al., "Atom-to-atom mapping: a benchmarking study of popular mapping algorithms and consensus strategies"
> > > > >
> > > > > [7] Schwaller et al., "Extraction of organic chemistry grammar from unsupervised learning of chemical reactions"
> > > > >
> > > > > [8] Rahman et al., "Reaction Decoder Tool (RDT): Extracting Features from Chemical Reactions"
> > > > >
> > > > > [9] Chen et al., "Precise atom-to-atom mapping for organic reactions via human-in-the-loop machine learning"
> > > > >
> > > > > [10] Katharopoulos et al., "Transformers are RNNs: Fast Autoregressive Transformers with Linear Attention"
> > > > >
> > > > > [11] Schneider et al, "What’s What: The (Nearly) Definitive Guide to Reaction Role Assignment"

---

> > > > > > ### Comment · Reviewer_Gt4F · 2024-12-02
> > > > > >
> > > > > > Thanks for answering all my questions. The discussion was, indeed, exciting and engaging. I am raising the score accordingly.

---

> > > > > > > ### Author Response · Authors · 2024-12-02
> > > > > > >
> > > > > > > Thank you once again for all your time and effort, our paper is in a much better shape now thanks to this discussion.

---

### Official Review · Reviewer_uWes · 2024-11-19

**Soundness:** 3
**Presentation:** 3
**Contribution:** 3
**Rating:** 6
**Confidence:** 3

**Summary:**

This paper presents a novel approach to graph-to-graph translation using diffusion models. The authors propose methods to address the inherent limitations of equivariant models in aligning nodes between input and output graphs. The paper introduces several strategies for breaking equivariance and empirically validates their approach on retrosynthesis tasks. While the paper is innovative in addressing this important problem, certain theoretical justifications and experimental evaluations could be strengthened.

**Strengths:**

- The paper addresses a novel and relevant problem in graph diffusion literature, particularly in breaking symmetries for graph-to-graph translation, which has not been explored in depth.
- The idea of using equivariant models and tackling their limitations is well-motivated, and the theoretical underpinning is clear and compelling.
- The alignment techniques proposed for mapping nodes between input and output graphs are intuitive and demonstrate promising preliminary results, especially compared to unaligned versions.

**Weaknesses:**

- The justification for certain design choices, such as using the absorbing state distribution instead of marginal distributions (as in DiGress), is insufficiently explained. A more thorough motivation or comparison would be beneficial.
- Some key claims, particularly in Section 3.2, are vague or lack detailed explanation. For example, the statement regarding the model's capability to handle graph transformations requires clearer elaboration.
- The experimental validation, while promising, lacks sufficient quantitative comparisons to existing methods (e.g., RetroBridge). More thorough benchmarking and analysis would strengthen the empirical contribution of the paper.

**Questions:**

Please refer to  the weaknesses.

---

> ### Author Response · Authors · 2024-11-21
> **Response to reviewer uWes**
>
> We thank the reviewer for taking the time to review our work and for recognizing the value of our contribution. Below we address their concerns.
>
> **Justification for design choices.** For studying the equivariance properties of the denoiser, the exact choice of the transition matrix does not matter. We initially focused on the absorbing state transitions due to prior empirical evidence of it performing better than alternatives [1,2], and being the transition of choice in many recent works [2,3,4]. We also noted in our earlier experiments that the marginal transition was similar or performed marginally worse than the absorbing state transition.
>
> **[...]  the statement regarding the model's capability to handle graph transformations requires clearer elaboration.** We have revised this sentence in our text (highlighted in teal). We have replaced it with
>
> "The motivation is that for many graph translation tasks, such as retrosynthesis, the optimal denoising function is close to the identity denoising function due to the inductive bias of structural similarity between the graphs."
>
> We believe that this reflects more accurately the reason why the identity data is a useful toy model.
>
> **The experimental validation, while promising, lacks sufficient quantitative comparisons to existing methods (e.g., RetroBridge). [...]** Upon the request of reviewer Gt4F, we have moved Table A.3 comparing our method to other retrosynthetic baselines from Appendix A to the main text (now Section 9, Table 2). In addition, we included results highlighting our performance compared to RetroBridge [6] in particular, following their evaluation scheme (i.e. discarding formal charges and stereochemical information from the generated samples).
>
> |Model|k=1|k=3|k=5|k=10|MRR|
> |-----|---|---|---|----|---|
> |RetroBridge (T=500)|50.8|74.1|**80.6**|**85.6**|0.622|
> |Ours-uncharged (T=100)|**56.3**|**75.2**|80.4|84.0|**0.658**|
>
> As seen from the table above, we outperform RetroBridge significantly on top-1 and MRR with considerably less denoising steps. Comparing our Figure 5.b to RetroBridge's Figure 5 also highlights that our results are barely affected by lowering the denoising steps, while RetroBridge's top-1 accuracy is close to 0 for steps < 10. These results and more comments are available in Appendix A, Table A.4.
>
> **References**
>
> [1] Austin et al., "Structured Denoising Diffusion Models in Discrete State-Spaces", NeurIPS 2021
>
> [2] Lou et al., "Discrete Diffusion Modeling by Estimating the Ratios of the Data Distribution", ICML 2024
>
> [3] Lou et al., "Discrete Diffusion Modeling by Estimating the Ratios of the Data Distribution", ICML 2024
>
> [4] Sahoo et al., "Simple and Effective Masked Diffusion Language Models", NeurIPS 2024
>
> [5] Shi et al., "Simplified and Generalized Masked Diffusion for Discrete Data", NeurIPS 2024
>
> [6] Igashov et al., "RetroBridge: Modeling Retrosynthesis with Markov Bridges", ICLR 2024

---

### Author Response · Authors · 2024-11-21
**General response**

We thank the reviewers for their detailed feedback. Below we give a summarized response to a few recurrent points in the reviews.

**Choice of transition matrix**   While other transitions, like uniform and marginal [1] are equally possible, the absorbing state model [2] is somewhat simpler and has been empirically found to outperform other transitions [2,3,4,5]. In our early experiments, we found that the uniform transition performed worse while marginal and absorbing performed similarly. Since the main content of the paper is not dependent on the particular choice of transition, we decided to focus the experiments on the somewhat simpler absorbing state model.

**Clarifying the format of the denoiser $D_\theta(X,Y)$ output** Reviewers Gt4F and gPTC expressed concerns about the notation of $D_\theta(X,Y)$, its relation to the one-hot vector $P^{Y\to X}Y$, and to the probabilities $\tilde p(X_0|X_t,Y)$. We have now clarified the connections in the text when introducing $D_\theta(X,Y)$ and other relevant modelling objects. Specifically, we consider the space of one-hot vectors of dimension $K$ as $\text{vert}(\Delta^{K-1})$, where vert(.) denotes the vertices of the probability simplex $\Delta^{K-1}$ with $K-1$ degrees of freedom. The output of $D_\theta(X,Y)$ as well as the one-hot-encoded vectors $X$ and $Y$ are vertices of this probability simplex, making the equality $D_\theta(X,Y) = P^{Y\to X}Y$ sensible mathematically.

**Delineating contributions vs background** We reorganized our sections to differentiate more easily between our contributions vs the framework laid out by [1]. We also added Figure 2 to the introduction to summarize our contributions at a glance.
* *Sec. 5 The Theoretical Limitations of Equivariant Denoisers*: Copying graphs as a case-study on the limitations of equivariance.(Fig. 4), The optimal permutation equivariant denoiser (Theorem 1).
* *Sec. 6 Solution Aligned Equivariant Denoisers*: Relaxing permutation equivariance through alignment (Fig. 6a), Aligned denoisers induce aligned permutation invariant distributions (Theorem 2).
* *Sec. 7 Suitable Alignment Methods*: Skip-connection, positional encodings, and input-alignment (Fig. A9).
* *Sec. 9 Experiments*: a) Proof of concept through simple graph copying tasks (Fig. 5), b)
Application: Retrosynthesis (Table 2), c) Guided-generation for retrosynthesis (Fig. 8a), d) Ibuprofen synthesis through inpainting (Fig. 7).

We also moved all the background information, including the framework of [1] and formal definitions of permutation equivariance and invariance as they pertain to diffusion model, to a dedicated section (Section 2, "Background").

**Results compared to RetroBridge** Following RetroBridge's evaluation procedure, which does not take into account atom charges nor stereochemistry, we outperform their results significantly with an over 5% increase in top-1 accuracy. We share the detailed top-k accuracies and mean reciprocal rank below, and include them in Appendix D, Table A.4.

|Model|k=1|k=3|k=5|k=10|MRR|
|-----|---|---|---|----|---|
|RetroBridge (T=500)|50.8|74.1|**80.6**|**85.6**|0.622|
|Ours-uncharged (T=100)|**56.3**|**75.2**|80.4|84.0|**0.658**|

Our model achieves this with 100 denoising steps, as opposed to 500 by Retrobridge, and achieves higher top-1 scores with 5 steps than Retrobridge with 100 steps (see their Figure 5 compared to our Figure 5b). We believe that this improvement is due to us using multiple methods for alignment, which were inspired by our theoretical framework.

**Performance compared to other graph-translation baselines** Prompted by reviewer HVWS, we higlight our model results compared to the seminal G2G [2] graph translation-based retrosynthesis method. Top-k accuracies and the mean reciprocal rank are listed below:

| Method | k = 1 | k = 3 | k = 5 | k = 10 | MRR |
|--------|-------|-------|-------|--------|-----|
| G2G | 48.9 | 67.6 | 72.5 | 75.5 | 0.582 |
| DiffAlign-PE+skip |**54.7** | **73.3**| **77.8** | **81.1** | **0.639** |

We note that this is a semi-template-based (or synthon-based) model with strong assumptions in retrosynthesis, namely the existence of a reaction center which is first broken to identify synthons. Such assumptions do not apply to all chemical reactions unfortunately, and thus G2G[2] is inherently limited by its inability to model reactions like the one shown in our Figure A11, Appendix D.

**References**

[1] Vignac et al., "DiGress: Discrete Denoising diffusion for graph generation", ICLR 2023

[2] Austin et al., "Structured Denoising Diffusion Models in Discrete State-Spaces", NeurIPS 2021

[3] Lou et al., "Discrete Diffusion Modeling by Estimating the Ratios of the Data Distribution", ICML 2024

[4] Sahoo et al., "Simple and Effective Masked Diffusion Language Models", NeurIPS 2024

[5] Shi et al., "Simplified and Generalized Masked Diffusion for Discrete Data", NeurIPS 2024

---

> ### Author Response · Authors · 2024-11-28
> **Summary of additional ablations**
>
> Prompted by the discussion with reviewers Gt4F and gPTC, we did the following additional ablations:
> 1. Comparison of our absorbing-state model against different transition matrices (uniform and marginal)
> 2. Comparison of our absorbing-state skip connection + Laplacian PE model with a model that only uses the skip connection as an alignment method.
> 3. An ablation disentangling the effect of the inductive bias brought by using the graph positional encodings and the inductive bias of matching the positional encodings across the conditioning graph $Y$ and the output graph $X$. This is done by comparing a model with only Laplacian positional encodings as alignment to a model with random Gaussian vectors matched across $X$ and $Y$ as alignment.
>
> For a fair and quick comparison, all of the models were evaluated at roughly 10% of the full training budget (80 epochs) and 512 samples from the test set.
>
> **The ablation on transition matrices** shows that while the uniform transitions perform the least well and the absorbing transitions perform the best, all of them can work with a reasonable accuracy.
>
> |Model|Top-1|top-3|top-5|top-10|top-50|MRR|
> |-|-|-|-|-|-|-|
> |Uniform-PE+Skip|43.75%|58.98%|62.89%|67.38%|72.66%|0.5156
> |Marginal-PE+Skip|44.73%|60.94%|63.28%|66.60%|69.92%|0.5244
> |Absorbing-PE+Skip|**49.22%**|**70.90%**|**74.61%**|**77.93%**|**84.96%**|**0.5952**
>
> **The ablation on using only the skip connection alignment method.** Below, we see that the model with only the skip connections as alignment does not perform well, although it is better than the entirely unaligned model (4% top-1 in our Table 1). This is likely due to limited expressivity: the neural network does not get the alignment information at all as input in this case.
>
> |Model|Top-1|top-3|top-5|top-10|top-50|MRR|
> |-|-|-|-|-|-|-|
> |DiffAlign-Skip|9.77%|17.38%|24.61%|29.30%|32.23%|0.1517|
> |DiffAlign-PE+Skip (Laplacian)|**49.22%**|**70.90%**|**74.61%**|**77.93%**|**84.96%**|**0.5952**|
>
> **The ablation with the Gaussian positional encodings vs. Laplacian positional encodings (disentangling graph inductive biases from alignment)**. Below, we see that the difference of using the graph Laplacian positional encodings vs. matched Gaussian noise vectors as the positional encodings is not large in practice. This is evidence for the proposition that the alignment is the key inductive bias here, and the specific type of graph positional encoding is less important.
>
> |Model|Top-1|top-3|top-5|top-10|top-50|MRR|
> |-|-|-|-|-|-|-|
> |DiffAlign-PE-Gaussian|45.51%|68.55%|73.24%|75.78%|83.20%|0.5649|
> |DiffAlign-PE-Laplacian|46.09%|68.75%|72.85%|75.98%|83.98%|0.5686|
>
> We would like to thank everyone for the discussion and the constructive suggestions on additional experiments. We believe it has significantly improved the paper.

---

### Meta-Review · Area_Chair_qony · 2024-12-20

**Metareview:**

This work tackles an interesting problem setting of graph-to-graph translation, utilising discrete denoising diffusion models as well as equivariant structures. As the reviewers reaches a consensus that the paper is clearly written and contribution is substantial, with support of interesting chemical synthesis applications, it would be recommended for the work to appear in ICLR.

**Additional Comments On Reviewer Discussion:**

The discussion has been vibrant and in detail, resolving most of reviewers' discussion, while the manuscript has been improved accordingly.

---

### Decision · Program_Chairs · 2025-01-22

Accept (Poster)